

# Characteristics of biogenically-derived aerosols over the Amundsen Sea, Antarctica

Jinyoung Jung[1], Sang-Bum Hong[1], Meilian Chen[2,5], Jin Hur[2], Liping Jiao[3], Youngju Lee[1], Keyhong Park[1], Doshik Hahm[4], Jung-Ok Choi[1], Eun Jin Yang[1], Jisoo Park[1], Tae-Wan Kim[1], and SangHoon Lee[1]

[1]Korea Polar Research Institute, 26 Songdomirae-ro, Yeonsu-gu, Incheon 21990, Republic of Korea
[2]Sejong University, 209 Neungdong-ro, Gwangjin-gu, Seoul 05006, Republic of Korea
[3]Third Institute of Oceanography, State Oceanic Administration, Xiamen 361005, China
[4]Pusan National University, Busan 46241, Republic of Korea
[5]Environmental program, Guangdong Technion-Israel Institute of Technology, Shantou 515063, China

*Correspondence to:* Jinyoung Jung (jinyoungjung@kopri.re.kr)

**Abstract.** To investigate the influence of marine biological activity on aerosols, aerosol and seawater samples were collected over the Southern Ocean (43°S−70°S) and the Amundsen Sea (70°S−75°S) during the ANA06B cruise conducted in the austral summer of 2016 aboard the Korean icebreaker IBR/V *Araon*. Over the Southern Ocean, atmospheric methanesulfonic acid (MSA) concentration was low ($0.10 \pm 0.002$ µg m$^{-3}$), whereas its concentration increased sharply up to 0.57 µg m$^{-3}$ in the Amundsen Sea where *Phaeocystis antarctica* (*P. antarctica*), a producer of dimethylsulfide (DMS), was the dominant phytoplankton species. Unlike MSA, mean non-sea-salt sulfate (nss-SO$_4^{2-}$) concentration in the Amundsen Sea was comparable to that in the Southern Ocean, suggesting significant influences of marine biological activity on atmospheric sulfur species in the Amundsen Sea. Water-soluble organic carbon (WSOC) concentrations over the Southern Ocean and the Amundsen Sea varied from 0.048–0.16 µgC m$^{-3}$ and 0.070–0.18 µgC m$^{-3}$, with averages of $0.087 \pm 0.038$ µgC m$^{-3}$ and $0.097 \pm 0.038$ µgC m$^{-3}$, respectively. For water-insoluble organic carbon (WIOC), its mean concentrations over the Southern Ocean and the Amundsen Sea were $0.25 \pm 0.13$ µgC m$^{-3}$ and $0.26 \pm 0.10$ µgC m$^{-3}$, varying from 0.083–0.49 µgC m$^{-3}$ and 0.12–0.38 µgC m$^{-3}$, respectively. WIOC was the dominant organic carbon species in both the Southern Ocean and the Amundsen Sea, accounting for 73–75% of total aerosol organic carbon. WSOC and WIOC were highly enriched in the submicron sea spray particles, especially in the Amundsen Sea where biological productivity was much higher than the Southern Ocean. In addition, the submicron WIOC concentration was quite related to the relative biomass of *P. antarctica*, suggesting that extracellular polysaccharide mucus produced by *P. antarctica* was a significant factor affecting atmospheric WIOC concentration in the Amundsen Sea. The fluorescence properties of WSOC investigated using fluorescence excitation-emission matrix coupled with parallel factor analysis (EEM-PARAFAC) revealed that protein-like components were dominant in our marine aerosol samples, representing 69–91% of the total intensity. Protein-like components also showed positive relationships with the relative biomass of diatoms; however, they were negatively correlated with the relative biomass of *P. antarctica*. These results suggest that protein-like components are most likely produced as a result of biological processes of diatoms, which play a



crucial role in forming the submicron WSOC observed over the Southern Ocean and the Amundsen Sea, and that phytoplankton community structure is a significant factor affecting atmospheric organic carbon species. The results from this study provide significant new observational data on biogenically-derived sulfur and organic carbon species in the Amundsen Sea.

## 1 Introduction

Marine aerosols have been recognized to play an essential role in global climate due to their impact on the radiation budget and cloud microphysics by scattering solar radiation and acting as cloud condensation nuclei (CCN) (Andreae and Crutzen, 1997; O'Dowd et al., 2004). Considerable efforts have been devoted to investigating physicochemical characteristics of marine aerosols to elucidate the source and formation of CCN in the marine environment because of their importance for understanding the cloud-mediated effects of aerosols on climate (e.g., Hegg et al., 1991; O'Dowd et al., 1997; Vallina et al.,

2006; Furutani et al., 2008; Meskhidze and Nenes, 2010; Mochida et al., 2011; Gras and Keywood, 2017).

The biogeochemical cycle of sulfur between the marine atmosphere and the ocean has received much attention, especially after the proposal by Charlson et al. (1987) who postulated that the most significant source of CCN in the marine environment is non-sea-salt sulfate (nss-$SO_4^{2-}$) derived from atmospheric oxidation of dimethylsulfide (DMS). DMS produced by marine phytoplankton is the dominant sulfur species in ocean surface waters and is transported to the atmosphere through

sea-to-air flux (Bates et al., 1992). After emission to the atmosphere, DMS is oxidized by the hydroxyl (OH), nitrate ($NO_3$), and bromine oxide (BrO) radicals to form either methanesulfonic acid (MSA) or sulfur dioxide ($SO_2$) which is further oxidized to nss-$SO_4^{2-}$ (e.g., Andreae et al., 1985; Gondwe et al., 2004; Read et al., 2008). The conversion of DMS into nss-$SO_4^{2-}$ aerosols is an essential process because of the potential interaction of sulfur aerosols with incoming solar radiation, and their role on cloud microphysics which could result in a negative climate feedback mechanism (Legrand and Pasteur, 1998). However,

Quinn and Bates (2011) questioned the role of sulfur-containing aerosols derived from DMS in the climate feedback, but it is still clear that DMS emission contributes significantly to sulfur-containing aerosols acting as CCN over the oceans (Lana et al., 2012).

In addition to atmospheric sulfur species, marine organic aerosols have recently drawn increasing attention due to their potentially significant contribution to CCN budget over the remote ocean (e.g., O'Dowd et al., 2004; Quinn and Bates, 2011;

Gantt and Meskhidze, 2013). Ocean surface waters are enriched with small particulate organic materials including phytoplankton, bacteria, viruses, fragments of larger organisms and organic detritus (Quinn and Bates, 2011) as well as dissolved organic matter released or exuded by phytoplankton during growth, predation by grazing organisms and viral lysis (Biersmith and Benner, 1998). The organic fraction in surface waters can be broadly characterized as lipids, amino and fatty acids, mono- and polysaccharides and humic substances (Benner et al., 1992; Mochida et al., 2002; Gantt and Meskhidze,

2013), which can be emitted into the marine atmosphere with sea-salt particles through bubble bursting processes (Russell et al., 2010; Gantt et al., 2011). Previous studies have highlighted the significance of ocean-derived organic aerosol as a critical component of the aerosol-cloud-climate feedback system (e.g., Kawamura et al., 1996; Kanakidou et al., 2005; Kawamura et



al., 2010; Russell et al., 2010; Miyazaki et al., 2011; Quinn and Bates, 2011; Fu et al., 2015; Wilson et al., 2015; Miyazaki et al., 2016). For example, O'Dowd et al. (2004) revealed clear differences in ocean-derived organic carbon (OC) concentrations during periods of high and low biological activity, suggesting that OC derived from biological activity can enhance CCN concentration in the marine atmosphere, and that ocean-derived OC is a significant component of the aerosol-cloud-climate feedback system involving marine biota.

Although the importance of marine biogenic source contribution to CCN concentration has motivated numerous studies, quantitative measurements of the size-dependent composition of marine aerosols over high southern latitudes, especially the Antarctic remain sparse, due to its inaccessibility. Because of the difficulty in conducting a field observation, the sources and evolution of aerosols over the Antarctic are still a subject of many open questions (Giordano et al., 2017). It is, therefore, necessary to fill the data gap in the knowledge of biogenically-derived aerosols in the Antarctic to improve understanding of the effect of ocean ecosystem on the marine aerosol-cloud-climate system.

Polynyas, recurring areas of seasonally open water surrounded by sea ice in high latitude regions, often exhibit high primary productivity (e.g., Arrigo et al., 2003; Arrigo et al., 2012; Yager et al., 2012; Hahm et al., 2014) because they are the first polar marine systems to be exposed to the increasing springtime solar radiation (Arrigo and van Dijken, 2003; Criscitiello et al., 2013). Coastal polynyas surrounding Antarctica exhibit massive phytoplankton blooms during the austral summer (December–February), with most peaking in January. The productive polynyas are located in the Ross and Amundsen Seas while annual production in the polynyas of East Antarctica is generally low (Arrigo and van Dijken, 2003). The Amundsen Sea, located in the West Antarctica, hosts two coastal polynyas: the Amundsen Sea Polynya (ASP) and Pine Island Bay Polynya (PIBP) (Arrigo and van Dijken, 2003). The ASP is the most productive polynya (per unit area) among 37 identified coastal polynya systems in the Antarctic (Arrigo and van Dijken, 2003) due to the combined effects of the enhanced light condition (Park et al., 2017), the abundance of macronutrients, supply of iron (Fe) from melting sea ice and/or glaciers and continental shelf sediments resulted from the intrusion of relatively warm, salty, and nutrient-rich Circumpolar Deep Water (CDW) (Arrigo et al., 2012; Dutrieux et al., 2014; Sherrell et al., 2015). Consequently, the Amundsen Sea is an ideal place to monitor a direct link between biological production and local emissions of sulfur compounds and organics, not only because of its remoteness from anthropogenic activity but also because it is an area of exceptionally high seasonal primary production. However, little is known about the distributions of atmospheric sulfur and organic species in the Amundsen Sea.

To understand the influence of marine biological activities on atmospheric biogenically-derived aerosols in the Amundsen Sea, we have investigated characteristics of MSA, nss-$SO_4^{2-}$, water-soluble organic carbon (WSOC), and water-insoluble organic carbon (WIOC) in marine aerosols collected over the Amundsen Sea, Antarctica. Besides, we have carried out a hydrographic survey to examine the link between biological production and atmospheric sulfur and organic species in the Amundsen Sea. The objectives of this study are to (1) investigate the distributions of MSA and nss-$SO_4^{2-}$ over the Amundsen Sea, (2) examine the factors influencing atmospheric MSA concentration in the Amundsen Sea, (3) estimate the contribution of biogenic nss-$SO_4^{2-}$ to total nss-$SO_4^{2-}$, (4) investigate the distributions of atmospheric WSOC and WIOC over the Amundsen Sea, and (5)





estimate the source of atmospheric WSOC using a fluorescence technique. The results from this study provide quantitative insight into ambient levels of biogenically-derived sulfur and OC species in the marine boundary layer in the Amundsen Sea.

## 2 Method

Aerosol and seawater samples were collected during the ANA06B cruise conducted over the Southern Ocean and the Amundsen Sea, Antarctica aboard the Korean icebreaker IBR/V *Araon* (Fig. 1). The cruise started from Christchurch, New Zealand on 6 January 2016, sailed over the Amundsen Sea for 23 days (14 January−5 February), and returned to Christchurch, New Zealand on 24 February 2016. In this study, the Southern Ocean and the Amundsen Sea are defined as the regions between 43°S and 70°S, and between 70°S and 75°S, respectively. Although the cruise track covered the Southern Ocean (43°S−70°S) and the Amundsen Sea (70°S−75°S), a significant portion of our cruise was allocated to the ASP and near-ice shelf surveys (< 2 km from ice shelves) adjacent to the Dotson, the Getz, and the Pine Island ice shelves.

### 2.1 Aerosol collection

Two high-volume aerosol samplers (HV-1000R, Sibata Scientific Technology Ltd.) were installed on the front of the upper deck (20 m a.s.l.) and used to collect marine aerosols on pre-combusted (at 550°C for 6 hours) quartz filters (25 × 20 cm, QR-100, Sibata Scientific Technology Ltd.). Particle size selectors for PM2.5 and PM10 were installed to each aerosol sampler to collect fine (D < 2.5 μm) and coarse modes (2.5 μm < D < 10 μm) aerosols on the filters, respectively. A wind-sector controller was used to avoid contamination from the ship's exhaust during the cruise (Jung et al., 2013, 2014). The wind-sector controller was set to collect marine aerosol samples only when the relative wind directions were within plus or minus 100° relative to the ship's bow and the relative wind speeds were over 1 m s$^{-1}$. The flow rate was 1000 L min$^{-1}$ and the total sampling time was 3−4 days, representing a total sampling air volume of 1300−4500 m$^3$. After sampling, the filters were stored frozen at −24°C before chemical analysis. Procedural blanks (n = 4) were obtained by placing quartz filters in the aerosol sampler for 5 min on idle systems (i.e., no air flow through the filters) and processed as other aerosol samples. Meteorological variables (e.g., wind speed and wind direction) were continuously monitored by weather monitoring systems equipped on the research vessel during the cruise.

### 2.2 Chemical analysis

#### 2.2.1 Ionic species

The quartz filters, on which aerosols were collected, were cut into four equivalent subsamples. The subsamples were placed in acid-cleaned polypropylene bottles with the dusty side facing up. Fifty ml of Milli-Q water (> 18 MΩ cm$^{-1}$, Millipore Co.) was added to the bottles, and the bottles were covered using polypropylene screw caps. The subsamples were sonicated for 30 min. The extraction solution was then filtered through a 13-mm diameter, 0.45-μm pore size membrane filter (PTFE syringe filter, Millipore Co.). The filtrates were analyzed by ion chromatography (IC; ICS 2000, Thermo Scientific Dionex)



for anions (Cl⁻, MSA, NO₃⁻, and SO₄²⁻ ) and cations (Na⁺, NH₄⁺, K⁺, Mg²⁺, and Ca²⁺). The instrumental detection limits were: $Cl^-$ 0.05 µg L⁻¹, MSA 0.02 µg L⁻¹, $NO_3^-$ 0.02 µg L⁻¹, $SO_4^{2-}$ 0.02 µg L⁻¹, $Na^+$ 0.02 µg L⁻¹, $NH_4^+$ 0.14 µg L⁻¹, $K^+$ 0.16 µg L⁻¹, $Mg^{2+}$ 0.08 µg L⁻¹, and $Ca^{2+}$ 0.20 µg L⁻¹ (Hong et al., 2015). From replicate injections, the analytical precision was estimated to be < 5%. The concentrations of nss-$SO_4^{2-}$ were calculated as total $SO_4^{2-}$ minus $Na^+$ concentration times 0.2516, the $SO_4^{2-}$/$Na^+$ mass ratio in bulk seawater (Millero and Sohn, 1992).

### 2.2.2 Water-soluble organic carbon

Other subsamples of the quartz filters were ultrasonically extracted using the same method for ionic species measurements. The filtrates were analyzed by a total organic carbon (TOC) analyzer (Model TOC-L, Shimadzu Inc.) for determination of total dissolved organic carbon (TDOC), which was defined as water-soluble organic carbon (WSOC) in this study. In the analytical system, inorganic carbon was removed by acidifying the samples to pH 2 by 2 M HCl and sparging for 8 min before analysis of the total carbon content. Carbon dioxide ($CO_2$) derived from the conversion of TDOC by high temperature (680°C) catalytic oxidation was measured by a nondispersive infrared (NDIR) detector to quantify TDOC (Miyazaki et al., 2011). Milli-Q water and consensus reference material (CRM, 42–45 µM C for DOC, deep Florida Strait water obtained from University of Miami) were measured every sixth analysis to check the accuracy of the measurements. The procedural mean blank for WSOC was 180 µgC L⁻¹, which represented 14% of mean WSOC concentration in aerosols. The detection limit calculated as three times of the standard deviation of the procedural blanks was 21 µgC L⁻¹. The relative standard deviations of WSOC analysis for reproducibility test (at least three measurements per sample) was less than 3%.

### 2.2.3 Organic and elemental carbon

Concentrations of OC and elemental carbon (EC) were measured using the thermal optical transmission (TOT) method on a Sunset lab OC/EC analyzer (Model-4, USA) (Birch and Cary, 1996). The analytical procedures for OC and EC measurements are described in detail elsewhere (e.g., Miyazaki et al., 2011; Niu et al., 2012, 2013). In brief, a filter punch of 1.54 cm² was placed in the oven and heated in a completely pure helium environment up to 850°C to convert all OC into $CO_2$. For EC measurement, the oven was cooled to 550°C and then heated until the oven temperature stepping up back to 850°C under a 2% oxygen-containing helium environment. All $CO_2$ derived from the conversion of both OC and EC was measured by an NDIR detector. The equivalent OC concentration from filed blank accounted for ~15% of the average OC concentrations of the actual samples. Based on field blank uncertainties, detection limits for OC and EC were 0.1 µgC cm⁻² and 0.02 µgC cm⁻², respectively. In this study, WIOC was defined as the difference between OC and WSOC (i.e., WIOC = OC − WSOC) (Miyazaki et al., 2011). Using the propagating errors of each parameter, the uncertainty of WIOC concentration was estimated to be 5.8%.



### 2.2.4 Optical measurements and excitation-emission matrix coupled with parallel factor analysis

Absorption spectra of atmospheric WSOC were obtained from 240 to 800 nm on a Shimadzu 1800 ultraviolet-visible (UV-Vis) spectrophotometer (Shimadzu Inc.). Three-dimensional fluorescence excitation-emission matrices (EEMs) were scanned using a Hitachi F-7000 luminescence spectrometer (Hitachi Inc.) at the excitation/emission (Ex/Em) wavelengths of

250−500/280−550 nm. The excitation and the emission scans were set at 5 nm and 1 nm steps, respectively. The UV–Vis spectra were used for inner filter correction for the EEMs according to McKnight et al. (2001). Further details on the EEM measurements and the procedure of post-acquisition corrections are available in previous studies (Chen et al., 2010, 2017, 2018). The procedure for Raman Unit (RU) normalization can be found elsewhere (Lawaetz and Stedmon, 2009). Parallel factor (PARAFAC) modeling was performed using MATLAB 7.0.4 with the DOMFluor toolbox. All corrected EEMs of

aerosol samples were used for modeling. The number of fluorescent components was determined based on split-half validation. The biological index (BIX), an index of recent biological and autochthonous contribution, was calculated according to Huguet et al. (2009).

### 2.3 Hydrographic data

### 2.3.1 Chlorophyll-a

**2.3.1.1 In situ measurement**

Seawater samples were collected from 4 to 5 layers in the upper 100 m at 46 stations in the Amundsen Sea using a conductivity-temperature-depth (CTD) and a rosette system holding 24-10L Niskin bottles (SeaBird Electronics, SBE 911 plus) (Fig. 1). Seawater sample for chlorophyll-a (Chl-a) analysis was filtered through a GF/F filter (47 mm, Whatman), which was then extracted with 90% acetone for 24 hours. Chl-a was measured onboard using a fluorometer (Trilogy, Turner Designs,

USA) (Lee et al., 2016).

### 2.3.1.2 Satellite observation

Monthly composite of sea surface Chl-a concentration (mg m$^{-3}$) was obtained from Moderate Resolution Imaging Spectroradiometer (MODIS) Aqua data available from the Goddard Space Flight Center, NASA. The spatial resolution of the data was approximately 9 km per pixel (http://oceandata.sci.gsfc.nasa.gov).

**2.3.2 Dissolved organic carbon**

Seawater sample was drawn from the Niskin bottle by gravity filtration through an inline pre-combusted (at 550°C for 6 hours) Whatman GF/F filter held in an acid-cleaned (0.1 M HCl) polycarbonate 47 mm filter holder (PP-47, ADVANTEC). The filter holder was attached directly to the Niskin bottle spigot. The filtrate was collected in an acid-cleaned glass bottle and then distributed into two pre-combusted 20 ml glass ampoules with a sterilized serological pipette. Each ampoule was sealed





with a torch, quick-frozen, and preserved at –24°C until the analysis in our land laboratory. Analysis of dissolved organic carbon (DOC) was performed by high-temperature combustion using a Shimadzu TOC-L analyzer. Milli-Q water (blank) and consensus reference material (CRM, 42–45 µM C for DOC, deep Florida Strait water obtained from University of Miami) were measured every sixth analysis to check the accuracy of the measurements. Analytical errors based on the standard

deviations for replicated measurements (at least three measurements per sample) were within 5% for DOC.

### 2.3.3 Particulate organic carbon

For determination of particulate organic carbon (POC), seawater sample was drawn from the Niskin bottle into an amber polyethylene bottle. Known volumes (500 ml–1 L) of seawater were filtered onto pre-combusted Whatman GF/F filters (25 mm) under gentle vacuum at < 0.1 MPa. The filter samples were stored at –80°C until the analysis in our land laboratory.

Before POC analysis, the filter samples were freeze-dried and then exposed to HCl fumes for 24 hours in a desiccator to remove inorganic carbon from the samples. Measurement of POC was carried out with a CHN elemental analyzer (vario MACRO cube, Elementar, Germany). Acetanilide was used as a standard. The precision of these measurements was ± 4%.

### 2.3.4 Phytoplankton taxonomic composition

Phytoplankton taxonomic composition was assessed using High Performance Liquid Chromatography (HPLC) analysis

of accessory photosynthetic pigment concentrations. For the analysis of photosynthetic pigments, 1−2 L of seawater samples were filtered onto 47 mm Whatman GF/F filters and stored at −80°C until the analysis in our land laboratory. The filters were extracted with 3 ml of 100% acetone, ultrasonicated for 30 seconds and maintained under 4°C in dark for 15 hours. Debris was removed by filtering through 0.45 µm Teflon syringe filters. Before injection, the extracts were diluted with distilled water (1 ml of extract plus 0.3 ml of distilled water) to avoid the peak distortion of the first eluting pigments. Pigments were assessed

by HPLC analysis following the method of Zapata et al. (2000). Before analysis, the instrument (Agilent series 1200 chromatographic system, Germany) was calibrated using standard pigments (DHI, Denmark). A $C_8$ column (250 mm × 4.6 mm, 5 µm particle size, Agilent XDB-C8, USA) was used for pigment separation. HPLC pigment data was processed using CHEMTAX (CHEMical TAXonomy), a matrix factorization program to calculate the absolute Chl-a biomass of major algal groups. Twelve pigments were chosen for CHEMTAX analysis, and seven pigment algal groups were defined according to

Wright et al. (2010), including *Phaeocystis antarctica* (*P. antarctica*), diatoms, and cryptophytes.

### 3 Results and discussion

### 3.1 Atmospheric MSA over the Southern Ocean and the Amundsen Sea

Concentration of MSA in bulk (fine + coarse) aerosols during the cruise ranged from 0.038–0.57 µg m⁻³, with an average of 0.22 ± 0.14 µg m⁻³. About ~80% (median values for all data) of MSA was enriched in the fine mode aerosols. Over the

Southern Ocean (43°S−70°S, samples A1–A2), the MSA concentration was relatively low (mean: 0.10 ± 0.002 µg m⁻³),



whereas its concentration over the Amundsen Sea (70°S–75°S, samples A3–A9) increased sharply up to 0.35 µg m$^{-3}$ and showed high spatial variability (range: 0.096–0.57 µg m$^{-3}$, mean: 0.28 ± 0.15 µg m$^{-3}$), with the highest value observed in the ASP (Fig. 2a). As expected, however, the MSA concentration gradually decreased from 0.29 µg m$^{-3}$ to 0.038 µg m$^{-3}$ with distance from the ASP (i.e., from the Amundsen Sea to the Southern Ocean, samples, A10–A14, mean: 0.18 ± 0.11 µg m$^{-3}$).

The MSA concentration from this study was in reasonable agreement with previously published results obtained over the regions near the Antarctic continent during the austral summer when air masses originated form the western Antarctic coastal regions (60°S−70°S, 0.05–0.26 µg m$^{-3}$, Chen et al., 2012). However, our mean MSA concentration in the Amundsen Sea was about 1.8–4.4 times higher than those observed at Antarctic research stations during the austral summer, such as Palmer Station (64.77°S, 64.05°W, 0.122 ± 0.127 µg m$^{-3}$, December 1990–March 1991, Savoie et al., 1993), Halley Station (75°35′S, 26°19′W,

0.14 µg m$^{-3}$, January–February 2005, Read et al., 2008), Neumayer Station (70°39′S, 8°15′W, 0.154 ± 0.077 µg m$^{-3}$, monthly mean in January from 1983–1995, Minikin et al., 1998), and Dumont D'Urville Station (66°40′S, 140°1′E, 0.063 ± 0.019 µg m$^{-3}$, monthly mean in January from 1991–1995, Minikin et al., 1998). Although the mean MSA concentration was much higher than those observed at the Antarctic research stations, the maximum of MSA in the Amundsen Sea was comparable to the highest concentration measured at Palmer Station (0.57 µg m$^{-3}$, Savoie et al., 1993), implying that the source strength for MSA

is greater in the Amundsen Sea. These differences in atmospheric MSA concentrations among coastal Antarctic regions presumably reflect regional differences in factors controlling MSA abundance in the marine atmosphere, such as spatial variations of phytoplankton bloom, phytoplankton taxonomic composition, emission flux of DMS, atmospheric oxidative capacity, atmospheric transport and removal (Minikin et al., 1998; Gondwe et al., 2004; Chen et al., 2012; Criscitiello et al., 2013; Jung et al., 2014).

**3.2 Factors influencing atmospheric MSA concentration over the Amundsen Sea**

Various factors appear to influence atmospheric MSA concentration in the Amundsen Sea. According to Arrigo and van Dijken (2003), 5 years (1997–2002) averaged Chl-a level in the ASP during the month of January (6.98 ± 3.32 mg m$^{-3}$) is more than 2 times higher than that in the Ross Sea Polynya (2.67 ± 0.82 mg m$^{-3}$). Indeed, satellite ocean color images (http://oceancolor.sci.gsfc.nasa.gov) exhibited persistently high Chl-a levels in the ASP during the sampling period (Fig. 3),

implying a strong influence of biogenic source on atmospheric MSA in the Amundsen Sea.

In addition to the high productivity, phytoplankton taxonomic assemblages could be a significant factor influencing atmospheric MSA abundance. In general, phytoplankton taxonomic abundances in the Amundsen Sea are dominated by haptophyte *P. antarctica* (e.g., Lee et al., 2016; Yager et al., 2016), which produces large amounts of DMS in Antarctic waters (Liss et al., 1994; Schoemann et al., 2005). During the cruise, *P. antarctica* was the dominant phytoplankton species in the

upper 50 m, accounting for 42 ± 19% of phytoplankton biomass (Chl-a), with lesser abundances of diatoms (39 ± 17%) found throughout the polynya and sea ice zone (see Supplement, Fig. S1). In addition, extremely high concentrations (> 150 nM) and fluxes (85 ± 119 µmol m$^{-2}$ d$^{-1}$) of DMS have been observed in the region where phytoplankton assemblages were dominated





by *P. antarctica* during the cruise (Kim et al., 2017), which are consistent with previous results observed in the Amundsen Sea (Tortell et al., 2012).

We investigated the relationships between atmospheric MSA concentration and the variables mentioned above, to examine factors influencing atmospheric MSA concentration in the Amundsen Sea. Atmospheric MSA concentration showed no significant relationship with either in situ sea surface Chl-a concentration ($r = 0.029$, $p > 0.05$) (Fig. S2) or the relative biomass of *P. antarctica* ($r = 0.30$, $p > 0.05$). This suggests that Chl-a concentration and phytoplankton taxonomic composition are not direct factors determining atmospheric MSA concentration over the Amundsen Sea. On the other hand, DMS flux (data from Kim et al., 2017) averaged for the duration of each aerosol sampling showed a somewhat similar variation trend to that of atmospheric MSA concentration (Fig. 4a), but the correlation between atmospheric MSA and DMS flux was weak ($r = 0.18$, $p > 0.05$, Fig. 4b); however, if the single outlying point (i.e., assuming that relatively low atmospheric MSA concentration compared to the highest DMS flux is affected by the highest mean wind speed during the sampling duration of sample A6) is excluded from Fig. 4b, the correlation coefficient (r) between atmospheric MSA concentration and DMS flux increases to 0.78 ($p < 0.05$). DMS concentration in surface water showed a similar variation trend to that of DMS flux since DMS concentration in surface water is a key factor determining sea–air DMS flux. Similar result was found between atmospheric MSA and DMS concentrations in surface water, showing a weak positive correlation ($r = 0.20$, $p > 0.05$). However, the relationship between atmospheric MSA and DMS concentrations in surface water become statistically significant ($r = 0.77$, $p < 0.05$) when the single outlying point (i.e., sample A6) is excluded (not shown). Given the lifetime of DMS is approximately 1–2 days (Kloster et al., 2006; Read et al., 2008), our results revealed that the local sea–air DMS flux affected directly atmospheric MSA concentration in the Amundsen Sea, and that the higher atmospheric MSA concentrations observed over the Amundsen Sea compared to those over the Southern Ocean and in coastal Antarctic regions were attributed to the higher DMS concentrations produced by *P. antarctica* and to the higher DMS fluxes in the Amundsen Sea.

### 3.3 Atmospheric nss-SO$_4^{2-}$ over the Southern Ocean and the Amundsen Sea

Concentration of nss-SO$_4^{2-}$ in bulk (fine + coarse) aerosols during the cruise ranged from 0.30–0.87 µg m$^{-3}$, with ~79% (median values for all data) of nss-SO$_4^{2-}$ being present on fine mode aerosols (Fig. 2b). Mean concentration of nss-SO$_4^{2-}$ during the cruise was $0.61 \pm 0.17$ µg m$^{-3}$. Over the Southern Ocean (43°S−70°S, samples A1–A2 and A11–A14), the nss-SO$_4^{2-}$ concentration ranged from 0.30–0.87 µg m$^{-3}$ (mean: $0.60 \pm 0.23$ µg m$^{-3}$), whereas its concentration over the Amundsen Sea (70°S–75°S, samples A3–A10) varied from 0.49–0.81 µg m$^{-3}$ (mean: $0.62 \pm 0.10$ µg m$^{-3}$). Unlike MSA, the mean nss-SO$_4^{2-}$ concentration in the Amundsen Sea was comparable to that in the Southern Ocean, although the variation trend of nss-SO$_4^{2-}$ in the Amundsen Sea was similar to that of MSA, suggesting that nss-SO$_4^{2-}$ was affected by both marine and anthropogenic sources. Indeed, nss-SO$_4^{2-}$ showed a strong correlation ($r = 0.98$, $p < 0.01$) with MSA in the Amundsen Sea, whereas the relationship in the Southern Ocean was statistically insignificant ($r = 0.51$, $p > 0.05$) (Fig. S3), suggesting that the local emission of DMS is a significant source of nss-SO$_4^{2-}$ in the Amundsen Sea and that nss-SO$_4^{2-}$ in the Amundsen Sea was formed by the condensation of DMS products onto existing particles (Sanchez et al., 2018). On the other hand, the statistically



insignificant relationship between nss-SO$_4^{2-}$ and MSA could result from strong influence of anthropogenic sources and low biological activity in the Southern Ocean.

Surprisingly, our mean nss-SO$_4^{2-}$ concentration in the Amundsen Sea was about 1.5 and 2.5 times higher than those observed at American Samoa (14.25°S, 170.58°W, 0.41 ± 0.17 µg m$^{-3}$, seasonal average (December–February) from 1990–
1992, Savoie et al., 1994) and over the South Pacific (8°S–55°S, mean: 0.25 ± 0.17 µg m$^{-3}$, range: 0.094–0.62 µg m$^{-3}$, January–March 2009, Jung et al., 2014), respectively. In addition, the mean nss-SO$_4^{2-}$ concentration in the Amundsen Sea was also a factor of 1.6–4.4 higher than those observed at Palmer Station (0.24 ± 0.16 µg m$^{-3}$, December 1990–March 1991, Savoie et al., 1993), Halley Station (0.14 ± 0.017 µg m$^{-3}$, monthly mean in January from 1991–1993, Legrand and Pasteur, 1998), Neumayer Station (0.38 ± 0.13 µg m$^{-3}$, monthly mean in January from 1983–1995, Minikin et al., 1998), and Dumont D'Urville
Station (0.34 ± 0.039 µg m$^{-3}$, monthly mean in January from 1991–1995, Minikin et al., 1998) during the austral summer. Considering the high MSA concentration and DMS flux in the Amundsen Sea as mentioned above in section 3.2, the high nss-SO$_4^{2-}$ concentration in the Amundsen Sea from this study is likely affected much more by biogenic than anthropogenic sources.

### 3.4 Contribution of biogenically-derived nss-SO$_4^{2-}$ to total nss-SO$_4^{2-}$

While nss-SO$_4^{2-}$ can have multiple sources, including DMS oxidation, volcanic and industrial sulfur emissions, MSA
is formed exclusively from DMS (Gondwe et al., 2003). Thus MSA was proposed as a useful tracer to distinguish between marine biogenic and anthropogenic nss-SO$_4^{2-}$ (Legrand and Pasteur, 1998). Considerable efforts have been devoted to investigating the contribution of biogenically-derived atmospheric sulfur species over the various geographical locations by using the MSA/nss-SO$_4^{2-}$ ratio observed in aerosols, which shows the spatial (higher ratios at high latitudes) and seasonal variability (summer maxima and winter minima) (e.g., Savoie and Prospero, 1989; Prospero et al., 1991; Bates et al., 1992;
Arimoto et al., 2001; Savoie et al., 2002; Yang et al., 2009; Jung et al., 2014; Legrand et al., 2017). Mungall et al. (2018), however, pointed out that the MSA/nss-SO$_4^{2-}$ ratio could have limitation that may preclude its use in quantitatively unravelling the chemical and biological processes at play in the marine boundary layer due to the conversion of MSA to nss-SO$_4^{2-}$ by OH radical in aerosol particles (the order of days to weeks), although it remains useful as a qualitative indicator of marine biological influence.

During the cruise, the MSA/nss-SO$_4^{2-}$ ratio in bulk aerosols varied from 0.12 to 0.70 (mean: 0.35 ± 0.17), with lower ratios in marine aerosols collected over the Southern Ocean (range: 0.12–0.51, mean: 0.26 ± 0.14) and higher values over the Amundsen Sea (range: 0.20–0.70, mean: 0.44 ± 0.16), showing a similar variation trend to that of MSA (Fig. 5a). This result suggests that atmospheric MSA plays a key role in the variation in MSA/nss-SO$_4^{2-}$ ratio over the Southern Ocean and the Amundsen Sea during the austral summer, and that atmospheric nss-SO$_4^{2-}$ observed in the Southern Ocean was affected more
by anthropogenic sources since atmospheric nss-SO$_4^{2-}$ concentrations in the Southern Ocean were quite comparable to the values in the Amundsen Sea.

It is worth noting that this study was carried out over the Southern Ocean and the Amundsen Sea during the austral summer. Consequently, to estimate the contribution of biogenically-derived nss-SO$_4^{2-}$ to total nss-SO$_4^{2-}$, it is required to use



the MSA/nss-$SO_4^{2-}$ ratios observed through long-term monitoring in the Southern Ocean and the Amundsen Sea during the austral summer. Nevertheless, due to the lack of long-term monitoring data for MSA/nss-$SO_4^{2-}$ ratios in the Southern Ocean and Amundsen Sea, we applied the MSA/nss-$SO_4^{2-}$ ratio of 0.508 observed during the austral summer through long-term monitoring at Palmer Station (Savoie et al., 1993) which is geographically closest to the Amundsen Sea among the Antarctic

coastal stations. Based on this ratio, mean concentrations of biogenically-derived nss-$SO_4^{2-}$ over the Southern Ocean and the Amundsen Sea were estimated to be 0.31 ± 0.19 µg m$^{-3}$ (range: 0.074–0.57 µg m$^{-3}$) and 0.56 ± 0.30 µg m$^{-3}$ (range: 0.19–1.1 µg m$^{-3}$), accounting for 52 ± 28% (range: 24–101%) and 86 ± 32% (range: 39–138%) of total nss-$SO_4^{2-}$, respectively (Fig. 5b). As expected, the concentration and contribution of biogenic nss-$SO_4^{2-}$ were much higher in the Amundsen Sea, indicating that marine biogenic emission of DMS dominates the atmospheric budget of nss-$SO_4^{2-}$ in the Amundsen Sea. However, when

the MSA/nss-$SO_4^{2-}$ ratio by Savoie et al. (1993) was used in this study, the concentration and contribution of biogenically-derived nss-$SO_4^{2-}$ for samples, where the MSA/nss-$SO_4^{2-}$ ratios were higher than 0.508 (i.e., samples A3, A8–A10), were somewhat overestimated, so that anthropogenic nss-$SO_4^{2-}$ concentrations (i.e., total nss-$SO_4^{2-}$ – biogenically-derived nss-$SO_4^{2-}$) in those samples were underestimated, showing negative values. Nevertheless, anthropogenic nss-$SO_4^{2-}$ in the other samples showed a similar variation trend to that of nitrate ($NO_3^-$, an indicator of anthropogenic contribution) (Savoie et al., 2002),

varying from 0.10–0.63 µg m$^{-3}$ (mean: 0.34 ± 0.18 µg m$^{-3}$) in the Southern Ocean, and 0.13–0.30 µg m$^{-3}$ (mean: 0.19 ± 0.079 µg m$^{-3}$) in the Amundsen Sea (Figs. 5c and 5d). The contributions of anthropogenic nss-$SO_4^{2-}$ in the Southern Ocean and the Amundsen Sea were estimated to be 56 ± 19% (range: 33–76%) and 35 ± 18% (range: 23–61%), respectively. Our mean concentration of anthropogenic nss-$SO_4^{2-}$ in the Southern Ocean was a factor of 1.4 higher than the result by Jung et al. (2014) who reported that mean concentration of nss-$SO_4^{2-}$ observed over the South Pacific (10°S−55°S, January–March 2009) was

0.25 ± 0.17 µg m$^{-3}$. In comparison, our mean concentration of anthropogenic nss-$SO_4^{2-}$ in the Amundsen Sea was a factor of 1.3 lower than the result by Jung et al. (2014). The high anthropogenic nss-$SO_4^{2-}$ concentrations were estimated during the collection of aerosol samples A1 and A13 (Fig. 5c). Similar to anthropogenic nss-$SO_4^{2-}$, the high $NO_3^-$ concentrations were observed during the collection of those samples, when air masses originated from the Southern Ocean and thereafter swept over large regions of New Zealand (Figs. 5d and S4). These results indicate that these air masses were most likely affected by

strong anthropogenic sources, which could contribute to the high anthropogenic nss-$SO_4^{2-}$ and $NO_3^-$ concentrations in the Southern Ocean. In this study, the contribution of biogenically-derived nss-$SO_4^{2-}$ to total nss-$SO_4^{2-}$ was estimated using only results observed during the limited sampling period, but our results would be valuable for filling the data gap, especially for the Amundsen Sea, and be helpful for validation of modeling of sulfur-containing aerosols.

### 3.5 Atmospheric WSOC and WIOC over the Southern Ocean and the Amundsen Sea

Concentration of WSOC in bulk (fine + coarse) aerosols during the cruise ranged from 0.048–0.18 µgC m$^{-3}$, with an average of 0.092 ± 0.037 µgC m$^{-3}$ (Fig. 6a). The WSOC concentrations over the Southern Ocean (43°S−70°S, samples A1– A2 and A11–A14) and the Amundsen Sea (70°S–75°S, samples A3–A9) varied from 0.048–0.16 µgC m$^{-3}$ and 0.070–0.18 µgC m$^{-3}$, with averages of 0.087 ± 0.038 µgC m$^{-3}$ and 0.097 ± 0.038 µgC m$^{-3}$, respectively. For WIOC, its mean concentrations



over the Southern Ocean and the Amundsen Sea were $0.25 \pm 0.13$ µgC m$^{-3}$ and $0.26 \pm 0.10$ µgC m$^{-3}$, varying from 0.083–0.49 µgC m$^{-3}$ and 0.12–0.38 µgC m$^{-3}$, respectively (Fig. 6b). We expected much higher WSOC and WIOC concentrations in the Amundsen Sea than the Southern Ocean because of extremely high Chl-a concentration in the Amundsen Sea (Fig. 3). However, no significant differences of mean WSOC and WIOC concentrations were found between the Southern Ocean and the

Amundsen Sea, suggesting that Chl-a concentration is not a direct factor controlling atmospheric OC concentration in our study area, although a significant correlation between atmospheric OC and Chl-a concentrations was observed in the Austral Ocean (Amsterdam Island, 37°48′S, 77°34′E, Sciare et al., 2009).

Both WSOC and WIOC mainly existed in fine mode particles, and the enrichment of WSOC and WIOC in fine mode particles were ~93% and ~74%, respectively (median value for all data). During the cruise, WIOC was the dominant OC

species in both the Southern Ocean and the Amundsen Sea, accounting for 75% and 73% of total aerosol organic carbon, respectively (Fig. 6c). These results were consistent with the previous studies. For example, O'Dowd et al. (2004) observed that the contribution of OC fraction to the submicrometer aerosol mass increased from 15% to 63% between the low and high biological activity periods in the North Atlantic, and that WIOC and WSOC in the fine mode (0.06–1 µm) accounted for 45% and 18% of OC during bloom periods, respectively. Moreover, Facchini et al. (2008) reported that OC observed through a

bubble bursting experiment and a field measurement (at Mace Head) was mainly water-insoluble, accounting for $77 \pm 5\%$ of the primary marine aerosol fraction in the submicron size range, and that WIOC consisted of colloids and aggregates exuded by phytoplankton. Given that atmospheric WIOC is mechanically produced through bubble bursting processes from hydrophobic organic matter that accumulates in the ocean surface (Facchini et al., 2008; Gantt et al., 2011; Miyazaki et al., 2011), the dominance of WIOC suggests that the bubble bursting process by local wind speeds is a significant formation

mechanism of atmospheric WIOC in our study area.

Our mean values of WSOC and WIOC in the Amundsen Sea were comparable to the results by Sciare et al. (2009) who reported that the mean concentrations of WSOC and WIOC observed at Amsterdam Island (37°48′S, 77°34′E) during the austral summer (January) were $0.083 \pm 0.028$ µgC m$^{-3}$ and $0.19 \pm 0.062$ µgC m$^{-3}$, respectively. It is worth noting that atmospheric WSOC and WIOC show seasonal variations, with maximum values during austral summer (January) and

minimum concentrations during winter (Sciare et al., 2009). These variations are attributable to pronounced seasonal variations in biogenic marine productivity. Given that our study was carried out during the austral summer, the concentrations of WIOC and WSOC from this study would be considered as maximum values in the Amundsen Sea. Although our results were supported by previous studies as mentioned above, the spatial variability in WSOC and WIOC concentrations has been observed over various oceanic regions by previous studies. For instance, Fu et al. (2005) reported that atmospheric OC species

concentrations observed at Alert, in the Canadian High Arctic were 0.186 µgC m$^{-3}$ (range: 0.041–0.30 µgC m$^{-3}$) for WSOC and 0.068 µgC m$^{-3}$ (range: 0.022–0.12 µgC m$^{-3}$) for WIOC. The results by Fu et al. (2005) were 1.9 times higher and 3.8 times lower than our mean concentrations of WSOC and WIOC in the Amundsen Sea, respectively. In addition, Miyazaki et al. (2011) reported that mean concentrations of WSOC and WIOC observed over the western North Pacific (40°N–44°N) were $0.65 \pm 0.27$ µgC m$^{-3}$ and $0.56 \pm 0.19$ µgC m$^{-3}$, which were a factor of 6.7 and 2.2 higher than those in the Amundsen Sea,



respectively. These differences in WSOC and WIOC concentrations among the oceanic regions presumably reflect regional differences in factors influencing atmospheric WSOC and WIOC concentrations, such as source strength for volatile organic compounds emitted from biogenic sources (BVOCs), atmospheric oxidative capacity (e.g., OH, $NO_3$ and ozone), meteorological condition (e.g., wind speed), DOC and POC concentrations in sea surface water, atmospheric transport and removal (e.g., Facchini et al., 1999; Kanakidou et al., 2005; Sun et al., 2006).

## 3.6 Factors influencing atmospheric WSOC and WIOC concentrations in sea spray aerosols

Breaking waves on the ocean surface generate air bubbles that scavenge organic matter from seawater. When injected into the atmosphere, these bubbles burst, yielding sea spray aerosols enriched in organic matter, relative to seawater (Quinn et al., 2014). Sea spray aerosols have been defined as the hydrated droplets encapsulating dissolved sea-salt and entrained organic matter (O'Dowd et al., 2008). Previous studies have revealed that organic matter is enriched in sea spray aerosol particles produced by bubble bursting processes in the fine and ultrafine aerosol size fractions, suggesting that sea spray aerosol particles have an important role in transferring organic matter from the sea surface to the atmosphere (Facchini et al., 2008; O'Dowd et al., 2008). During the cruise, ~76% of $Na^+$, a tracer of sea spray, was enriched in the coarse mode particle (Fig. 7a). Moreover, statistically significant relationships were found between mean wind speed and $Na^+$ concentrations in fine ($r = 0.54$, $p < 0.05$) and coarse ($r = 0.69$, $p < 0.01$) mode aerosols, reflecting that $Na^+$ was formed from bubble bursting by local wind speed. Although $Na^+$ was predominantly associated with the coarse mode particle, WSOC and WIOC were highly enriched in the fine mode sea spray particles, especially in the Amundsen Sea where biological productivity was much higher than the Southern Ocean (Fig. 7b). In the Southern Ocean, the WSOC/$Na^+$ ratio in the fine mode particles varied from 0.045–0.40, whereas the WSOC/$Na^+$ ratio in the Amundsen Sea ranged from 0.17–0.89. The average WSOC/$Na^+$ ratio in the Amundsen Sea ($0.40 \pm 0.24$) was substantially higher than that in the Southern Ocean ($0.16 \pm 0.12$). For the WIOC/$Na^+$ ratio in the fine mode particles, similar results were observed. The WIOC/$Na^+$ ratios in the Southern Ocean and the Amundsen Sea varied from 0.038–0.97 and 0.26–2.4, with averages of $0.35 \pm 0.31$ and $0.91 \pm 0.73$, respectively; however, WIOC was much more enriched in the fine mode sea spray particles than WSOC, suggesting that bubble bursting at the ocean surface is a major source of atmospheric WIOC and that WIOC is more accumulated in the sea surface water. These results show that the higher marine biological activities in the Amundsen Sea can be a significant factor leading to the higher enrichment of OC in sea spray aerosols, indicating a linkage between OC emission and biological activities (Fig. 3).

In addition to marine biological activities, the sea surface microlayer (SML) coverage, which directly couples biological processes to atmosphere-ocean exchange, could be an important factor influencing organic mass fraction of sea spray aerosol since organic matter is concentrated in the SML relative to the bulk seawater (Russell et al., 2010; Miyazaki et al., 2016). Recently, Gantt et al. (2011) revealed that the organic mass fraction of sea spray aerosol is inversely correlated with sea surface wind speed because the coverage of sea surface by SML increases with decreasing sea surface wind speed. During the sampling period, both WSOC/$Na^+$ and WIOC/$Na^+$ ratios showed inverse relationships with mean wind speed ($r = 0.82$, $p < 0.01$ and $r = 0.76$, $p < 0.01$, respectively) (Figs. 7c and 7d). The highest WSOC/$Na^+$ and WIOC/$Na^+$ ratios were observed in the aerosol



sample (i.e., A5) collected under calm conditions (i.e., mean wind speed was lowest (4.4 m s$^{-1}$)), and the WSOC/Na$^+$ and WIOC/Na$^+$ ratios drastically decreased with increasing mean wind speed. According to Gantt et al. (2011), an increase in wind speed above 3–4 m s$^{-1}$ causes a rapid decrease in the SML coverage, and the wave breaking thoroughly mixes the SML with the underlying water resulting in the homogeneous water column when the wind speeds exceed 8 m s$^{-1}$. Therefore, the inverse

relationships between WSOC/Na$^+$, WIOC/Na$^+$ ratios and the mean wind speed suggest that the SML coverage affected the organic mass fractions of sea spray aerosols in our study region.

We found more pieces of evidence which reflect that both marine biological activities and the SML coverage can affect the organic mass fraction of sea spray aerosol. Sea surface DOC and POC concentrations have been used as a proxy for the organic mass fraction of sea spray aerosol (e.g., Facchini et al., 2008; Russell et al., 2010; Gantt et al., 2011; Quinn and Bates,

2011; Quinn et al., 2014) since DOC and POC are derived from biological activities in the ocean (Hansell and Carlson, 2001; Henson et al., 2012). As shown Figs. 7e and 7f, both the WSOC/Na$^+$ and the WIOC/Na$^+$ ratios showed strong positive relationships with sea surface DOC concentrations measured in the Amundsen Sea when the mean wind speed exceeded 7 m s$^{-1}$; however, when the mean wind speed was lowest (i.e., aerosol sample A5), both the WSOC/Na$^+$ and the WIOC/Na$^+$ ratios sharply increased, although sea surface DOC concentration was low. These results indicate that both biological activity and

the SML coverage are crucial factors influencing the organic mass fraction of sea spray aerosol as mentioned above. Unlike the relationships with sea surface DOC concentration, the WSOC/Na$^+$ and the WIOC/Na$^+$ ratios showed no statistically significant relationships with sea surface POC concentrations in the Amundsen Sea when the aerosol sample A5 was excluded ($r = 0.30$, $p > 0.5$ and $r = 0.17$, $p > 0.5$, respectively; not shown). Similar results have been revealed by Quinn et al. (2014) who found no significant relationship between Chl-a (or POC) and organic mass enrichment in sea spray aerosol in both high- and

low-chlorophyll waters. Quinn et al. (2014) also concluded that there is a large reservoir of OC in surface seawater that results in the enrichment of organic matter in sea spray aerosol. They further suggested that this reservoir of OC is uncoupled from and overwhelms any influence of local biological activity as measured Chl-a (or POC) over large ocean regions since a variability in DOC is often uncorrelated with Chl-a or primary production. Phytoplankton release or exude organic matter during their growth, predation by grazing organisms and viral lysis. Phytoplankton exudates include exopolymer gels

consisting of polysaccharides, which are insoluble, thermally stable, highly surface active, highly hydrated and readily sequester dissolved organic matter (Quinn and Bates, 2011 and references therein). Besides, *P. antarctica*, which was a dominant phytoplankton species in the Amundsen Sea (see section 3.2), generates a substantial amount of extracellular polysaccharide mucus in its colonial matrix, and it has been suggested that large amounts of this material ultimately enter the DOC pool (Smith et al., 1998). Indeed, the submicron WIOC concentration showed a good correlation ($r = 0.87$, $p < 0.05$)

with the relative biomass of *P. antarctica* in the Amundsen Sea (Fig. S5). Therefore, the significant relationships between the WSOC/Na$^+$, the WIOC/Na$^+$ ratios and DOC concentration in the Amundsen Sea suggest that the organic mass fraction of sea spray aerosol could be uncoupled with Chl-a and phytoplankton biomass, but related to biological activities under certain meteorological conditions. Since sea surface DOC concentration was measured only in the Amundsen Sea, we cannot discuss our aerosol data set collected in the Southern Ocean. However, given the higher WSOC/Na$^+$ and WIOC/Na$^+$ ratios in the



Amundsen Sea, the inverse relationships with the mean wind speed and the strong positive relationships with surface DOC concentrations, the WSOC and WIOC mass fractions of sea spray aerosols most likely resulted from complex linkage between meteorological conditions and biological activities rather than the sole effect of biological activities in the Amundsen Sea.

## 3.7 Relationships of WIOC and WSOC with $Na^+$ over the Southern Ocean and the Amundsen Sea

Organic components in the marine surface layer are emitted into the marine boundary layer as sea spray via bubble bursting and breaking waves (Fu et al., 2015). Organic aerosols in the marine boundary layer are proposed to have different sources that can be broadly classified as primary organic aerosol (POA) and secondary organic aerosol (SOA) (Gantt et al., 2011). POA is emitted in the marine boundary layer as sea spray via bubble bursting and breaking waves, whereas SOA is derived from the condensation of precursor BVOCs emitted by phytoplankton and macroalgae on preexisting particles (e.g.,

sea spray aerosols) or from the chemical transformation of primary or secondary components present in the condensed phase (Rinaldi et al., 2010). Because sea spray aerosols are emitted as a result of wind-driven bubble bursting, correlations of organic aerosol mass concentrations with sea spray aerosols (i.e., $Na^+$), whose concentration are related to local wind speeds, have been used to attribute their origin to marine sources because submicron $Na^+$ is known to form primary aerosols from evaporating seawater droplets (Russell et al., 2010). In this study, we investigated relationships of WIOC and WSOC with $Na^+$

concentrations in the fine modes since both WIOC and WSOC were primarily associated with the fine mode particles (Figs. 6a and 6b). As shown in Fig. 8a, the submicron WIOC showed no statistically significant relationships with submicron $Na^+$ over the Southern Ocean and the Amundsen Sea, although WIOC was highly enriched in the fine mode sea spray particles (Fig. 7b). Similarly, Boreddy et al. (2018) found no correlation between sea-salt and WIOC in the western North Pacific. Theses insignificant relationships between WIOC and $Na^+$ in the fine modes could result from the differences in local wind

speeds and local biological activities, such as sea surface DOC concentration, because wind speed, a key factor determining sea spray aerosols, controls the local flux rather than local concentration of marine particles (Monahan and O'Muircheartaigh, 1986). Our results are further supported by the study of Ceburnis et al. (2008), who found WIOC and sea-salt exhibited upward fluxes observed through gradient flux measurements, suggesting a primary production mechanism for WIOC. Although we found no significant relationships between WIOC and $Na^+$ in the fine modes, the high enrichment of WIOC in the fine mode

sea spray particles (Fig. 7b) and the strong correlation of submicron WIOC/$Na^+$ ratio with sea surface DOC concentration when the mean wind speed exceeded 7 m s$^{-1}$ (Fig. 7f) indicate that WIOC was predominantly of primary origin (Ceburnis et al., 2008).

Unlike the relationship between WIOC and $Na^+$, submicron WSOC showed a strong correlation (r = 0.94, p < 0.01) with submicron $Na^+$ in the Amundsen Sea (Fig. 8b). In addition, we also found a significant correlation (r = 0.93, p < 0.01)

between WSOC and MSA concentrations in the Amundsen Sea (Fig. 8c). However, in the Southern Ocean, WSOC showed no significant relationship with submicron $Na^+$ or MSA. MSA is produced by atmospheric oxidation of DMS, which is released as a gas phase from marine biological activities and thus can be used as an indicator of secondary aerosols of marine biogenic origin (Miyazaki et al., 2011). As described in section 3.2, Kim et al. (2017) observed extremely high DMS concentrations (>



150 nM) in surface water during our cruise, and MSA concentration showed a strong correlation with DMS flux in the Amundsen Sea. Consequently, the strong correlations between WSOC, Na$^+$ and MSA in the Amundsen Sea implies that the Amundsen Sea that has the most productive polynya in the Antarctic is a strong source region of BVOCs, and that WSOC was formed by the condensation of BVOCs released from sea surface onto preexisting submicron sea spray aerosols through gas-

to-particle conversion due to a higher surface-to-volume ratio of submicron aerosols (Romakkaniemi et al., 2011). On the other hand, the poor correlations between WSOC, Na$^+$, and MSA in the Southern Ocean implies the differences in local source strength of BVOCs and that the presence of DMS in seawater and its subsequent oxidation to MSA were not necessarily linked to the formation of submicron WSOC over the Southern Ocean (Miyazaki et al., 2016).

**3.8 Fluorescence properties of WSOC over the Southern Ocean and the Amundsen Sea**

10        Fluorescence excitation-emission matrix coupled with parallel factor analysis (EEM-PARAFAC) has been widely used to investigate the sources and optical properties of dissolved organic matter in terrestrial and oceanic systems (e.g., Coble, 1996, 2007; Stedmon et al., 2003; Yamashita et al., 2011; Retelletti Brogi et al., 2018). Moreover, recent field studies demonstrated that EEM-PARAFAC provides useful information for characterizing atmospheric OC in aerosols and rainwater (e.g., Fu et al., 2015; Miyazaki et al., 2018; Yang et al., 2019).

15        As described in section 3.7, our results strongly suggested that the submicron WSOC observed in the Amundsen Sea might be formed by the condensation of BVOCs onto preexisting submicron sea spray aerosols by showing the strong correlations with Na$^+$ and MSA. To further elucidate the sources of WSOC, we investigated the fluorescence properties of WSOC in the fine mode using EEM-PARAFAC. Fluorophores in WSOC were divided into three primary types on the basis of their peak position (Fig. 9). The spectral features of component 1 (C1, Ex/Em: 300/344 nm) was similar to the component

previously identified in in coastal and oceanic waters as well as the polar ocean and was thought to be phytoplankton-derived (or ice algae-derived) protein-like component (Stedmon et al., 2007, 2011, Retelletti Brogi et al., 2018). Component 2 (C2, Ex/Em: 276/326 nm) was assigned as a tryptophan-like fluorophore, which has been considered to be a labile component produced as a result of biological production in marine environments (Coble et al., 1998; Yamashita and Tanoue 2003). Component 3 (C3, Ex/Em: <260/458 nm) spectra were characterized as representing terrestrial humic-like fluorophores (Coble

et al., 1996; Yamashita and Tanoue, 2003; Chen et al., 2010; Chen et al., 2018). During the cruise, the C1 fluorescence intensity was much higher and variable, varying between 0.0133 and 0.139 RU (Fig. 10a). In comparison, the fluorescence intensity of C2 (range: 0.0195–0.0518 RU) and C3 (range: 0.00857–0.0351 RU) was much less variable. Among the three components, the protein-like C1 was the dominant fluorescence component in our marine aerosol samples, accounting for 31–73% of the total intensity, and the relative contributions of tryptophan-like C2 and terrestrial humic-like C3 were found to represent 17–

50% and 8–31%, respectively (Fig. 10b). In our marine aerosol samples, protein-like components (i.e., C1 and C2) represented 69–91% of the total intensity.

        Despite the extremely high Chl-a concentration in the Amundsen Sea (Fig. 3), we found no significant difference of the average values of protein-like C1 and tryptophan-like C2 fluorescence intensity between the Amundsen Sea and the Southern



Ocean. However, relatively much higher values of C1 and C2 fluorescence intensity were observed when the ship approached the Amundsen Sea (i.e., samples A3 and A4), passing through the sea ice zone (Figs. 1b and 10a). The C1 and C2 fluorescence intensity values sharply decreased and remained relatively low in the Amundsen Sea, and then gradually increased from the Amundsen Sea to the Southern Ocean. As mentioned in section 3.2, *P. antarctica* was the dominant phytoplankton species in the Amundsen Sea, whereas diatoms formed a major group in the marginal sea ice zone (Lee et al., 2016). Interestingly, fluorescence intensity of C1 showed a significant positive relationship with the relative biomass of diatoms ($r = 0.89$, $p < 0.01$); however, it was negatively correlated with the relative biomass of *P. antarctica* ($r = 0.79$, $p < 0.05$) (Figs. 10c and 10d). Similar results were found between fluorescence intensity of C2 and the relative biomass of diatoms and *P. antarctica*. Phytoplankton can emit several types of BVOCs, such as isoprene, monoterpenes, and amines that have the potential to form SOA (Meskhidze et al., 2011). Sabolis (2010) reported that diatoms were the largest emitters of most of the observed BVOC (including isoprene, monoterpenes, chloroform, and iodomethane) and that the other phytoplankton species (e.g., dinoflagellates, haptophytes, and Prochlorococcus) showed variable production rates for different BVOC, showing a strong dependence on phytoplankton speciation for BVOC production. Consequently, our results suggest that protein-like components are most likely produced as a result of biological processes of diatoms, which play a key role in forming the submicron WSOC observed over the Southern Ocean and the Amundsen Sea, and that phytoplankton community structure is a significant factor affecting atmospheric OC species since the submicron WIOC was quite related to the relative biomass of *P. antarctica* (see section 3.6).

The biological index (BIX) has been used to estimate the contribution of autochthonous biological activity (Fu et al., 2015; Miyazaki et al., 2018). An increase in BIX is related to an increase in the contribution of microbially derived organics. High BIX values ($> 1$) have been shown to correspond to a predominantly biological or microbial origin of dissolved organic matter and to the presence of organic matter freshly released into water, whereas low values ($< 0.6$) indicate little biological material (Huguet et al., 2009). In this study, the BIX values ranged from 1.17–3.61, with an average of $2.23 \pm 0.807$ (Fig. 10b). The high BIX values also supported that the majority of WSOC was derived from biological processes.

## 4 Conclusions

Characteristics of biogenically-derived atmospheric sulfur (i.e., MSA and nss-SO$_4^{2-}$) and OC (i.e., WSOC and WIOC) species in marine aerosols, and the environmental factors influencing their distributions were investigated over the Southern Ocean and the Amundsen Sea during the austral summer. In the Amundsen Sea, atmospheric MSA concentration drastically increased (up to 0.57 µg m$^{-3}$), suggesting significant influences of marine biological activities on atmospheric sulfur species. Furthermore, biogenically-derived nss-SO$_4^{2-}$ dominated the atmospheric budget of nss-SO$_4^{2-}$ in the Amundsen Sea, contributing ~86% to total nss-SO$_4^{2-}$. These results were attributed to exceptionally high seasonal primary production during the austral summer, the dominance of *P. antarctica* with respect to phytoplankton biomass, and extremely high DMS concentrations produced by DMS-producing algae species (e.g., *P. antarctica*) in the Amundsen Sea.

WIOC was the dominant OC species in both the Southern Ocean and the Amundsen Sea, accounting for 75% and 73%, respectively. Despite extremely high Chl-a concentration in the Amundsen Sea, no significant differences of mean WSOC and



WIOC concentrations were found between the Southern Ocean and the Amundsen Sea. However, WSOC and WIOC were highly enriched in the submicron sea spray particles, especially in the Amundsen Sea where biological productivity was much higher than the Southern Ocean. We found significant inverse relationships between WSOC/Na$^+$, WIOC/Na$^+$ ratios and the mean wind speed, suggesting that the SML coverage affected the organic mass fractions of sea spray aerosols in our study region. In addition, these ratios showed strong correlations with DOC concentration in the Amundsen Sea when the wave breaking thoroughly mixes the SML with the underlying water. These results revealed that the organic mass fraction of sea spray aerosol could be uncoupled with Chl-a and phytoplankton biomass, but related to biological activities under certain meteorological conditions.

It is worth noting that the simultaneous measurements of chemical species in marine aerosols as well as chemical and biological properties of seawater in the Amundsen Sea allowed a better understanding of the effect of ocean ecosystem on biogenically-derived OC species. A good correlation was found between the relative biomass of *P. antarctica* and the submicron WIOC concentration, suggesting that extracellular polysaccharide mucus generated by *P. antarctica* is a significant source of atmospheric WIOC in the Amundsen Sea. Moreover, the fluorescence properties of WSOC revealed that the majority of WSOC (i.e., protein-like components) was most likely derived from BVOCs as a result of biological processes of diatoms, by showing significant positive relationships between the relative biomass of diatoms and protein-like components in marine aerosols in the Amundsen Sea.

Ice shelves and glaciers in the Amundsen Sea have been shrinking at a remarkable rate (Rignot et al., 2008). Moreover, sea ice coverage is decreasing fast in the western Antarctic (Stammerjohn et al., 2012). Because ocean buoyancy, stratification, and trace metal distribution are affected by these changes, the regional oceanography, phytoplankton community structure and biogeochemical cycles of sulfur and carbon in the Amundsen Sea are likely affected as well (Yager et al., 2012). Further studies, therefore, are required to understand more clearly biogeochemical cycles of sulfur and carbon between the ocean and the marine atmosphere and should focus on long-term monitoring of atmospheric sulfur and OC species in the Amundsen Sea.

## 5 Data availability

The data used in this study is available on request to the correspondence author Jinyoung Jung (jinyoungjung@kopri.re.kr).

## Author contributions

JJ designed the research, carried out the experiments, processed the data, and wrote the paper. SBH, MC and LJ analyzed the aerosol samples. YL and EJY provided the marine biological data. JOC and JP helped in obtaining satellite products. JH, KP, DH and EJY contributed the scientific discussion and paper correction. TWK and SHL organized the field campaign.



**Acknowledgements**

We are grateful to the captain and crews of IBR/V *Araon* for their enthusiastic assistance during the ANA06B cruise.
This study was supported by grants from Korea Polar Research Institute (KOPRI) (PE19060).

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





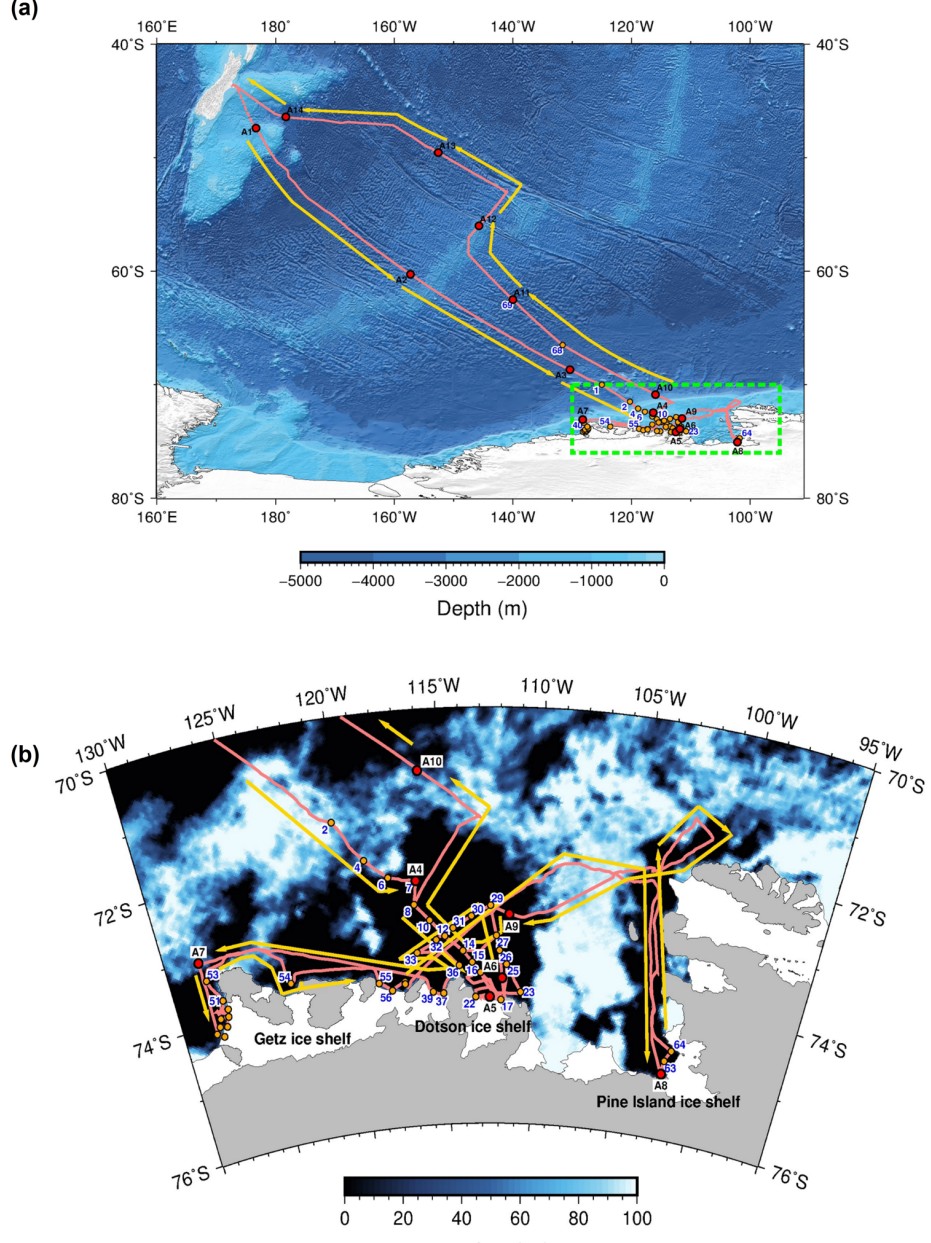

**Figure 1: Cruise track of the ANA06B. Panel (a) shows the entire cruise track (pink line) with bathymetry, aerosol (red circles) and seawater sampling (orange circles) locations. Each aerosol sampling start point represents the end of the previous sampling period. Dotted pink line (from Christchurch, New Zealand to A1 sampling location) indicates that no aerosol sampling was conducted. Yellow arrows and green dotted rectangle indicate the moving direction of the ship and the location of the Amundsen Sea, respectively. Panel (b) shows the cruise tack in the Amundsen Sea with sea ice concentration on 6 January 2016. Sea ice concentration was obtained from the Advanced Microwave Scanning Radiometer (AMSR) 2 sea ice maps (Spreen et al., 2008).**



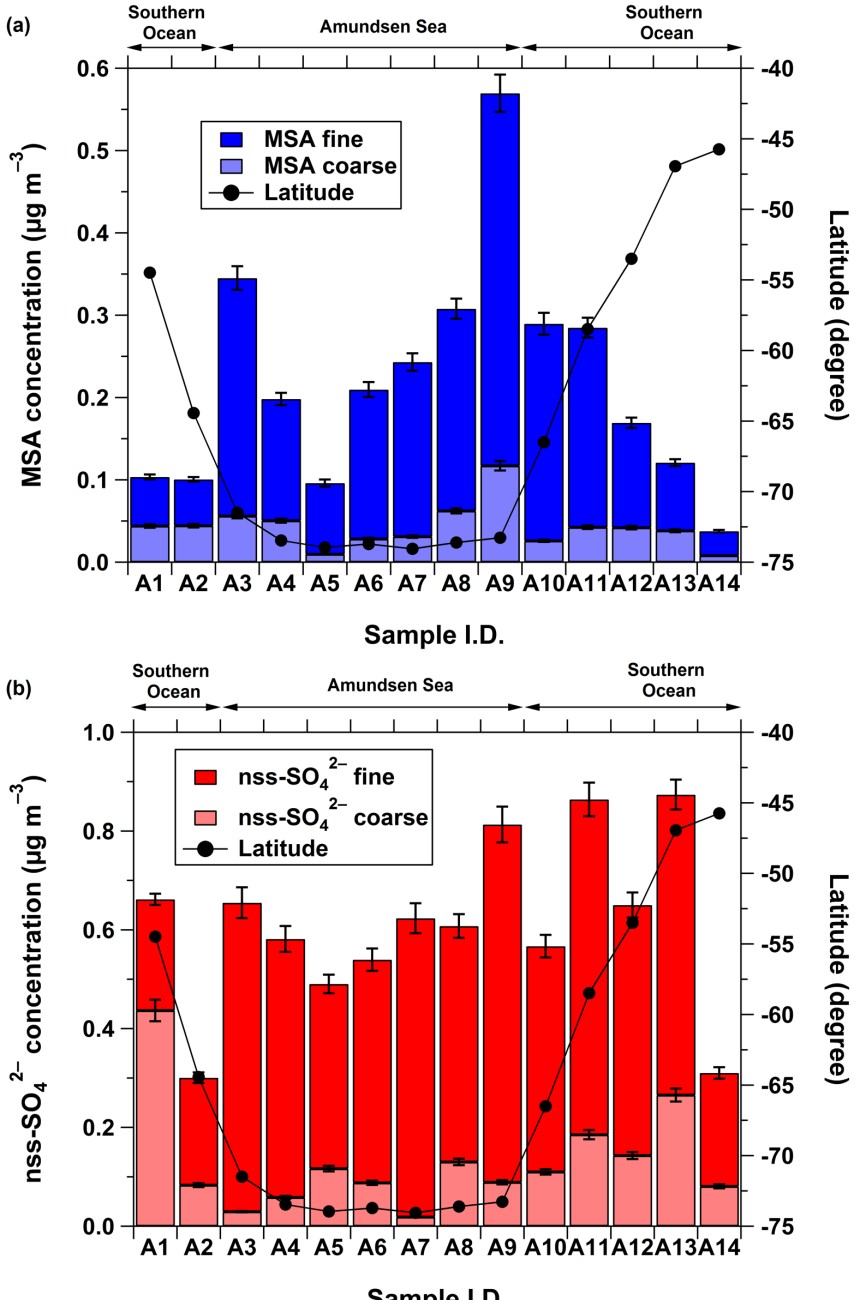

**Figure 2. Concentrations of methanesulfonic acid (MSA) (a) and non-sea-salt sulfate (nss-SO$_4^{2-}$) (b) against sample I.D. in aerosols collected over the Southern Ocean (43°S−70°S) and the Amundsen Sea (70°S−75°S). Black solid circle lines in panels (a) and (b) indicate the latitude of half-way point between each aerosol sampling start and endpoint. Double arrows in upper panels show the sampling locations of aerosol samples.**





**Figure 3. Monthly composite chlorophyll-a concentrations in the Southern Ocean (a) and the Amundsen Sea (b) for January 2016. The cruise track (pink line) and aerosol sampling locations (red circles) are also shown. Each aerosol sampling start point represents the end of the previous sampling period. The green dotted rectangle in panel (a) indicates the location of the Amundsen Sea. Sea ice distribution on 6 January 2016 is shown in panel (b).**

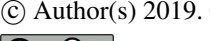



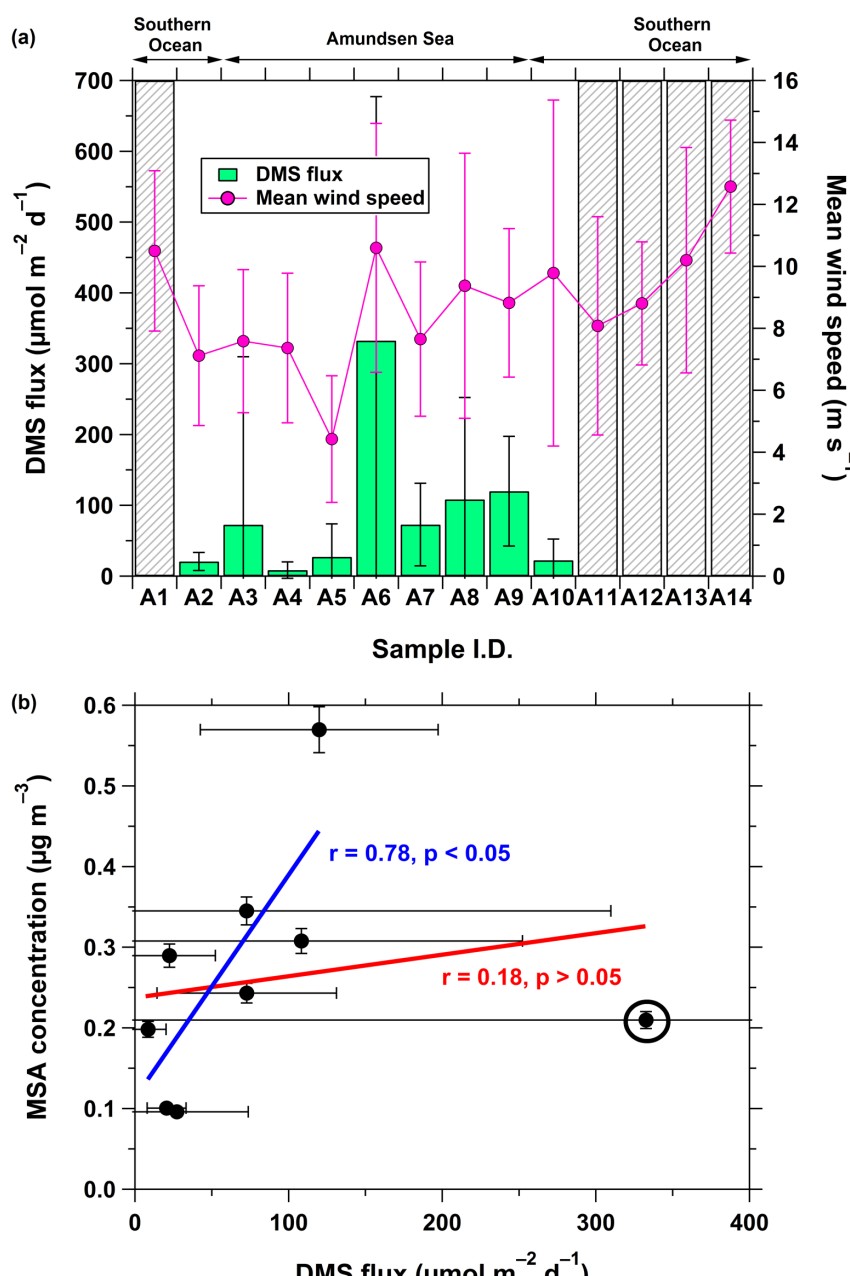

**Figure 4.** Dimethylsulfide (DMS) flux (data from Kim et al., 2017) (a) against aerosol sample I.D. and the correlation between MSA concentration and DMS flux (b). Each DMS flux in both panels (a) and (b) represents its mean value for each aerosol sampling time. The gray hatched areas in panel (a) denote that no DMS measurement was conducted. Pink solid circle line in panel (a) indicates the mean and standard deviation of wind speeds for each aerosol sampling time. Double arrows in panel (a) show the sampling locations of aerosol samples. In panel (b), the red line indicates the correlations between MSA concentration and DMS flux when all data sets were used, and the blue line shows their correlation when the outlier (circled point; i.e., sample A6) is excluded.







**Figure 5.** MSA/nss-SO$_4^{2-}$ ratio (a), concentrations of biogenic nss-SO$_4^{2-}$ (b), anthropogenic nss-SO$_4^{2-}$ (c) and nitrate (NO$_3^-$) (d) against aerosol sample I.D. Double arrows in upper panels show the sampling locations of aerosol samples. Black solid circle lines in panels (a) and (d) indicate the latitude of half-way point between each aerosol sampling start and endpoint. The contributions (black solid triangle lines) of biogenic nss-SO$_4^{2-}$ and anthropogenic nss-SO$_4^{2-}$ to total nss-SO$_4^{2-}$ are shown in panels (b) and (c), respectively. The gray hatched areas in panel (c) denote that anthropogenic nss-SO$_4^{2-}$ was not estimated (see text).



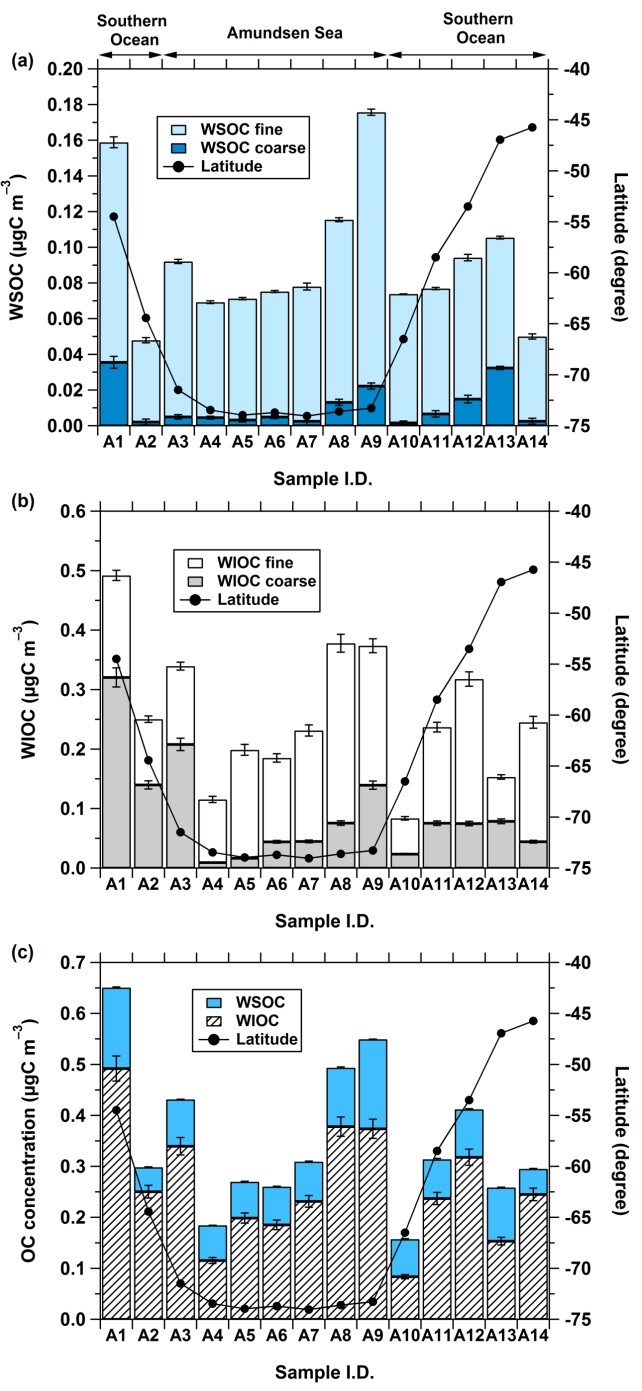

**Figure 6.** Concentrations of water-soluble organic carbon (WSOC) (a), water-insoluble organic carbon (WIOC) (b), and organic carbon (i.e., OC = WSOC + WIOC) (c) against sample I.D. in aerosols collected over the Southern Ocean (43°S−70°S) and the Amundsen Sea (70°S−75°S). Black solid circle lines indicate the latitude of half-way point between each aerosol sampling start and endpoint. Double arrows in panel (a) show the sampling locations of aerosol samples.





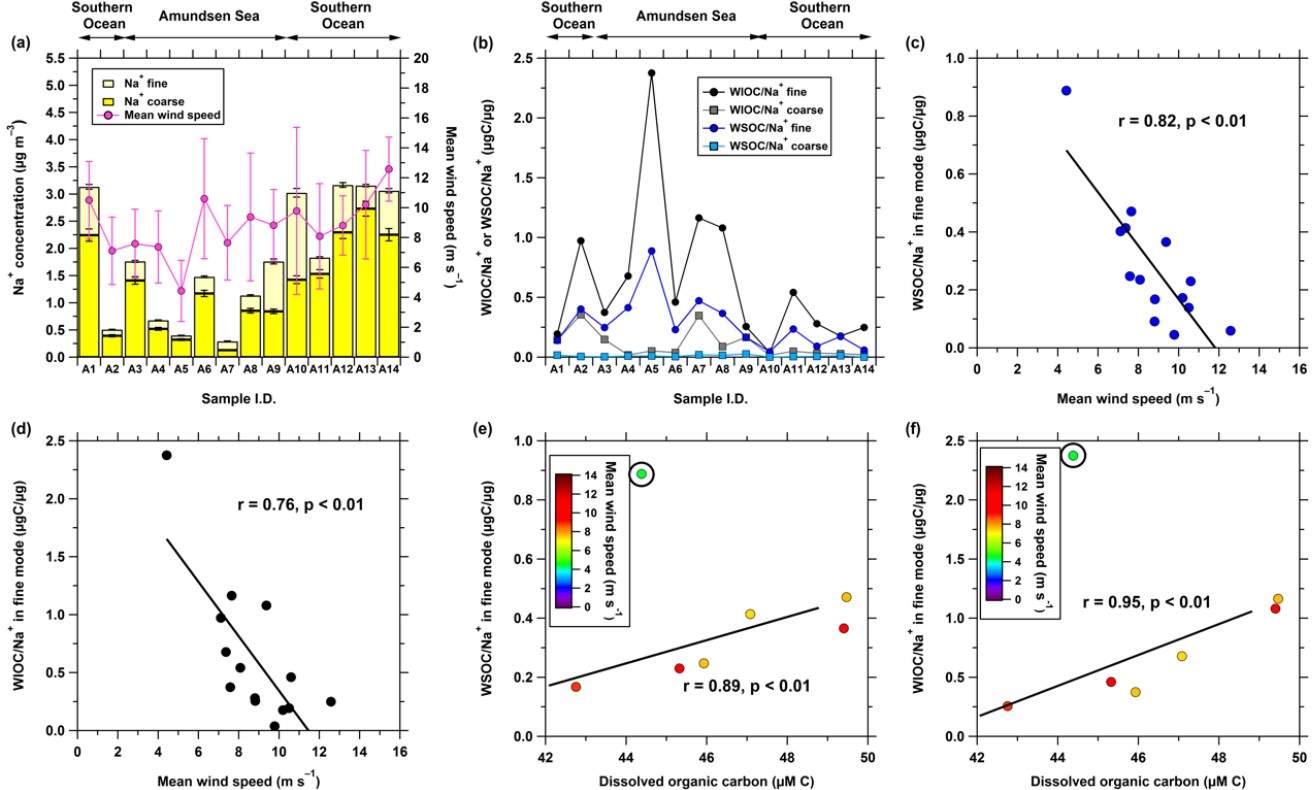

**Figure 7. Concentration of Na$^+$ (a), WIOC/Na$^+$ and WSOC/Na$^+$ ratios in both fine and coarse modes (b) against aerosol sample I.D. Pink solid circle line in panel (a) indicates the mean and standard deviation of wind speeds for each aerosol sampling time. The relationships between WSOC/Na$^+$, WIOC/Na$^+$ in the fine modes and mean wind speed are shown in panels (c) and (d). Panels (e) and (f) show the relationships of WSOC/Na$^+$ and WIOC/Na$^+$ in the fine modes with sea surface DOC in the Amundsen Sea, respectively, and the mean wind speeds for each aerosol sampling time are illustrated in the color bar. Note that sea surface DOC was measured only in the Amundsen Sea. The correlations in both panels (e) and (f) were obtained when the outlier (circled point; i.e., sample A5) is excluded (see text).**





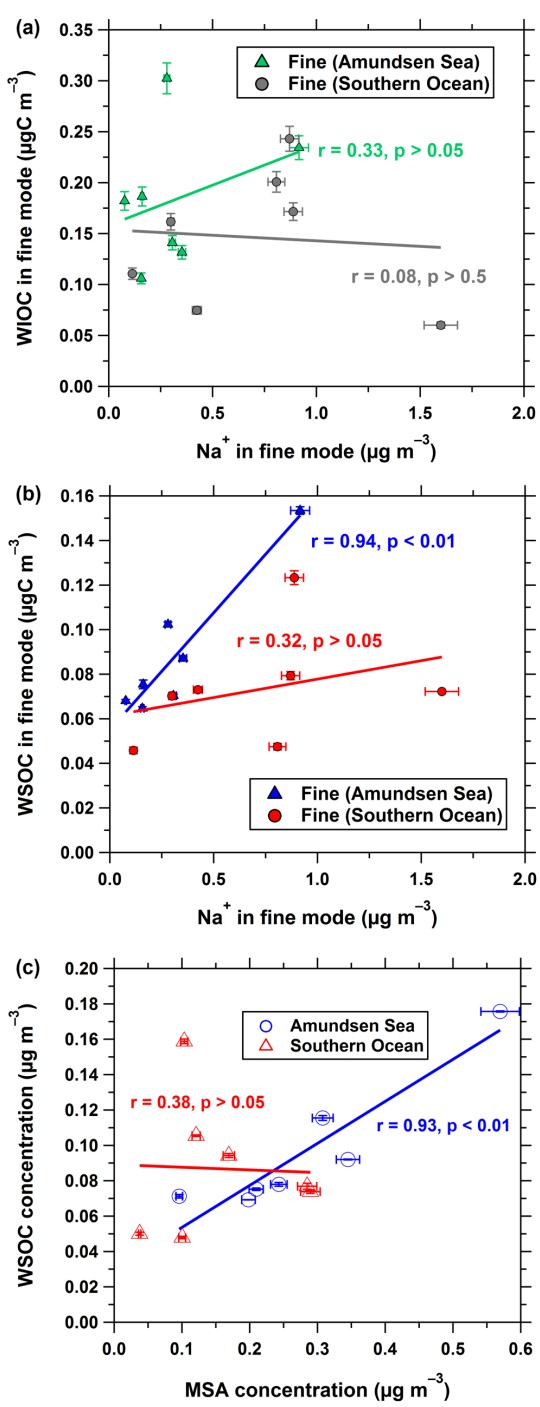

**Figure 8. Relationships of WIOC (a) and WSOC (b) with Na$^+$ in fine modes over the Southern Ocean (solid triangles) and the Amundsen Sea (solid circles). Panel (c) shows relationships between WSOC and MSA concentrations over the Southern Ocean (open triangles) and the Amundsen Sea (open circles).**



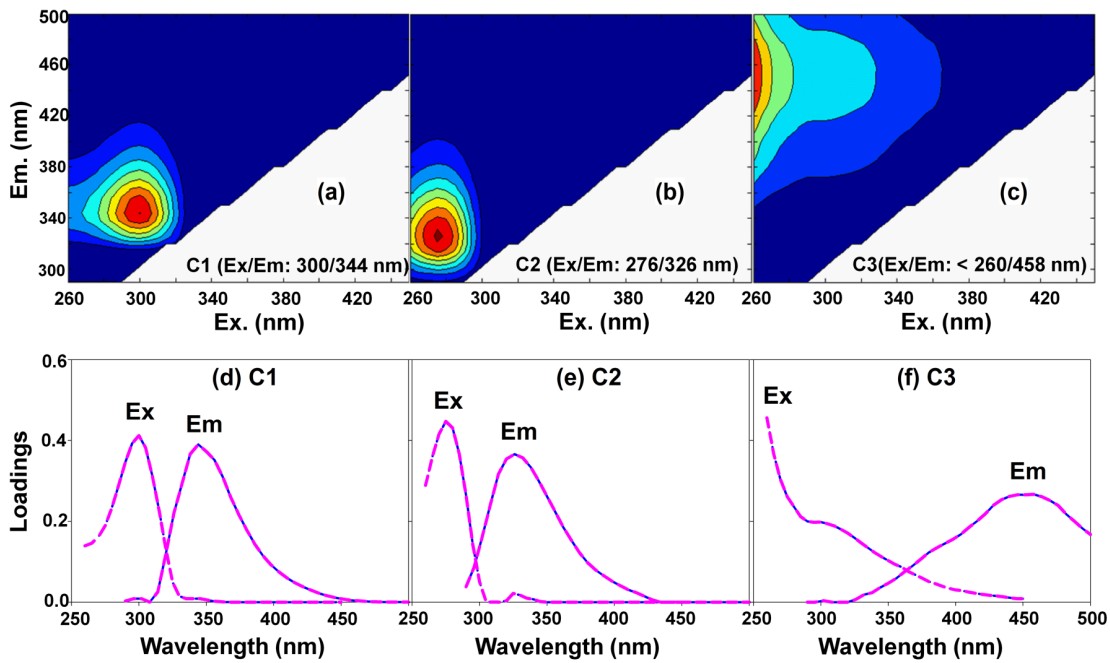

Figure 9. EEM contour plots of three fluorescent components C1−C3 (a−c) identified using EEM-PARAFAC in the fine mode aerosol particles (D < 2.5 μm) collected over the Southern Ocean and the Amundsen Sea, and excitation and emission spectra of C1−C3 (d−f).





**Figure 10. Variations of fluorescence intensities (a), relative contributions of C1−C3 and biological index (black solid square line) (b) in the fine mode aerosol particles (D < 2.5 μm) collected over the Southern Ocean and the Amundsen Sea. Black solid circle line in panel (a) indicates the latitude of half-way point between each aerosol sampling start and endpoint. Double arrows in panel (a) show the sampling locations of aerosol samples. Panels (c) and (d) show the relationships of fluorescence intensity of C1 and C2 with the relative biomass of diatoms and *Phaeocystis antarctica* (*P. antarctica*) observed in the Amundsen Sea, respectively.**