# Peer review of "Characteristics of biogenically-derived aerosols over the Amundsen Sea, Antarctica"

_Atmospheric Chemistry and Physics, 2019_

## Referee Comment (RC1) · Anonymous Referee #4 · 22 Mar 2019

Aerosol and seawater samples were collected and analyzed during a cruise over the Southern Ocean and Amundsen Sea polynyas in this article. Aerosol compositions, specifically MSA, nss-sulfate, WSOC, WIOC and sea salt (represented by Na+), are reported and their spatial and temporal behaviors are investigated with environmental factors influencing their distributions. WSOC was also analyzed by fluorescence excitation-emission matrices to elucidate its sources.

This work presents a unique data set of atmospheric and seawater measurement in west Antarctica region and shows some interesting results, such as the domination of biogenically-derived nss-sulfate in the Amundsen Sea, strong correlations between OC enrichment on sea salt particles and DOC concentration in the Amundsen Sea, and different biological processes that could account for WSOC and WIOC formation.

[Figure]

I will support the publication of this manuscript if the authors can properly address the following comments.

Major Comments:

In section 3.6, the authors attributed the negative correlation of the wind speed and WSOC/Na+ and WIOC/Na+ to sea surface micro-layer coverage, suggesting that lower wind speed led to higher sea surface micro-layer coverage and higher OC enrichment on sea salt particles. I am not sure if this is clearly supported by the measurements. In order to come to this conclusion, WSOC and WIOC could only be exclusively ocean-generated and internally mixed with sea salt particles. This is not directly supported by the measurements since the particle mixing state was not presented in this study. The correlation of DOC concentration in sea surface micro-layer to wind speed (Figure 7e and Figure 7f) is too weak to account for the strong negative correlation of the wind speed and WSOC/Na+ and WIOC/Na+ (Figure 7c and Figure 7d). The most intuitive reason of this wind dependence of OC to sea salt ratio is that wind speed is higher in the Southern Ocean and thus more sea salt particles were emitted from the ocean whereas OC is not as strongly affected by wind as sea salt. According to section 3.7, WSOC and WIOC are from secondary and primary sources, respectively, yet their ratio to sodium showed a somewhat similar trend against wind speed in Figure 7c and Figure 7d. This probably means that sodium, instead of OC, was more sensitive to wind speed changes.

Minor Comments:

1. P2 Line 11- P3 Line 5: These two paragraphs are about marine and biological CCN. Not sure such a detailed introduction is needed since CCN is mentioned in only section 1 and not discussed in later sections.

2. P9 Line10-16: Since there are only 14 samples (or in this case 9 samples), removal of outliers needs better discussion. Please specify how "the highest mean wind speed" affects DMS flux and possibly provide either reference or calculation.

3. P9 Line 28: "the variation trend of " Can the authors provide correlation coefficient?

4. P11 Line 12: "somewhat overestimated... showing negative values" Can the authors discuss possiblities causing this overestimation?

5. P11 Line 14: "variation trend" Again consider show correlation coefficient.

6. P 17 Line 7: "negatively correlated with the relative biomass of P. antarctica (r =0.79, p < 0.05)" Should be "-0.79" if using the Pearson correlation coefficient. Otherwise, the authors should specify the sign. This also applies to some other place where r is less than 0 including some of the figures.

7. P18 Line 18: Consider citing this article (Bromwich et al., 2013).

Editorial Comments:

1. P4 Line 20: "Procedural blanks (n = 4)..." Change to "Four procedural blanks..."

2. P13 Line 15:" reflecting that Na+ was formed from bubble by local wind speed. " Delete "speed".

3. P16 Line 15-16:" our results strongly suggested that the submicron WSOC observed in the Amundsen Sea might be formed..." Remove "might"

4. Figure 8c: Check unit

References

Bromwich, D. H., Nicolas, J. P., Monaghan, A. J., Lazzara, M. A., Keller, L. M., Weidner, G. A., and Wilson, A. B.: Central West Antarctica among the most rapidly warming regions on Earth, Nature Geoscience, 6, 139-145, 10.1038/ngeo1671, 2013.

---

## Referee Comment (RC2) · Anonymous Referee #3 · 16 Apr 2019

The manuscript presents shipborne measurements of specific atmospheric aerosol, mainly methane sulfonic acid (MSA), non-sea-salt sulfate, OC/EC, and water-insoluble and soluble organic carbon (WIOC, WSOC) over the Southern Ocean and the Amundsen Sea. Measurements also include atmospheric and oceanic factors like wind speed, chlorophyll, dissolved organic carbon, particulate organic carbon, dimethyl sulfide (DMS), and taxonomy of phytoplankton. The authors explore the sources and variability of MSA, nss-sulfate, WSOC, and WIOC and attempt to link them to environmental and biological factors. For example, the authors do an adequate job of linking MSA to DMS, although their statistics are low; removing one measurement point changes their correlation. Nevertheless, these are important measurements that will eventually need to be published. The manuscript is well written and could benefit from shortening. I

recommend publication if the following comments are properly addressed.

Major comments: 1. The first major comment reiterates the comment already raised by anonymous reviewer #4. The authors suggest that the inverse relationship between WSOC/Na+ and WIOC/Na+ with wind speed implies that organics in the atmosphere are controlled by winds due to the breakage of the surface microlayer for periods of times associated with high winds. However, I believe that there are several inconsistencies in the text that will need to be clearly addressed prior to publication, mainly:

a. Both WSOC/Na+ and WIOC/Na+ are inversely related to wind speeds but the authors claim that only WIOC have a primary source whereas WSOC is mostly formed by oxidation of biogenic precursors. How do the authors then explain that both ratios correlated well with wind speed?

b. The authors hypothesize that WIOC is of primary origin, but their measurements indicate no correlation between WIOC and Na+. The authors attribute this to transport but their measurements clearly indicate a correlation between Na+ and wind speed, suggesting transport cannot completely rule out a correlation between WIOC and Na+. Also, why does WIOC correlate with the relative abundance of P. Antarctica? Presumably, this implies that a part of WIOC is formed from oxidation of BVOC?

c. Finally, why do WSOC concentrations (presumably of secondary origin, i.e., from the oxidation of BVOC) correlate strongly with Na+ and with DOC? The authors hypothesize that the WSOC relation on Na+ is due to higher surface area (from salt particles) and therefore higher WSOC concentrations are a result of a larger condensation sink. However, the authors present no compelling evidence to substantiate that hypothesis. Can the author examine the relation between WSOC and average short wavelength radiation? This could provide the evidence needed to argue that WSOC is formed from secondary sources. The linkage between WSOC and WIOC with biology needs to be better articulated.

2. What happens to the correlations in Figures 7 if the authors looked at WSOC and

[Figure]

WIOC instead of WSOC/Na+ and WIOC/Na+? This could remove the cross-correlation with wind speed (I encourage the authors to add these graphs as part of the SI).

3. The authors often mention correlation strength even though the p-value for the regression is higher than the specified threshold (in this case 0.05). A p-value larger than the threshold implies there is no confidence in rejecting the null hypothesis (which is: the variables are not correlated). I suggest the authors reframe their results accordingly.

Minor comments: P7 L29: specify at least once what +/- refers to, one standard error or the one standard deviation?

P9 L9-L10: The authors describe a weak but not significant relation between MSA and DMS. If the p-value is not significant than the authors cannot justify that a positive or negative correlation (see major comment 2). Please adjust elsewhere in the text where applicable.

P9 L13: The authors should provide more thorough reasoning for removing the point to the right. Indeed, removing that one point changes the correlation between DMS and MSA from non-existent to significant, due to the small sample number.

P9 L27-30: "Unlike MSA, the mean nss-SO42- concentration in the Amundsen Sea..." please provide justification to how the trend for nss-sulfate was comparable to that of MSA. Visual inspection of the data is not sufficient. Also, I am not sure how the authors arrived at the conclusion that nss-sulfate is influenced by marine and anthropogenic sources given the information in that sentence.

P10 L17: "... showing a similar variation trend to that of MSA". Quantify the agreement.

P11 L14-19: "We expected much higher WSOC and WIOC concentrations in the Amundsen Sea than the Southern Ocean because of extremely high Chl-a concentrations...". This result could be consistent with findings of Quinn et al. (2014) who argued that organic enrichment in the aerosol phase is likely controlled by the

dissolved organic carbon (DOC) pool rather than chlorophyll concentrations.

P10 L20-21: Specify how the enrichment of WSOC and WIOC in the aerosol phase was calculated.

P10 L23: "For example, O'Dowd et al. (2004) observed that the contribution of OC fraction to the submicrometer aerosol mass increased from 15% to 63% between low and high biological activity periods in the North Atlantic." I am confused by this sentence. The measurements presented in this article show no relation of WSOC and WIOC on chl-a concentrations (as explicitly written in the previous paragraph). Please clarify.

P11 L31: "the dominance of WIOC suggests that the bubble bursting process by local wind speeds is a significant formation mechanism of atmospheric WIOC in our area." If this is true, then why don't the author observe lower WIOC concentration over the Amundsen Sea compared to the Southern Ocean given that wind speed in the Southern Ocean were larger compared to over the Amundsen Sea?

---

## Referee Comment (RC3) · Anonymous Referee #1 · 16 Apr 2019

The authors present measurements of aerosol and ocean carbon from the southern ocean and Amundsen Sea linking biological processes to marine particle concentrations and composition. This is an important processes, though not well understood, partly due to the interdisciplinary aspect of the subject. Better understanding this process possibly has significant implication on understanding future climate change. The authors do well to discuss and provide evidence for their claims. I am suggesting the article be accepted after major revisions. First, I agree with comments from referees 3 and 4 and will not repeat them here. The other main concerns I have involve the claims made in section 3.3-3.4, specifically that a significant amount of nssSO4 is from anthropogenic sources. While possible, I do not think the authors have provided strong evidence. Previous studies in the SO have shown that almost no continen-

tal/anthropogenic pollution is transported over the southern ocean particularly south of ~50S (See Hudson et al. 1998). While the Hudson et al. 1998 measurements are in a different part of the Southern Ocean, I expect the result to be similar. A back trajectory analysis would strengthen your case if there really is anthropogenic sources.

Major Comments: You noted the weakness of using MSA/nssSO4 to identify the contribution of biological nssSO4 to the total nssSO4, and yet you still rely heavily on this method. The nssSO4 likely correlates strongly to MSA in the Amundsen Sea simply because the nssSO4 source (MSA) is local. It is common for nssSO4 to be transported large distance, often in the free troposphere, resulting in little to no correlation with local sources/tracers (Sanchez et al. 2018) as you have seen in the Southern ocean with MSA (section 3.3). The lack of MSA alone is not a strong argument for the presence of anthropogenic nssSO4. Based on the fact that there are no anthropocentric sources remotely close to most of these locations, this is the perfect example on why this method of identifying biological/anthropogenic nssSO4 contributions is flawed.

Also, while nitrate can be attributed to anthropogenic emission, particle nitrate concentration (and nssSO4 concentration) can also be enhanced by aqueous phase processing (in clouds). On Figure 5d, with the exception of samples A1 and A13 (both of which are between 45S and 50S on figure 1a, and relatively close to anthropogenic sources) the nitrate concentration seems fairly consistent and likely have little to no anthropogenic influence.

I would be surprised to find that at times anthropogenic sources account for even close to half the nssSO4 in the Amundsen Sea, but you state biological sources account for 39%-138% nssSO4 (i.e less than half at times). There are no relatively nearby anthropogenic sources to account for this. More evidence would be needed to make such a claim. With the exception of the northern most stations, I find it hard to believe anthropogenic sources would be a major source of nssSO4.

Hudson, J.G., Y. Xie, and S.S. Yum, Vertical distributions of cloud condensation nuclei

spectra over the summertime Southern Ocean, J. Geophys. Res., 103, 16,609-16,624, 1998

---

## Author Comment (AC1) · 9 Jul 2019

Please find our responses to Referee #4 in the supplement.

Please also note the supplement to this comment:
https://www.atmos-chem-phys-discuss.net/acp-2019-133/acp-2019-133-AC1-supplement.zip

---

## Author Comment (AC2) · 9 Jul 2019

Please find our responses to Referee #3 in the supplement.

Please also note the supplement to this comment:
https://www.atmos-chem-phys-discuss.net/acp-2019-133/acp-2019-133-AC2-supplement.zip

———————————————

---

## Author Comment (AC3) · 9 Jul 2019

We already have revised our manuscript according to Referee #1's comments and uploaded the revised manuscript file on 19 March 2019.

---

## Author Response (AR1)

**Responses to Reviewers' comments**

09/07/2019

Journal: Atmospheric Chemistry and Physics

Title: Characteristics of biogenically-derived aerosols over the Amundsen Sea, Antarctica

Authors: Jinyoung Jung, Sang-Bum Hong, Meilian Chen, Jin Hur, Liping Jiao, Youngju Lee, Keyhong Park, Doshik Hahm, Jung-Ok Choi, Eun Jin Yang, Jisoo Park, Tae-Wan Kim, and SangHoon Lee

Manuscript number: acp-2019-133

Manuscript type: Research Article

Note: Reviewers' comments are high lightened in **blue**, while our responses to reviewers are in **black**. The revisions in the manuscript was highlighted to make easily readable for the editors and reviewers.

Dr. Ulrich Pöschl

Chief Executive Editor

Atmospheric Chemistry and Physics

Max Planck Institute for Chemistry,

Mainz, Germany

Dear Doctor Pöschl,

We have attached an electronic copy of manuscript file ready to go to press entitled "Characteristics of biogenically-derived aerosols over the Amundsen Sea, Antarctica" by Jinyoung Jung, Sang-Bum Hong, Meilian Chen, Jin Hur, Liping Jiao, Youngju Lee, Keyhong Park, Doshik Hahm, Jung-Ok Choi, Eun Jin Yang, Jisoo Park, Tae-Wan Kim, and SangHoon Lee for publication in Atmospheric Chemistry and Physics (acp-2019-133). We have modified our text based on the referee's comments. We appreciated that the comments from referees improved our manuscript a lot. We believe that the comments from referees were clearly responded in our modified manuscript.

We are looking forward to hearing about your decision. Thank you for your consideration.

Sincerely yours,

Jinyoung Jung

**Anonymous Referee #4:**

**Major Points:**

In section 3.6, the authors attributed the negative correlation of the wind speed and WSOC/Na+ and WIOC/Na+ to sea surface micro-layer coverage, suggesting that lower wind speed led to higher sea surface micro-layer coverage and higher OC enrichment on sea salt particles. I am not sure if this is clearly supported by the measurements. In order to come to this conclusion, WSOC and WIOC could only be exclusively ocean-generated and internally mixed with sea salt particles. This is not directly supported by the measurements since the particle mixing state was not presented in this study. The correlation of DOC concentration in sea surface micro-layer to wind speed (Figure 7e and Figure 7f) is too weak to account for the strong negative correlation of the wind speed and WSOC/Na+ and WIOC/Na+ (Figure 7c and Figure 7d). The most intuitive reason of this wind dependence of OC to sea salt ratio is that wind speed is higher in the Southern Ocean and thus more sea salt particles were emitted from the ocean whereas OC is not as strongly affected by wind as sea salt. According to section 3.7, WSOC and WIOC are from secondary and primary sources, respectively, yet their ratio to sodium showed a somewhat similar trend against wind speed in Figure 7c and Figure 7d. This probably means that sodium, instead of OC, was more sensitive to wind speed changes.

(Response) We agree to Referee #4's comments. We realized that our discussion about the negative correlations of the wind speed with WSOC/Na$^+$ and WIOC/Na$^+$ was not supported by the measurements. To explain these relationships more clearly, we have modified Figures 7c and 7d by adding in situ Chl-a concentrations averaged for each aerosol sampling time (page 35). Although in situ Chl-a was not measured during the cruise from New Zealand to the entrance to the Amundsen Sea due to the limited ship time, our results, shown in Figures 7c and 7d, suggest that the wind speed is a significant factor influencing the WSOC/Na$^+$ and WIOC/Na$^+$ ratios in our study region. According to Referee #4's comments, we revised our manuscript as follows: "In addition to marine biological activities, the wind speed has been reported to have an effect on the organic mass fraction of sea spray aerosol (Rinaldi et al., 2013; Ceburnis et al., 2016). Gantt et al. (2011) demonstrated that the organic mass fraction of sea spray aerosol depends not only on marine biological activity in oceanic surface waters but also on sea surface wind speed. During the sampling period, both WSOC/Na$^+$ and WIOC/Na$^+$ ratios showed inverse relationships with mean wind speed (r = –0.82, p < 0.01 and r = –0.76, p < 0.01, respectively) (Figs. 7c and 7d). The highest WSOC/Na$^+$ and WIOC/Na$^+$ ratios were observed in the aerosol sample (i.e., A5) collected in biologically active region (i.e., in situ Chl-a concentration was highest (6.9 mg m$^{-3}$)) under calm conditions (i.e., mean wind speed was lowest (4.4 m s$^{-1}$)), and the WSOC/Na$^+$ and WIOC/Na$^+$ ratios drastically decreased with increasing mean wind speed. This result suggests that organic matters produced in high biological activity conditions were enriched in the sea surface water under low wind speed conditions, making a favorable condition for transferring organic matters to the atmosphere through the bubble bursting process (Rinaldi et al., 2013). However, as shown in Figures 7c and 7d, the WSOC/Na$^+$ and WIOC/Na$^+$ ratios observed in the biologically active region (i.e., higher in situ Chl-a concentrations) did not always show higher values when the wind

speeds exceeded 7 m s$^{-1}$. According to previous studies (Gantt et al., 2011; Rinaldi et al., 2013), the wave breaking caused by high wind speeds (> 8 m s$^{-1}$) thoroughly mixes the organic-enriched sea surface layer with the underlying relatively organic-poor waters resulting in the homogeneous water column and considerably reduces the organic mass fraction of sea spray aerosol. Therefore, the inverse relationships between WSOC/Na$^+$, WIOC/Na$^+$ ratios and the mean wind speed suggest that the wind speed is a significant factor influencing the organic mass fractions of sea spray aerosols in our study region." (page 14, line number 3–19). Besides, we added the references (i.e., Rinaldi et al., 2013; Ceburnis et al., 2016) to the manuscript (page 26, line number 4–6; page 20, line number 29–30). We also revised "the SML coverage" to "the wind speed" (page 14, line number 20; page 14, line number 27–28; page 18, line number 23). We also revised "In addition, these ratios showed strong correlations with DOC concentration in the Amundsen Sea when the wave breaking thoroughly mixes the SML with the underlying water." to "In addition, these ratios showed strong correlations with DOC concentration in the Amundsen Sea when the wave breaking thoroughly mixes the organic-enriched sea surface layer with the underlying relatively organic-poor water." (page 18, line number 25). We believe that we revised our manuscript properly according to Referee #4's comments.

**Minor Comments:**

1. P2 Line 11- P3 Line 5: These two paragraphs are about marine and biological CCN. Not sure such a detailed introduction is needed since CCN is mentioned in only section 1 and not discussed in later sections.

(Response) This study focuses on the characteristics of biogenically-derived aerosols, especially atmospheric sulfur and organic carbon species, over the Amundsen Sea. In the two paragraphs Referee #4 mentioned about, we described the formation processes and the importance of biogenically-derived atmospheric sulfur-containing and organic aerosols in the marine environment. Although Referee #4 questioned whether the detailed introduction is needed, we believe that the introduction of biogenically-derived sulfur-containing and organic aerosols is necessary.

2. P9 Line10-16: Since there are only 14 samples (or in this case 9 samples), removal of outliers needs better discussion. Please specify how "the highest mean wind speed" affects DMS flux and possibly provide either reference or calculation.

(Response) As Referee #4 suggested, we discussed the influence of wind speed on DMS flux. DMS fluxes typically rely on gas transfer velocity, which is frequently parameterized as a function of wind speed (Wanninkhof, 2014). Measurement and parameterization of the gas transfer velocity are more challenging and subject to greater uncertainty, particularly at high wind speeds (Smith et al., 2018). As the gas transfer velocity increases with increasing wind speed, DMS flux can be overestimated especially, in higher latitudes where DMS is commonly found at high concentrations in surface water, and where both low temperatures and high winds are typical (McGillis et al., 2000). Despite the uncertainty in DMS flux (Fig. 4), given the lifetime of DMS is approximately 1–2 days (Kloster et al., 2006; Read et al., 2008), our results

revealed that the local sea–air DMS flux affected directly atmospheric MSA concentration in the Amundsen Sea, and that the higher atmospheric MSA concentrations observed over the Amundsen Sea compared to those over the Southern Ocean and in coastal Antarctic regions were attributed to the higher DMS concentrations produced by *P. antarctica* and to the higher DMS fluxes in the Amundsen Sea. (page 9, line number 17–26). We also added the references (i.e., Wanninkhof, 2014; Smith et al., 2018, McGillis et al., 2000) to References (page 27, line number 28–29; page 27, line number 4–6; page 24, line number 4–5).

**3. P9 Line 28: "the variation trend of " Can the authors provide correlation coefficient?**

(Response) We already described about the correlation coefficients between $nss\text{-}SO_4^{2-}$ and MSA in the Amundsen Sea and Southern Ocean (page 10, line number 2–5).

**4. P11 Line 12: "somewhat overestimated... showing negative values" Can the authors discuss possibilities causing this overestimation?**

(Response) We already described the limitation in quantifying biogenically-derived $nss\text{-}SO_4^{2-}$ using the $MSA/nss\text{-}SO_4^{2-}$ ratio (page 10, line number 25–29). Nevertheless, according to Referee #4's comment, we added the following sentences to the manuscript. "However, if we apply for the highest $MSA/nss\text{-}SO_4^{2-}$ ratio (0.7) obtained from this study, the contributions of biogenically-derived $nss\text{-}SO_4^{2-}$ over the Southern Ocean and the Amundsen Sea decrease to $38 \pm 20\%$ (range: 17–73%) and $62 \pm 23\%$ (range: 28–100%), respectively. This result suggests that the estimate of biogenically-derived $nss\text{-}SO_4^{2-}$ contribution to total $nss\text{-}SO_4^{2-}$ using the $MSA/nss\text{-}SO_4^{2-}$ ratio could have limitation (Mungall et al., 2018) as mentioned above." (page 11, line number 18–22). We believe that we revised our manuscript properly according to Referee #4's comments.

**5. P11 Line 14: "variation trend" Again consider show correlation coefficient.**

(Response) According to Referee #4's comment, we added the correlation coefficient between $NO_3^-$ and anthropogenic $nss\text{-}SO_4^{2-}$ in the sentence (page 11, line number 23).

**6. P 17 Line 7: "negatively correlated with the relative biomass of P. antarctica (r =0.79, p < 0.05)" Should be "-0.79" if using the Pearson correlation coefficient. Otherwise, the authors should specify the sign. This also applies to some other place where r is less than 0 including some of the figures.**

(Response) According to Referee #4's comment, we revised "r = 0.79" to "r = –0.79" (page 17, line number 26). We also revised the correlation coefficient in Figure 10 (page 38).

**7. P18 Line 18: Consider citing this article (Bromwich et al., 2013).**

(Response) As Referee #4 suggested, we added Bromwich et al. (2013) to the manuscript with the following sentence: "West Antarctica is one of the fastest-warming regions globally." (page 19, line number 3). We also added Bromwich et al. (2013) to reference (page 20, line number 23–25).

Editorial Comments:

1. P4 Line 20: "Procedural blanks (n = 4)..." Change to "Four procedural blanks..."

(Response) According to Referee #4's comment, we revised "Procedural blanks (n = 4)…" to "Four procedural blanks…" (page 4, line number 20).

2. P13 Line 15:" reflecting that Na+ was formed from bubble by local wind speed. " Delete "speed".

(Response) According to Referee #4's comment, we deleted "speed" from the sentence (page 13, line number 24).

3. P16 Line 15-16:" our results strongly suggested that the submicron WSOC observed in the Amundsen Sea might be formed..." Remove "might"

(Response) According to Referee #4's comment, we revised "might be formed" to "was formed" (page 17, line number 2).

4. Figure 8c: Check unit

(Response) According to Referee #4's comment, we revised the unit of WSOC in Figure 8c "µg m$^{-3}$" to "µgC m$^{-3}$" (page 36).

**References**

Bromwich, D. H., Nicolas, J. P., Monaghan, A. J., Lazzara, M. A., Keller, L. M., Weidner, G. A. and Wilson, A. B.: Central West Antarctica among the most rapidly warming regions on Earth, Nature Geosci, 6(2), 1–8, doi:10.1038/ngeo1671, 2013.

Ceburnis, D., Rinaldi, M., Ovadnevaite, J., Martucci, G., Giulianelli, L. and O'Dowd, C. D.: Marine submicron aerosol gradients, sources and sinks, Atmos. Chem. Phys., 16(19), 12425–12439, doi:10.5194/acp-16-12425-2016, 2016.

McGillis, W. R., Dacey, J., Frew, N. M., Bock, E. J. and Nelson, R. K.: Water-air flux of dimethylsulfide, J. Geophys. Res. Oceans, 105(C1), 1187–1193, doi:10.1029/1999JC900243, 2000.

Rinaldi, M., Fuzzi, S., Decesari, S., Marullo, S., Santoleri, R., Provenzale, A., Hardenberg, Von, J., Ceburnis, D., Vaishya, A., O'Dowd, C. D. and Facchini, M. C.: Is chlorophyll-a the best surrogate for organic matter enrichment in submicron primary marine aerosol? J. Geophys. Res. Atmos., 118(10), 4964–4973, doi:10.1002/jgrd.50417, 2013.

Smith, M. J., Walker, C. F., Bell, T. G., Harvey, M. J., Saltzman, E. S. and Law, C. S.: Gradient flux measurements of sea-air DMS transfer during the Surface Ocean Aerosol Production (SOAP) experiment, Atmos. Chem. Phys., 18(8), 5861–5877, doi:10.5194/acp-18-5861-2018, 2018.

Wanninkhof, R.: Relationship between wind speed and gas exchange over the ocean revisited, Limnol. Oceanogr. Methods, 12, 351–362, doi:10.4319/lom.2014.12.351, 2014.

**Anonymous Referee #3:**

**Major comments:**

1. The first major comment reiterates the comment already raised by anonymous reviewer #4. The authors suggest that the inverse relationship between WSOC/Na+ and WIOC/Na+ with wind speed implies that organics in the atmosphere are controlled by winds due to the breakage of the surface microlayer for periods of times associated with high winds. However, I believe that there are several inconsistencies in the text that will need to be clearly addressed prior to publication, mainly:

a. Both WSOC/Na+ and WIOC/Na+ are inversely related to wind speeds but the authors claim that only WIOC have a primary source whereas WSOC is mostly formed by oxidation of biogenic precursors. How do the authors then explain that both ratios correlated well with wind speed?

(Response) We agree to Referee #3's comments. To explain these relationships more clearly, we have modified Figures 7c and 7d by adding in situ Chl-a concentrations averaged for each aerosol sampling time (page 35). Although in situ Chl-a was not measured during the cruise from New Zealand to the entrance to the Amundsen Sea due to the limited ship time, our results, shown in Figures 7c and 7d, suggest that the wind speed is a significant factor influencing the WSOC/Na$^+$ and WIOC/Na$^+$ ratios in our study region. According to Referee #3's comments, we revised our manuscript as follows: "In addition to marine biological activities, the wind speed has been reported to have an effect on the organic mass fraction of sea spray aerosol (Rinaldi et al., 2013; Ceburnis et al., 2016). Gantt et al. (2011) demonstrated that the organic mass fraction of sea spray aerosol depends not only on marine biological activity in oceanic surface waters but also on sea surface wind speed. During the sampling period, both WSOC/Na$^+$ and WIOC/Na$^+$ ratios showed inverse relationships with mean wind speed (r = –0.82, p < 0.01 and r = –0.76, p < 0.01, respectively) (Figs. 7c and 7d). The highest WSOC/Na$^+$ and WIOC/Na$^+$ ratios were observed in the aerosol sample (i.e., A5) collected in biologically active region (i.e., in situ Chl-a concentration was highest (6.9 mg m$^{-3}$)) under calm conditions (i.e., mean wind speed was lowest (4.4 m s$^{-1}$)), and the WSOC/Na$^+$ and WIOC/Na$^+$ ratios drastically decreased with increasing mean wind speed. This result suggests that organic matters produced in high biological activity conditions were enriched in the sea surface water under low wind speed conditions, making a favorable condition for transferring organic matters to the atmosphere through the bubble bursting process (Rinaldi et al., 2013). However, as shown in Figures 7c and 7d, the WSOC/Na$^+$ and WIOC/Na$^+$ ratios observed in the biologically active region (i.e., higher in situ Chl-a concentrations) did not always show higher values when the wind speeds exceeded 7 m s$^{-1}$. According to previous studies (Gantt et al., 2011; Rinaldi et al., 2013), the wave breaking caused by high wind speeds (> 8 m s$^{-1}$) thoroughly mixes the organic-enriched sea surface layer with the underlying relatively organic-poor waters resulting in the homogeneous water column and considerably reduces the organic mass fraction of sea spray aerosol. Therefore, the inverse relationships between WSOC/Na$^+$, WIOC/Na$^+$ ratios and the mean wind speed suggest that the wind speed is a significant factor influencing the organic mass fractions of sea spray aerosols in our study region." (page 14, line number 3–19). Besides, we added the references (i.e., Rinaldi et al., 2013; Ceburnis et al., 2016) to the manuscript (page 26, line number 4–6; page 20, line number 29–

30). We also revised "the SML coverage" to "the wind speed" (page 14, line number 20; page 14, line number 27–28; page 18, line number 23). We also revised "In addition, these ratios showed strong correlations with DOC concentration in the Amundsen Sea when the wave breaking thoroughly mixes the SML with the underlying water." to "In addition, these ratios showed strong correlations with DOC concentration in the Amundsen Sea when the wave breaking thoroughly mixes the organic-enriched sea surface layer with the underlying relatively organic-poor water." (page 18, line number 25). We believe that we revised our manuscript properly according to Referee #3's comments.

b. The authors hypothesize that WIOC is of primary origin, but their measurements indicate no correlation between WIOC and Na+. The authors attribute this to transport but their measurements clearly indicate a correlation between Na+ and wind speed, suggesting transport cannot completely rule out a correlation between WIOC and Na+. Also, why does WIOC correlate with the relative abundance of P. Antarctica? Presumably, this implies that a part of WIOC is formed from oxidation of BVOC?

(Response) In section 3.7, we discussed the insignificant relationships between WIOC and $Na^+$ in the fine modes over the Southern Ocean and the Amundsen Sea. As mentioned in section 3.6, the wave breaking caused by high wind speed thoroughly mixes the organic-rich sea surface layer with the underlying relatively organic-poor waters resulting in the homogeneous water column and considerably reduces the organic mass fraction of sea spray aerosol. Thus, these insignificant relationships between WIOC and $Na^+$ in the fine modes could result from the differences in local wind speeds and local biological activities, such as sea surface DOC concentration, because wind speed, a key factor determining sea spray aerosols, controls the local flux rather than local concentration of marine particles (page 15, line number 33–page 16, line number 6). Besides, as Referee #3 pointed out, the WIOC production by secondary processes cannot be excluded. Thus, we added the following sentence to the manuscript: "However, the WIOC production by secondary processes cannot be completely excluded either (Ceburnis et al., 2016), but we have no evidence of that (page 16, line number 11–12).

Referee #3 asked why WIOC concentration showed a good correlation with the relative biomass of *P. antarctica* in the Amundsen Sea. As we already described in section 3.6, phytoplankton exudates include exopolymer gels consisting of polysaccharides, which are insoluble, thermally stable, highly surface active, highly hydrated and readily sequester dissolved organic matter (Quinn and Bates, 2011 and references therein). Besides, *P. antarctica*, which was a dominant phytoplankton species in the Amundsen Sea (see section 3.2), generates a substantial amount of extracellular polysaccharide mucus in its colonial matrix, and it has been suggested that large amounts of this material ultimately enter the DOC pool (Smith et al., 1998). Thus, the significant correlation ($r = 0.87$, $p < 0.05$) between WIOC and the relative biomass of *P. antarctica* in the Amundsen Sea suggests that water-insoluble organic matter released by *P. antarctica* was emitted in the Amundsen Sea via bubble bursting and breaking waves (page 15, line number 7–10). We believe that we revised our manuscript properly according to Referee #3's comments.

c. Finally, why do WSOC concentrations (presumably of secondary origin, i.e., from the oxidation of BVOC) correlate strongly with Na+ and with DOC? The authors hypothesize that the WSOC relation on Na+ is due to higher surface area (from salt particles) and therefore higher WSOC concentrations are a result of a larger condensation sink. However, the authors present no compelling evidence to substantiate that hypothesis. Can the author examine the relation between WSOC and average short wavelength radiation? This could provide the evidence needed to argue that WSOC is formed from secondary sources. The linkage between WSOC and WIOC with biology needs to be better articulated.

(Response) As we already described in section 3.7, submicron WSOC showed a strong correlation (r = 0.94, p < 0.01) with submicron $Na^+$ in the Amundsen Sea (Fig. 8b). In addition, we also found a significant correlation (r = 0.93, p < 0.01) between WSOC and MSA concentrations in the Amundsen Sea (Fig. 8c). However, in the Southern Ocean, WSOC showed no significant relationship with submicron $Na^+$ or MSA. MSA is produced by atmospheric oxidation of DMS, which is released as a gas phase from marine biological activities and thus can be used as an indicator of secondary aerosols of marine biogenic origin (Miyazaki et al., 2011). As described in section 3.2, Kim et al. (2017) observed extremely high DMS concentrations (> 150 nM) in surface water during our cruise, and MSA concentration showed a strong correlation with DMS flux in the Amundsen Sea. Consequently, the strong correlations between WSOC, $Na^+$ and MSA in the Amundsen Sea implies that the Amundsen Sea that has the most productive polynya in the Antarctic is a strong source region of BVOCs, and that WSOC was formed by the condensation of BVOCs released from sea surface onto preexisting submicron sea spray aerosols through gas-to-particle conversion due to a higher surface-to-volume ratio of submicron aerosols (Romakkaniemi et al., 2011). On the other hand, the poor correlations between WSOC, $Na^+$, and MSA in the Southern Ocean implies the differences in local source strength of BVOCs and that the presence of DMS in seawater and its subsequent oxidation to MSA were not necessarily linked to the formation of submicron WSOC over the Southern Ocean (Miyazaki et al., 2016). Therefore, our results provide the evidence that WSOC was formed by secondary processes in the Amundsen Sea (page 16, line number 13–26).

Referee #3 questioned that why WSOC concentration showed a strong correlation with Na+. However, WSOC concentration did not show a good correlation with DOC, but WSOC/$Na^+$ ratio did, as shown in Figure 7e. We already discussed the reason why WSOC/$Na^+$ was correlated with DOC in section 3.6 (page 14, line number 20–page 15, line number 17).

Unfortunately, we do not have any short wavelength radiation data. However, it would be interesting to investigate the relationship between WSOC and short wavelength radiation in the future study. Although we cannot discuss the relationship between WSOC and short wavelength radiation, the significant relationship between WSOC, MSA and $Na^+$ in the Amundsen Sea (Figs. 8b and 8c) suggest that WSOC was formed by the condensation of BVOCs released from sea surface onto preexisting submicron sea spray aerosols through gas-to-particle conversion as we mentioned above.

Referee #3 pointed out that the linkage between WSOC and WIOC with biology needs to be better articulated. However, in section 3.6, we already discussed the significant relationship between WIOC concentration and the relative biomass of *P. antarctica* in the Amundsen Sea, suggesting that water-insoluble organic matter released by *P. antarctica* was emitted in the Amundsen Sea via bubble bursting and breaking waves (page 15, line number 9–10). Besides, in section 3.8, we investigated the fluorescence properties of WSOC over the Southern Ocean and the Amundsen Sea. Interestingly, fluorescence intensity of C1 (i.e., protein-like component) showed a significant positive relationship with the relative biomass of diatoms ($r = 0.89$, $p < 0.01$); however, it was negatively correlated with the relative biomass of *P. antarctica* ($r = -0.79$, $p < 0.05$) (Figs. 10c and 10d). These results suggest that protein-like component is most likely produced as a result of biological processes of diatoms, which play a key role in forming the submicron WSOC observed over the Southern Ocean and the Amundsen Sea, and that phytoplankton community structure is a significant factor affecting atmospheric OC species since the submicron WIOC was quite related to the relative biomass of *P. antarctica* (page 17, line number 24–page 18, line number 2). Thus, we believe that we discussed properly the linkage between WSOC and WIOC with biology in our manuscript.

2. What happens to the correlations in Figures 7 if the authors looked at WSOC and WIOC instead of WSOC/Na+ and WIOC/Na+? This could remove the cross-correlation with wind speed (I encourage the authors to add these graphs as part of the SI).

(Response) As Referee #3 suggested, we added the plots of WIOC concentration in fine mode versus mean wind speed and WSOC concentration in fine mode versus mean wind speed to Figure S6 in Supplementary material. We also added the following sentence to our manuscript: "In addition, no relationship was found between the submicron WIOC and mean wind speed (Fig. S6)." (page 15, line number 32).

3. The authors often mention correlation strength even though the p-value for the regression is higher than the specified threshold (in this case 0.05). A p-value larger than the threshold implies there is no confidence in rejecting the null hypothesis (which is: the variables are not correlated). I suggest the authors reframe their results accordingly.

(Response) We agree to Referee #3's comment. According to Referee #3's comments, we revised our manuscript as follows: "Atmospheric MSA concentration showed no relationship with either in situ sea surface Chl-a concentration ($r = 0.029$, $p > 0.05$) (Fig. S2) or the relative biomass of *P. antarctica* ($r = 0.30$, $p > 0.05$)." (page 9, line number 4–6), "but no correlation was found between atmospheric MSA and DMS flux ($r = 0.18$, $p > 0.05$, Fig. 4b)" (page 9, line number 9–10), "atmospheric MSA concentration showed a significant relationship with DMS flux ($r = 0.78$, $p < 0.05$)." (page 9, line number 12), "A similar result was found between atmospheric MSA and DMS concentrations in surface water, showing no correlation ($r = 0.20$, $p > 0.05$)." (page 9, line number 14–15), "whereas no relationship was found between them in the Southern Ocean ($r = 0.51$, $p > 0.05$) (Fig. S3)" (page 10, line number 3). We also removed "Similar results were found between fluorescence intensity of C2 and the relative biomass of diatoms and *P. antarctica*." because the fluorescence intensity of C2 showed no relationship with the relative biomass of diatoms or

and *P. antarctica* (this sentence was written on page 17, line number 7-8 in the unrevised manuscript). Besides, we modified Figure 10 (==page 38==), and revised "…by showing significant positive relationships between the relative biomass of diatoms and protein-like components in marine aerosols in the Amundsen Sea." to "…by showing the significant positive relationship between the relative biomass of diatoms and protein-like component in marine aerosols in the Amundsen Sea." (==page 19, line number 1–2==). We also revised "Protein-like components also showed positive relationships with the relative biomass of diatoms; however, they were negatively correlated with the relative biomass of *P. antarctica*. These results suggest that protein-like components are most likely produced as a result of biological processes of diatoms, which play a crucial role in forming the submicron WSOC observed over the Southern Ocean and the Amundsen Sea, and that phytoplankton community structure is a significant factor affecting atmospheric organic carbon species." to "Protein-like component also showed a significant positive relationship with the relative biomass of diatoms; however, it was negatively correlated with the relative biomass of *P. antarctica*. These results suggest that protein-like component is most likely produced as a result of biological processes of diatoms, which play a crucial role in forming the submicron WSOC observed over the Southern Ocean and the Amundsen Sea, and that phytoplankton community structure is a significant factor affecting atmospheric organic carbon species." (==page 1, line number 30–page 2, line number 2==).

**Minor comments:**

P7 L29: specify at least once what +/- refers to, one standard error or the one standard deviation?

(Response) As Referee #3 suggested, we added (mean ± one standard deviation) to the sentence (==page 7, line number 29==).

P9 L9-L10: The authors describe a weak but not significant relation between MSA and DMS. If the p-value is not significant than the authors cannot justify that a positive or negative correlation (see major comment 2). Please adjust elsewhere in the text where applicable.

(Response) As Referee #3 already pointed out in major comment 3, we revised our manuscript according to Referee #3's comment. Please see our response to major comment 3.

P9 L13: The authors should provide more thorough reasoning for removing the point to the right. Indeed, removing that one point changes the correlation between DMS and MSA from non-existent to significant, due to the small sample number.

(Response) As Referee #3 suggested, we discussed the influence of wind speed on DMS flux. DMS fluxes typically rely on gas transfer velocity, which is frequently parameterized as a function of wind speed (Wanninkhof, 2014). Measurement and parameterization of the gas transfer velocity are more challenging and subject to greater uncertainty, particularly at high wind speeds (Smith et al., 2018). As the gas transfer velocity increases with increasing wind speed, DMS flux can be overestimated especially, in higher latitudes where DMS is commonly found at high concentrations in surface water, and where both low temperatures and high winds are typical (McGillis et al., 2000). Despite the uncertainty in DMS flux (Fig.

4), given the lifetime of DMS is approximately 1–2 days (Kloster et al., 2006; Read et al., 2008), our results revealed that the local sea–air DMS flux affected directly atmospheric MSA concentration in the Amundsen Sea, and that the higher atmospheric MSA concentrations observed over the Amundsen Sea compared to those over the Southern Ocean and in coastal Antarctic regions were attributed to the higher DMS concentrations produced by *P. antarctica* and to the higher DMS fluxes in the Amundsen Sea. (page 9, line number 17–26). We also added the references (i.e., Wanninkhof, 2014; Smith et al., 2018, McGillis et al., 2000) to References (page 27, line number 28–29; page 27, line number 4–6; page 24, line number 4–5).

P9 L27-30: "Unlike MSA, the mean nss-SO42- concentration in the Amundsen Sea. . ." please provide justification to how the trend for nss-sulfate was comparable to that of MSA. Visual inspection of the data is not sufficient. Also, I am not sure how the authors arrived at the conclusion that nss-sulfate is influenced by marine and anthropogenic sources given the information in that sentence.

(Response) We already described about the correlations between $nss\text{-}SO_4^{2-}$ and MSA in the Amundsen Sea and Southern Ocean (page 10, line number 2–7). Besides, the reason why we believe that $nss\text{-}SO_4^{2-}$ was affected by both marine and anthropogenic sources is already described in section 3.4 (page 10, line number 19–page 12, line number 3).

P10 L17: ". . . showing a similar variation trend to that of MSA". Quantify the agreement.

(Response) As Referee #3 suggested, we added "(r = 0.92, p < 0.01)" to the sentence (page 10, line number 32).

P11 L14-19: "We expected much higher WSOC and WIOC concentrations in the Amundsen Sea than the Southern Ocean because of extremely high Chl-a concentrations. . .". This result could be consistent with findings of Quinn et al. (2014) who argued that organic enrichment in the aerosol phase is likely controlled by the dissolved organic carbon (DOC) pool rather than chlorophyll concentrations.

(Response) According to Referee #3's comment, we added "(Quinn et al., 2014, see section 3.6) to the sentence (page 12, line number 14).

P10 L20-21: Specify how the enrichment of WSOC and WIOC in the aerosol phase was calculated.

(Response) According to Referee #3's comment, we added "(i.e., the percentage of WSOC or WIOC present in fine aerosol particles)" to the sentence (page 12, line number 16–17).

P10 L23: "For example, O'Dowd et al. (2004) observed that the contribution of OC fraction to the submicrometer aerosol mass increased from 15% to 63% between low and high biological activity periods in the North Atlantic." I am confused by this sentence. The measurements presented in this article show no relation of WSOC and WIOC on chl-a concentrations (as explicitly written in the previous paragraph). Please clarify.

(Response) We referred to O'Dowd et al. (2004) to described that WIOC in fine mode aerosol particles is

dominant OC species during bloom periods, as our result showed it. We, therefore, revised our manuscript as follows: "For example, O'Dowd et al. (2004) observed a dominant water-insoluble OC fraction (~45%) in fine marine aerosol collected during periods of phytoplankton bloom in the North Atlantic." (page 12, line number 20–21).

P11 L31: "the dominance of WIOC suggests that the bubble bursting process by local wind speeds is a significant formation mechanism of atmospheric WIOC in our area." If this is true, then why don't the author observe lower WIOC concentration over the Amundsen Sea compared to the Southern Ocean given that wind speed in the Southern Ocean were larger compared to over the Amundsen Sea?

(Response) We agree to Referee #3's comment. Therefore, we revised our manuscript as follows: "the dominance of WIOC suggests that water-insoluble organic matter exuded by phytoplankton is more accumulated in sea surface water and emitted into the marine atmosphere via bubble bursting and breaking waves by local wind." (page 12, line number 26–28).

**References**

Ceburnis, D., Rinaldi, M., Ovadnevaite, J., Martucci, G., Giulianelli, L. and O'Dowd, C. D.: Marine submicron aerosol gradients, sources and sinks, Atmos. Chem. Phys., 16(19), 12425–12439, doi:10.5194/acp-16-12425-2016, 2016.

McGillis, W. R., Dacey, J., Frew, N. M., Bock, E. J. and Nelson, R. K.: Water-air flux of dimethylsulfide, J. Geophys. Res. Oceans, 105(C1), 1187–1193, doi:10.1029/1999JC900243, 2000.

Rinaldi, M., Fuzzi, S., Decesari, S., Marullo, S., Santoleri, R., Provenzale, A., Hardenberg, Von, J., Ceburnis, D., Vaishya, A., O'Dowd, C. D. and Facchini, M. C.: Is chlorophyll-a the best surrogate for organic matter enrichment in submicron primary marine aerosol? J. Geophys. Res. Atmos., 118(10), 4964–4973, doi:10.1002/jgrd.50417, 2013.

Smith, M. J., Walker, C. F., Bell, T. G., Harvey, M. J., Saltzman, E. S. and Law, C. S.: Gradient flux measurements of sea-air DMS transfer during the Surface Ocean Aerosol Production (SOAP) experiment, Atmos. Chem. Phys., 18(8), 5861–5877, doi:10.5194/acp-18-5861-2018, 2018.

Wanninkhof, R.: Relationship between wind speed and gas exchange over the ocean revisited, Limnol. Oceanogr. Methods, 12, 351–362, doi:10.4319/lom.2014.12.351, 2014.

**We already have revised our manuscript according to Referee #1's comments and uploaded the revised manuscript file on 19 March 2019 before open discussion started. Just in case, we show our responses to Referee #1 again.**

**Referee #1:**

1. The authors present measurements of aerosol and ocean carbon from the southern ocean and Amundsen Sea linking biological processes to marine particle concentrations and composition. This is an important processes, though not well understood, partly due to the interdisciplinary aspect of the subject. Better understanding this process possibly has significant implication on understanding future climate change. The authors do well to discuss and provide evidence for their claims. I am suggesting the article be accepted after major revisions. The main concerns I have involve the claims made in section 3.3-3.4, specifically that a significant amount of nssSO4 is from anthropogenic sources. While possible, I do not think they have provided strong evidence. Previous studies in the SO have shown that almost no continental/anthropogenic pollution is transported over the southern ocean particularly south of ~50S (See Hudson et al. 1998). Hudson et al. 1998 measurements are in a different part of the Southern Ocean, I expect the result to be similar. A back trajectory analysis would strengthen your case if there really is anthropogenic sources.

(Response) We thank for the comments from Referee #1. Referee #1 pointed out a significant amount of anthropogenic nss-$SO_4^{2-}$ in our study regions. We have already described that concentration of anthropogenic nss-$SO_4^{2-}$ estimated using the MSA/nss-$SO_4^{2-}$ ratio of 0.508 observed at Palmer Station, varied from 0.10–0.63 µg m$^{-3}$ (mean: 0.34 ± 0.18 µg m$^{-3}$) in the Southern Ocean, and 0.13–0.30 µg m$^{-3}$ (mean: 0.19 ± 0.079 µg m$^{-3}$) in the Amundsen Sea (Figs. 5c and 5d) (page 11, line number 2–16). To clarify the context, we have added a short explanation for the contributions of anthropogenic nss-$SO_4^{2-}$ in the Southern Ocean and the Amundsen Sea to section 3.4 (page 11, line number 16–17).

As referee #1 suggested, we have read the paper (Hudson et al., 1998). However, Hudson et al. (1998) reported only the number concentrations of condensation nuclei observed over the Southern Ocean, so we could not refer to it. To compare the anthropogenic nss-$SO_4^{2-}$ concentrations estimated in this study to previously published results, we have compared our anthropogenic nss-$SO_4^{2-}$ concentration estimated in this study to the results by Jung et al. (2014) who reported that mean concentration of nss-$SO_4^{2-}$ observed over the South Pacific ($10°S-55°S$, January–March 2009) was 0.25 ± 0.17 µg m$^{-3}$. Our mean concentration of anthropogenic nss-$SO_4^{2-}$ in the Southern Ocean was a factor of 1.4 higher than the result by Jung et al. (2014). In comparison, our mean concentration of anthropogenic nss-$SO_4^{2-}$ in the Amundsen Sea was a factor of 1.3 lower than the result by Jung et al. (2014) (page 11, line number 17–21).

As referee #1 suggested, we have calculated 7-day air mass backward trajectories and added it to the Supplement (Fig. S4). We also have added short explanations for the backward trajectory results to section 3.4 (page 11, line number 21–26).

2. The nssSO4 likely correlates strongly to MSA in the Amundsen Sea simply because the nssSO4 source (MSA) is local. It is common for nssSO4 to be transported large distance, often in the free troposphere, resulting in little to no correlation with local sources (MSA). (see Sanchez et al. 2018). While MSA is often used as a tracer, a correlation may not always be found. Gas phase and aqueous phase oxidation of MSA to sulfate is known to take place, potentially leading to MSA having a lifetime of only a few days (Mungall et al 2018).

(Response) As referee #1 suggested, we have investigated the relationship between $nss\text{-}SO_4^{2-}$ and MSA in the Southern Ocean and the Amundsen Sea and added the result to the Supplement (Fig. S3). We also have added the following sentences to section 3.3. "Indeed, $nss\text{-}SO_4^{2-}$ showed a strong correlation ($r = 0.98$, $p < 0.01$) with MSA in the Amundsen Sea, whereas the relationship in the Southern Ocean was statistically insignificant ($r = 0.51$, $p > 0.05$) (Fig. S3), suggesting that the local emission of DMS is a significant source of $nss\text{-}SO_4^{2-}$ in the Amundsen Sea and that $nss\text{-}SO_4^{2-}$ in the Amundsen Sea was formed by the condensation of DMS products onto existing particles (Sanchez et al., 2018). On the other hand, the statistically insignificant relationship between $nss\text{-}SO_4^{2-}$ and MSA could result from strong influence of anthropogenic sources and low biological activity in the Southern Ocean" (page 9, line number 30–page 10, line number 2). We also have added Sanchez et al. (2018) as a reference in the manuscript.

As Referee #1 mentioned, we have read the paper (Mungall et al., 2018) and added the following sentences to section 3.4. "Mungall et al. (2018), however, pointed out that the $MSA/nss\text{-}SO_4^{2-}$ ratio could have limitation that may preclude its use in quantitatively unravelling the chemical and biological processes at play in the marine boundary layer due to the conversion of MSA to $nss\text{-}SO_4^{2-}$ by OH radical in aerosol particles (the order of days to weeks), although it remains useful as a qualitative indicator of marine biological influence" (page 10, line number 20–24). We also have added Mungall et al. (2018) as a reference in the manuscript

3. While nitrate can be attributed to anthropogenic emission, particle nitrate concentration (and nssSO4 concentration) concentration can also be enhanced by aqueous phase processing (in clouds). On Figure 5d, with the exception of samples A1 and A13 (both of which are between 45S and 50S on figure 1a) the nitrate concentration seems fairly consistent.

(Response) We have agreed to Referee #1's opinion. However, we have used $NO_3^-$ solely as an indicator of anthropogenic contribution in this study. Moreover, we have focused on the characteristics of biogenically-derived sulfur and organic carbon species in our study regions. Thus, we believe that further discussions on $NO_3^-$ are beyond the scope of our manuscript.

4. I would be surprised to find that, at times, anthropogenic sources account for more than half the nssSO4 even in the Amundsen Sea (your range is 39%-138%). There are no relatively nearby anthropogenic sources to account for this. More evidence would be needed to make such a claim.

(Response) Referee #1 pointed out that the contribution of anthropogenic nss-SO$_4^{2-}$ to total nss-SO$_4^{2-}$ in the Amundsen Sea was unreasonably high (range: 39–138%), even though the influence of anthropogenic sources is weak in the Amundsen Sea. However, what Referee #1 pointed out was not the contribution of anthropogenic nss-SO$_4^{2-}$, but that of biogenically-derived one in the Amundsen Sea. In our manuscript, we have described in section 3.4 that mean concentrations of biogenically-derived nss-SO$_4^{2-}$ over the Southern Ocean and the Amundsen Sea were estimated to be $0.31 \pm 0.19$ µg m$^{-3}$ (range: 0.074–0.57 µg m$^{-3}$) and $0.56 \pm 0.30$ µg m$^{-3}$ (range: 0.19–1.1 µg m$^{-3}$), accounting for $52 \pm 28\%$ (range: 24–101%) and $86 \pm 32\%$ (range: 39–138%) of total nss-SO$_4^{2-}$, respectively (Fig. 5b) (page 11, line number 5–8). We therefore believe that Referee #1 might have misunderstood our manuscript.

References

Hudson, J. G., Xie, Y. and Yum, S. S.: Vertical distributions of cloud condensation nuclei spectra over the summertime Southern Ocean, J. Geophys. Res., 103, 16,609–16,624, 1998.

Mungall, E. L., Wong, J. P. S. and Abbatt, J. P. D.: Heterogeneous oxidation of particulate methanesulfonic acid by the hydroxyl radical: Kinetics and atmospheric implications, ACS Earth Space Chem., 2(1), 48–55, doi:10.1021/acsearthspacechem.7b00114, 2018.

Jung, J., Furutani, H., Uematsu, M. and Park, J.: Distributions of atmospheric non-sea-salt sulfate and methanesulfonic acid over the Pacific Ocean between 48°N and 55°S during summer, Atmos. Environ., 99(C), 374–384, doi:10.1016/j.atmosenv.2014.10.009, 2014.

Sanchez, K. J., Chen, C.-L., Russell, L. M., Betha, R., Liu, J., Price, D. J., Massoli, P., Ziemba, L. D., Crosbie, E. C., Moore, R. H., Mueller, M., Schiller, S. A., Wisthaler, A., Lee, A. K. Y., Quinn, P. K., Bates, T. S., Porter, J., Bell, T. G., Saltzman, E. S., Vaillancourt, R. D. and Behrenfeld, M. J.: Substantial seasonal contribution of observed biogenic sulfate particles to cloud condensation nuclei, Sci. Rep., 8(1), doi:10.1038/s41598-018-21590-9, 2018.

---

## Author Response (AR2)

**Responses to Reviewers' comments**

22/09/2019

Journal: Atmospheric Chemistry and Physics

Title: Characteristics of biogenically-derived aerosols over the Amundsen Sea, Antarctica

Authors: Jinyoung Jung, Sang-Bum Hong, Meilian Chen, Jin Hur, Liping Jiao, Youngju Lee, Keyhong Park, Doshik Hahm, Jung-Ok Choi, Eun Jin Yang, Jisoo Park, Tae-Wan Kim, and SangHoon Lee

Manuscript number: acp-2019-133

Manuscript type: Research Article

Note: Reviewers' comments are high lightened in **blue**, while our responses to reviewers are in **black**. The revisions in the manuscript was highlighted to make easily readable for the editors and reviewers.

Prof. Lynn M. Russell

Co-Editor

Atmospheric Chemistry and Physics

USCD, Scripps Institution of Oceanography

United States

Dear Professor Russell,

We have attached an electronic copy of manuscript file ready to go to press entitled "Characteristics of biogenically-derived aerosols over the Amundsen Sea, Antarctica" by Jinyoung Jung, Sang-Bum Hong, Meilian Chen, Jin Hur, Liping Jiao, Youngju Lee, Keyhong Park, Doshik Hahm, Jung-Ok Choi, Eun Jin Yang, Jisoo Park, Tae-Wan Kim, and SangHoon Lee for publication in Atmospheric Chemistry and Physics (acp-2019-133). We have modified our text based on the referee's comments. We appreciated that the comments from referees improved our manuscript a lot. We believe that the comments from referees were clearly responded in our modified manuscript.

We are looking forward to hearing about your decision. Thank you for your consideration.

Sincerely yours,

Jinyoung Jung

In the first review I asked the authors to provide additional evidence on the contribution of anthropogenic nssSO4, mainly due to the fact that the authors claimed the anthropogenic contribution accounted for more than half the nssSO4 at times (Biological nssSO4 accounted for as little as 39% therefore anthropogenic nssSO4 accounted for the remaining 61%). The method used by the authors to estimate the contribution of anthropogenic nssSO4 relies on the use of the MSA to nssSO4 ratio. I disagree with the accuracy of this method based on publications showing the lack of correlation due to long range transport of particles and previously mentioned limitations of the method due to the conversion of MSA to nssSO4 (the latter has been incorporated in the text by the authors). While the authors have indicated that the method can be useful as a qualitative indicator, they have provided quantitative results. The significant contribution of anthropogenic nssSO4 indicated in this manuscript is a major claim if accurate. The accuracy of this method is further questionable because of the applied MSA/nssSO4 ratio, which is derived from Palmer station measurements. While this location is geographically closest these ratios have been shown to range .065 to .508 (Savoie and prospero 1989, Arimoto et al 1996, ) and the contrast in the biological activity in the Asmundsen Sea and Palmer Station with the Southern Ocean would suggest a single value is not a good representation especially if providing quantitative results. The complications of calculating the non-biogenic contribution of nssSO4 is also discussed in Legrand et al 2017, Legrand and Pasteur 1998 and Piel et al. 2006. Finally, the negative values of anthropogenic nssSO4 mentioned are also indicative of the limitations in the accuracy of the method. Therefore, I find it necessary to provide more evidence due to the limitations in using the MSA/nssSO4 as an indicator, or the removal of quantitative results on the contribution of anthropogenic nssSO4.

(Response) We really appreciate for Referee #1's comments. We have realized that the method using the MSA/nss-SO$_4^{2-}$ ratio for the quantitative estimation of anthropogenic nss-SO$_4^{2-}$ have limitations, as Referee #1 pointed out. Now we have found out that the high contribution of anthropogenic nss-SO$_4^{2-}$ in our study region, especially in the Southern Ocean, was because we used the single value of MSA/nss-SO$_4^{2-}$ ratio observed at Palmer Station. Referee #1 also pointed out that more evidence should be provided to explain the high contribution of anthropogenic nss-SO$_4^{2-}$ in our study region. Unfortunately, we do not have any data on another anthropogenic tracer, such as [210]Pb. Consequently, we have decided to remove the description of the anthropogenic nss-SO$_4^{2-}$ contribution, according to Referee #1's comments. In addition, we have used the MSA/nss-SO$_4^{2-}$ ratios published by previous studies (i.e., Berresheim et al., 1990; Ayers et al., 1991; Bates et al., 1992; Savoie et al., 1993; Legrand and Pasteur, 1998; Chen et al., 2012; Jung et al., 2014) that were carried out in the Southern Ocean and coastal Antarctic stations during the austral summer to estimate the contribution of biogenically-derived nss-SO$_4^{2-}$.

According to Referee #1's comments, we have revised section 3.4 as follows: "During the cruise, the

MSA/nss-SO$_4^{2-}$ ratio in bulk aerosols varied from 0.12 to 0.70 (mean: 0.35 ± 0.17), with lower ratios in marine aerosols collected over the Southern Ocean (range: 0.12–0.51, mean: 0.26 ± 0.14) and higher values over the Amundsen Sea (range: 0.20–0.70, mean: 0.44 ± 0.16), showing a similar variation trend (r = 0.92, p < 0.01) to that of MSA (Fig. 5b). This result suggests that atmospheric MSA plays a key role in the variation in MSA/nss-SO$_4^{2-}$ ratio over the Southern Ocean and the Amundsen Sea during the austral summer since atmospheric nss-SO$_4^{2-}$ concentrations in the Southern Ocean were quite comparable to the values in the Amundsen Sea. It is worth noting that this study was carried out over the Southern Ocean and the Amundsen Sea during the austral summer. Consequently, to estimate the contribution of biogenically-derived nss-SO$_4^{2-}$ to total nss-SO$_4^{2-}$, it is required to use the MSA/nss-SO$_4^{2-}$ ratios observed through long-term monitoring in the Southern Ocean and the Amundsen Sea during the austral summer. There have been several field studies investigating the MSA/nss-SO$_4^{2-}$ ratio in the Southern Ocean during the austral summer (range: 0.32–0.53, the South Pacific (40°S−45°S), Berresheim et al., 1990; range: 0.12–0.24, Cape Grim (40°41′S , 144°41′E), Ayers et al., 1991; range: 0.17–0.32, the South Pacific (30°S−60°S), Bates et al., 1992; range: 0.096–0.49, the South Pacific (40°S−56°S), Jung et al., 2014). In this study, we applied an average value (0.29 ± 0.16) of the MSA/nss-SO$_4^{2-}$ ratios reported by previous studies mentioned above, to estimate biogenically-derived nss-SO$_4^{2-}$ in the Southern Ocean. For the Amundsen Sea, due to the lack of long-term monitoring data for MSA/ nss-SO$_4^{2-}$ in the Amundsen Sea, we also applied an average value (0.40 ± 0.29) of the MSA/nss-SO$_4^{2-}$ ratios measured in the coastal Antarctic region (range: 0.15–0.93, 50°S−70°S, Chen et al., 2012) and the coastal Antarctic stations (mean: 0.508, Palmer Station, Savoie et al., 1993; mean: 0.31 ± 0.05, Halley Station, Legrand and Pasteur, 1998; mean: 0.33 ± 0.06, Neumayer Station, Legrand and Pasteur, 1998; mean: 0.17 ± 0.02, Dumont D'Urville Station, Legrand and Pasteur, 1998) during the austral summer. Based on the average values of MSA/nss-SO$_4^{2-}$ ratios measured by previous studies in the Southern Ocean and Antarctic coastal regions, mean contributions of biogenically-derived nss-SO$_4^{2-}$ to the total nss-SO$_4^{2-}$ in the Southern Ocean and the Amundsen Sea were estimated to be 91 ± 48% and 109 ± 40%, respectively, suggesting that biogenically-derived nss-SO$_4^{2-}$ dominates the atmospheric budget of nss-SO$_4^{2-}$ in the Southern Ocean and the Amundsen Sea. However, the contributions of biogenically-derived nss-SO$_4^{2-}$ for several aerosol samples, where the MSA/nss-SO$_4^{2-}$ ratios were higher than the average values, were somewhat overestimated. The most likely cause of the overestimation, especially in the Amundsen Sea, is the differences of biological productivity and DMS source strength in the ocean adjacent to the coastal Antarctic stations since we used the average value of the MSA/nss-SO$_4^{2-}$ ratios measured in various geographical locations around Antarctica due to the lack of data for sulfur-containing aerosols over the coastal Antarctic region. Further observational field studies, therefore, are required to fill the data gap over the Southern Ocean and coastal Antarctic region." (Page 11, line number 20−page 12, line number 15).

Besides, we have removed from section 3.4 the following sentence: ", and that atmospheric nss-SO$_4^{2-}$ observed in the Southern Ocean was affected by anthropogenic sources" (page 11, line number 23−25).

We also have modified Figure 5 by removing concentrations of biogenic and anthropogenic nss-SO$_4^{2-}$ from Figure 5 (page 35) and have added the references (i.e., Berresheim et al., 1990; Ayers et al., 1991) to the manuscript (page 21, line number 9−11; page 20, line number 28−29).

The text below is each response to the reviewer responses to Reviewer #1.

1. No there were not anthropogenic nssSO4 concentrations in Hudson et al 1998, however, anthropogenic nssSO4 concentrations is not the only way of determining if the marine boundary layer is influenced by anthropogenic pollutants. Hudson et al. 1998 looked at particle concentration and back trajectories to identify cases that were influenced by continental/anthropogenic particles and found only the closest measurements (north of -44 S) were influenced by continental/anthropogenic particles. The cases with back trajectories over the continent consistently contained higher particle concentrations.

It is impossible to follow the back trajectories from their starting point. There are too many for a single plot. There are only a few trajectories that have intercepted with continental sources. Do these correspond to cases with higher anthropogenic nssSO4? I suspect the main anthropogenic source would be from over continents. If the authors have other references to suggest there are other major sources of anthropogenic nssSO4 in the Southern Ocean, then they should be included in the text.

(Response) We thank Referee #1 for Referee #1's comments. As Referee #1 suspected, we observed high nss-SO$_4^{2-}$ and NO$_3^-$ concentrations during the collection of aerosol samples A1 and A13 when air masses originated from the Southern Ocean and thereafter swept over large regions of New Zealand (Fig. S4) (page 10, line number 24−26). In addition, we could not find any reference to suggest that there are other major anthropogenic sources of nss-SO$_4^{2-}$ in the Southern Ocean. As we already have responded above, the main cause of the high contribution of anthropogenic nss-SO$_4^{2-}$ in the Southern Ocean was the use of the single value of MSA/nss-SO$_4^{2-}$ ratio measured at Palmer Station.

2. I do not understand your reasoning on why nssSO4 in the Amundsen Sea was formed by the condensation of DMS oxidation products onto existing particles. Sanchez et al. 2018 identified two sulfate particle types from single particle measurements and showed one, which contained mostly sulfate mass, corresponded to new particle formation in the free troposphere that was then entrained into the boundary layer. The second particle type was less than 50% sulfate and did not correspond to entrainment of particles from the free troposphere indicating it was likely from the condensation of DMS oxidation products onto pre-existing particles. I understand that new particle formation in the free troposphere is out of the scope of this paper, but with the measurements in this manuscript identifying whether nssSO4 is from DMS oxidation products condensed onto existing particles or new particle formation does not seem possible. My original reference to Sanchez et al. 2018 was intended to show that MSA and DMS do not necessarily correlate strongly with nssSO4 because of the possibility of long range transport of nssSO4 to regions

where MSA and DMS are low in concentration (Korhonen et al. 2008; Woodhouse et al. 2010) and physical processes, such as the mixing of DMS and particles across the boundary layer inversion, are necessary for the formation of nssSO4 particles. Therefore, the lack of a correlation of MSA and nssSO4 should not be considered an indicator of anthropogenic influence. At least not without another anthropogenic tracer.

(Response) We are grateful to Referee #1 for Referee #1's comments. We have realized that the description that $nss\text{-}SO_4^{2-}$ in the Amundsen Sea was formed by the condensation of DMS products onto existing particles was our speculation. Besides, we also have realized that the statistically insignificant relationship between $nss\text{-}SO_4^{2-}$ and MSA in the Southern Ocean does not always result from the influence of anthropogenic $nss\text{-}SO_4^{2-}$ because of the possibility of long-range transport of $nss\text{-}SO_4^{2-}$ to the regions where MSA and DMS concentrations are low, as Referee #1 pointed out. Consequently, according to Referee #1's comments, we have revised our manuscript as follows: "Unlike MSA, the mean $nss\text{-}SO_4^{2-}$ concentration in the Amundsen Sea was comparable to that in the Southern Ocean, although the variation trend of $nss\text{-}SO_4^{2-}$ in the Amundsen Sea was similar to that of MSA, suggesting that $nss\text{-}SO_4^{2-}$ was affected by marine source and large-scale transport (Korhonen et al., 2008). Indeed, $nss\text{-}SO_4^{2-}$ showed a strong correlation ($r = 0.98$, $p < 0.01$) with MSA in the Amundsen Sea, whereas no relationship was found between them in the Southern Ocean ($r = 0.51$, $p > 0.05$) (Fig. S3), suggesting that the local emission of DMS is a significant source of $nss\text{-}SO_4^{2-}$ in the Amundsen Sea. It is worth mentioning that $nss\text{-}SO_4^{2-}$ can be formed from the homogeneous nucleation of new particles involving $H_2SO_4$ or from the condensation of gas-phase DMS oxidation products onto existing particles (e.g., Covert et al., 1992; Quinn and Bates, 2011). Recently, Sanchez et al. (2018) identified two types of $SO_4^{2-}$ particles using an Event-Trigger Aerosol-Mass-Spectrometer (ET-AMS), and reported that 63% of $SO_4^{2-}$ was derived from newly formed particles in the free troposphere and 38% of $SO_4^{2-}$ was formed from the condensation of DMS products onto existing particles in the clean marine conditions, revealing the importance of phytoplankton-produced DMS emission for CCN in the Atlantic. In this study, it is hard to distinguish $nss\text{-}SO_4^{2-}$ derived from new particle formation from $nss\text{-}SO_4^{2-}$ formed by the condensation of DMS products onto existing particles because of the limitations related to the method to collect data. However, the significant correlation of $nss\text{-}SO_4^{2-}$ with MSA in the Amundsen Sea suggest that DMS emission from the Amundsen Sea plays a crucial role in the formation of $nss\text{-}SO_4^{2-}$ aerosol particles that can act as CCN. On the other hand, the statistically insignificant relationship between $nss\text{-}SO_4^{2-}$ and MSA in the Southern Ocean could result from the influence of anthropogenic sources in the Southern Ocean. During the cruise, as shown in Fig. 5a, high (> 0.1 µg m⁻³) nitrate ($NO_3^-$) concentrations were observed during the collection of aerosol samples A1 and A13 when air masses originated from the Southern Ocean and thereafter swept over large regions of New Zealand (Fig. S4); however, $NO_3^-$ concentrations in the other samples were quite low (< 0.03 µg m⁻³). Although $NO_3^-$ was observed in the other aerosol samples, we cannot assert that all aerosol samples were influenced by anthropogenic sources without another anthropogenic tracer data set (e.g., [210]Pb) because, in addition to anthropogenic sources, $NO_3^-$ can be produced in clouds (Hegg and Hobbs, 1988) or derived

from a weak oceanic source (Luo et al., 2018). Nevertheless, our results suggest that $nss\text{-}SO_4^{2-}$ in the two samples collected over the Southern Ocean were most likely affected by anthropogenic sources, resulting in the lack of correlation. Another possible explanation for the insignificant relationship between $nss\text{-}SO_4^{2-}$ and MSA is entrainment of nucleated $nss\text{-}SO_4^{2-}$ particles into the marine boundary layer of the Southern Ocean from the free troposphere by turbulent diffusion and large-scale transport (Korhonen et al., 2008; Woodhouse et al., 2010). Sanchez et al. (2018) reported that the lack of correlation between $SO_4^{2-}$ particle and atmospheric DMS (or its oxidation products) could result from the competition for DMS and its oxidation products with the competing sinks of condensation onto existing particles and vertical transport to the free troposphere. The lack of correlation between $nss\text{-}SO_4^{2-}$ and MSA in the Southern Ocean, therefore, could have resulted from the influence of anthropogenic sources and the input of $nss\text{-}SO_4^{2-}$ from the free troposphere. Although our data set is not sufficiently complete to allow a meaningful analysis of this likely explanation, the result for $nss\text{-}SO_4^{2-}$ concentrations from this study would be valuable for filling the data gap, especially for the Amundsen Sea during the austral summer, and be helpful for validation of modeling of sulfur-containing aerosols. (Page 10, line number 8−page 11, line number 7).

Besides, to clarify the context of section 3.3, we have moved the following sentences to the first paragraph of section 3.3: "Surprisingly, our mean $nss\text{-}SO_4^{2-}$ concentration in the Amundsen Sea was about 1.5 and 2.5 times higher than those observed at American Samoa (14.25°S, 170.58°W, $0.41 \pm 0.17$ µg m⁻³, seasonal average (December–February) from 1990–1992, Savoie et al., 1994) and over the South Pacific (8°S–55°S, mean: $0.25 \pm 0.17$ µg m⁻³, range: 0.094–0.62 µg m⁻³, January–March 2009, Jung et al., 2014), respectively. In addition, the mean $nss\text{-}SO_4^{2-}$ concentration in the Amundsen Sea was also a factor of 1.6–4.4 higher than those observed at Palmer Station ($0.24 \pm 0.16$ µg m⁻³, December 1990–March 1991, Savoie et al., 1993), Halley Station ($0.14 \pm 0.017$ µg m⁻³, monthly mean in January from 1991–1993, Legrand and Pasteur, 1998), Neumayer Station ($0.38 \pm 0.13$ µg m⁻³, monthly mean in January from 1983–1995, Minikin et al., 1998), and Dumont D'Urville Station ($0.34 \pm 0.039$ µg m⁻³, monthly mean in January from 1991–1995, Minikin et al., 1998) during the austral summer." (page 9, line number 32−page 10, line number 7).

We also have removed from section 3.3 the following sentence: "Considering the high MSA concentration and DMS flux in the Amundsen Sea as mentioned above in section 3.2, the high $nss\text{-}SO_4^{2-}$ concentration in the Amundsen Sea from this study is likely affected much more by biogenic than anthropogenic sources."

We have added the references (i.e., Hegg and Hobbs, 1988; Covert et al., 1992; Korhonen et al., 2008; Woodhouse et al., 2010; Luo et al., 2018) to the manuscript (page 23, line number 23−24; page 22, line number 14−15; page 24, line number 18−20; page 29, line number 14−16; page 25, line number 3−5).

3. I am not sure what part of my response the authors thought was an opinion. NO3 is produced in clouds

(Hegg and Hobbs 1988) and can have small marine sources (Luo et al 2018). Here are a few publications that show nitrate formation in clouds. Sulfate can also be produced in clouds (Hegg and Hobbs 1988). Therefore, your correlation of NO3 with nssSO4 in the remote southern ocean is much more likely due to cloud processing and not anthropogenic sources. It is not reasonable to use nitrate "solely as an indicator of anthropogenic contribution in this study" if the values are so low that they do not appear to be affected by anthropogenic sources. There are two spikes, A1 and A13, that could possibly be associated with anthropogenic sources. Maybe those cases have back trajectories over continents?

(Response) We apologize to Referee #1 for saying Referee #1's comment as an opinion. We have realized that $NO_3^-$ is not a good tracer for anthropogenic source because of the fact that $NO_3^-$ can be produced in clouds and derived from marine source, as Referee #1 pointed out. As Referee #1 mentioned, we observed the high $NO_3^-$ concentrations during the collection of aerosol samples, A1 and A13 when air masses originated from the Southern Ocean and thereafter swept over large regions of New Zealand; however, $NO_3^-$ concentrations in the other samples were quite low. Consequently, based on Referee #1's comments, we have revised our manuscript as follows: "On the other hand, the statistically insignificant relationship between $nss\text{-}SO_4^{2-}$ and MSA in the Southern Ocean could result from the influence of anthropogenic sources in the Southern Ocean. During the cruise, as shown in Fig. 5a, high ($> 0.1$ µg m$^{-3}$) nitrate ($NO_3^-$) concentrations were observed during the collection of aerosol samples A1 and A13 when air masses originated from the Southern Ocean and thereafter swept over large regions of New Zealand (Fig. S4); however, $NO_3^-$ concentrations in the other samples were quite low ($< 0.03$ µg m$^{-3}$). Although $NO_3^-$ was observed in the other aerosol samples, we cannot assert that all aerosol samples were influenced by anthropogenic sources without another anthropogenic tracer data set (e.g., $^{210}$Pb) because, in addition to anthropogenic sources, $NO_3^-$ can be produced in clouds (Hegg and Hobbs, 1988) or derived from a weak oceanic source (Luo et al., 2018). Nevertheless, our results suggest that $nss\text{-}SO_4^{2-}$ in the two samples collected over the Southern Ocean were most likely affected by anthropogenic sources, resulting in the lack of correlation." (page 10, line number 23−31).

4. I did not misunderstand your manuscript. I understood 39%- 139% is biologically derived contribution of nss-sulfate, but this number also indicates the anthropogenic contribution was up to 61% (when the biological contribution is only 39%) and therefore, at times, more than half of the nssSO4 is anthroprogenic in the Amundsen Sea. And even higher in the southern ocean (76%). Again stronger evidence is needed to make such a claim. This may simply not be possible with the measurements made in this experiment. An anthropogenic tracer that is also formed naturally in the remote southern ocean is necessary for such an analysis.

(Response) We apologize to Referee #1 for our misunderstanding of Referee #1's comments. As we already responded above, the main cause of the high contributions of anthropogenic $nss\text{-}SO_4^{2-}$ in the Southern Ocean and the Amundsen Sea was the use of the single value of $MSA/nss\text{-}SO_4^{2-}$ ratio measured

at Palmer Station. Furthermore, because of the limitations of method using MSA/nss-$SO_4^{2-}$ and the absence of another anthropogenic tracer, such as [210]Pb, in this study, we have removed the contribution of anthropogenic nss-$SO_4^{2-}$ from section 3.4 and revised our manuscript according to Referee #1 comments (page 11, line number 20−page 12, line number 15).

Reviewers 3 and 4 point out the discrepancies of WIOC and WSOC and wind speed. Figure S6 (should be in manuscript instead of supplement) clearly shows that the measured atmospheric WIOC and WSOC are not associated with the air sea flux of primary sea spray particles. This is not surprising for WSOC since it is produced in the atmosphere by secondary processes (good correlation with MSA in figure 8C) but why not WIOC? Similarly, in Figure 7, the WIOC and WSOC to Na ratios have similar relationships to wind speed and DOC. Why would that be true if WSOC is a secondary aerosol?

(Response) We are grateful to Referee #5 for Referee #5's comments. According to Referee #5's comment, we have added Figure S6 as Figure 8 to the manuscript (page 38). Thus, we also have changed the numbers of Figures 8−10 to Figures 9−11 (pages 39−41; page 17, line number 6; page 17, line number 7; page 17, line number 29; page 18, line number 3; page 18, line number 7; page 18, line number 12; page 18, line number 17).

Referee #5 asked why the measured atmospheric WIOC was not associated with the air-sea flux of primary sea spray particles. In this study, we investigated relationships of WIOC and WSOC concentrations with mean wind speed and $Na^+$ concentration in the fine modes since both WIOC and WSOC were primarily associated with the fine mode particles (Figs. 6a and 6b). As shown in Fig. 8a, no relationship was found between the submicron WIOC and mean wind speed. In addition, the submicron WIOC showed no statistically significant relationships with submicron $Na^+$ over the Southern Ocean and the Amundsen Sea (Fig. 9a), although WIOC was highly enriched in the fine mode sea spray particles (Fig. 7b) (page 16, line number 19−24). As mentioned in section 3.6, under higher wind speed conditions, the enrichment of organic matter in sea spray aerosols can be diluted by larger sea-salt particles. Furthermore, the increase in sea-salt particle flux under higher wind speed conditions shifts the sea spray aerosol size distribution towards larger sizes and accelerates their dry deposition and gravitational settling from the atmosphere (de Leeuw et al., 2011). Thus, theses insignificant relationships between WIOC and $Na^+$ in the fine modes could result from the differences in local wind speeds and local biological activities, such as sea surface DOC concentration, because wind speed, a key factor determining sea spray aerosols, controls the local flux rather than local concentration of marine particles (Monahan and O'Muircheartaigh, 1986) (page 16, line number 25−31).

In section 3.6, we have discussed the inverse relationships of WSOC/$Na^+$ and WIOC/$Na^+$ ratios with mean wind speed and the positive relationships of those ratios with sea surface DOC concentration in the Amundsen Sea. During the sampling period, both WSOC/$Na^+$ and WIOC/$Na^+$ ratios showed inverse relationships with mean wind speed (r = –0.82, p < 0.01 and r = –0.76, p < 0.01, respectively) (Figs. 7c and 7d). The highest WSOC/$Na^+$ and WIOC/$Na^+$ ratios were observed in the aerosol sample (i.e., A5)

[revised manuscript text omitted]

We also have added the references (i.e., Quinn et al., 2015; Liss and Duce, 1997; Ciuraru et al., 2015; Mungall et al., 2017; van Pinxteren et al., 2017; de Leeuw et al., 2011) to the manuscript (page 26, line number 32−33; page 24, line number 34; page 22, line number 6−7; page 26, line number 3−6; page 29, line number 4−6; page 22, line number 19−20).

The authors try to explain the OC/Na vs wind speed dependence on ocean mixing. How deep was the ocean mixed layer? If it was more than 1 m, wind/wave mixing should have little effect on surface OC content.

(Response) We thank Referee #5 for Referee #5's comment. We have realized that ocean mixing was not a main cause of the inverse relationships of WSOC/Na$^+$ and WIOC/Na$^+$ ratios with mean wind speed. As we already have responded above, the upper water column was already well mixed in the austral summer (average surface mixed layer depths: 25 ± 19 m, Yager et al., 2016) (page 15, line number 13−15). Previous studies (Quinn et al., 2015 and references therein) reported that the production of particles highly enriched in organic material derives fragmentation of the film cap from which much of the seawater has drained prior to bursting, leaving behind a film that is highly enriched surfactant material, resulting in an enrichment in OC relative to seawater in the freshly emitted sea spray aerosol. In addition, BVOCs can be produced photochemically from the biologically productive sea surface water and emitted to the atmosphere following the excitation of dissolved organic matter found in the presence of surfactant films, leading to the production of secondary organic aerosol (Liss and Duce, 1997; Ciuraru et al., 2015; Mungall et al., 2017). Our results, therefore, suggest that accumulation of organic matter at the sea surface in biologically productive seawater under low wind speed conditions makes a favorable condition for transferring the organic matter to the atmosphere through the bubble bursting process and the condensation of BVOCs on pre-existing sea spray aerosols, resulting in formation of sea spray aerosols that are considerably enriched in the organic matter (O'Dowd et al., 2004; van Pinxteren et al., 2017). However, the WSOC/Na$^+$ and WIOC/Na$^+$ ratios drastically decreased with increasing mean wind speed, suggesting that the WSOC/Na$^+$ and WIOC/Na$^+$ ratios were diluted by wind speed-dependent sea-salt emissions. Under higher wind speed conditions, larger sea-salt particles could be transported upward and change the size distribution of sea spray aerosols, resulting in the higher enrichment of sea-salt in sea spray aerosol, with very little contribution from organics (Gantt et al., 2011; de Leeuw et al., 2011) (page 14, line number 21−page 15, line number 1).

We also have removed the following sentence from section 4: "when the wave breaking thoroughly mixes the organic-enriched sea surface layer with the underlying relatively organic-poor water." (page 19, line number 16−18).

Even though I was not one of the original reviewers, I do not feel that the authors have addressed the discrepancies pointed out by the reviews in the revised manuscript. Also, the explanations for the OC/Na vs wind speed dependence are not convincing, nor is the discrepancy between why secondary components would be wind speed dependent.

(Response) We thank Referee #6 for Referee #6's comment. As Referee #6 pointed out, our previous responses were not enough to explain the discrepancies pointed out by the Referees. We have revised our manuscript according to Referees' comments. Please see our responses to Referee #1 and Referee #5.

Referee #6 pointed out that the explanation for the WSOC/Na$^+$ and WIOC/Na$^+$ ratios vs. wind speed dependence are not convincing and asked why secondary components would be wind speed dependent. During the sampling period, both WSOC/Na$^+$ and WIOC/Na$^+$ ratios showed inverse relationships with mean wind speed (r = –0.82, p < 0.01 and r = –0.76, p < 0.01, respectively) (Figs. 7c and 7d). The highest WSOC/Na$^+$ and WIOC/Na$^+$ ratios were observed in the aerosol sample (i.e., A5) collected in biologically active region (i.e., in situ Chl-a concentration was highest (6.9 mg m$^{-3}$)) under calm conditions (i.e., mean wind speed was lowest (4.4 m s$^{-1}$)). Previous studies (Quinn et al., 2015 and references therein) reported that the production of particles highly enriched in organic material derives fragmentation of the film cap from which much of the seawater has drained prior to bursting, leaving behind a film that is highly enriched surfactant material, resulting in an enrichment in OC relative to seawater in the freshly emitted sea spray aerosol. In addition, BVOCs can be produced photochemically from the biologically productive sea surface water and emitted to the atmosphere following the excitation of dissolved organic matter found in the presence of surfactant films, leading to the production of secondary organic aerosol (Liss and Duce, 1997; Ciuraru et al., 2015; Mungall et al., 2017). Our results, therefore, suggest that accumulation of organic matter at the sea surface in biologically productive seawater under low wind speed conditions makes a favorable condition for transferring the organic matter to the atmosphere through the bubble bursting process and the condensation of BVOCs on pre-existing sea spray aerosols, resulting in formation of sea spray aerosols that are considerably enriched in the organic matter (O'Dowd et al., 2004; van Pinxteren et al., 2017). However, the WSOC/Na$^+$ and WIOC/Na$^+$ ratios drastically decreased with increasing mean wind speed, suggesting that the WSOC/Na$^+$ and WIOC/Na$^+$ ratios were diluted by wind speed-dependent sea-salt emissions. Under higher wind speed conditions, larger sea-salt particles could be transported upward and change the size distribution of sea spray aerosols, resulting in the higher enrichment of sea-salt in sea spray aerosol, with very little contribution from organics (Gantt et al., 2011; de Leeuw et al., 2011). Furthermore, the WSOC/Na$^+$ and WIOC/Na$^+$ ratios observed in the biologically active region (i.e., higher in situ Chl-a concentrations) did not always show higher values. Therefore, these results from this study suggest that the wind speed is a significant factor influencing the organic mass fractions of sea spray aerosols in our study region and that Chl-a is not wholly adequate as a proxy for the biological activity

responsible for the organic material comprising the aerosol (page 14, line number 18-page 15, line number 5).

---

## Author Response (AR3)

**Responses to Co-Editor' comments**

12/11/2019

Journal: Atmospheric Chemistry and Physics

Title: Characteristics of methanesulfonic acid, non-sea-salt sulfate and organic carbon aerosols over the Amundsen Sea, Antarctica

Authors: Jinyoung Jung, Sang-Bum Hong, Meilian Chen, Jin Hur, Liping Jiao, Youngju Lee, Keyhong Park, Doshik Hahm, Jung-Ok Choi, Eun Jin Yang, Jisoo Park, Tae-Wan Kim, and SangHoon Lee

Manuscript number: acp-2019-133

Manuscript type: Research Article

Note: Co-Editor's comments are high lightened in **blue**, while our responses to reviewers are in **black**. The revisions in the manuscript was highlighted to make easily readable for the Co-Editor.

Prof. Lynn M. Russell

Co-Editor

Atmospheric Chemistry and Physics

USCD, Scripps Institution of Oceanography

United States

Dear Professor Russell,

We have changed the title of our manuscript to "Characteristics of methanesulfonic acid, non-sea-salt sulfate and organic carbon aerosols over the Amundsen Sea, Antarctica" according to Co-Editor's comment. We have attached an electronic copy of manuscript file ready to go to press entitled "Characteristics of methanesulfonic acid, non-sea-salt sulfate and organic carbon aerosols over the Amundsen Sea, Antarctica" by Jinyoung Jung, Sang-Bum Hong, Meilian Chen, Jin Hur, Liping Jiao, Youngju Lee, Keyhong Park, Doshik Hahm, Jung-Ok Choi, Eun Jin Yang, Jisoo Park, Tae-Wan Kim, and SangHoon Lee for publication in Atmospheric Chemistry and Physics (acp-2019-133). We have modified our text based on the Co-Editor's comments. We appreciated that the comments from Co-Editor improved our manuscript a lot. We believe that the comments from Co-Editor were clearly responded in our modified manuscript.

We are looking forward to hearing about your decision. Thank you for your consideration.

Sincerely yours,

Jinyoung Jung

**1. p.11 line 26 to p.12 line 15: omit this section. Extrapolating from summer to annual is not justified by the measurements or the literature. It detracts from the otherwise solid nature of the paper.**

(Response) We thank Co-Editor for Co-Editor's comment. As Co-Editor suggested, we have removed the sentences Co-Editor mentioned in section 3.4 (the sentences were written on page 11, line number 26−page 12, line number 15 in the previous manuscript). We also have removed from the section 3.4 the following sentences: "While nss-SO$_4$ can have multiple sources, including DMS oxidation, volcanic and industrial sulfur emissions, MSA is formed exclusively from DMS (Gondwe et al., 2003). Thus MSA was proposed as a useful tracer to distinguish between marine biogenic and anthropogenic nss-SO$_4$ (Legrand and Pasteur, 1998). Considerable efforts have been devoted to investigating the contribution of biogenically-derived atmospheric sulfur species over the various geographical locations by using the MSA/nss-SO$_4$ ratio observed in aerosols, which shows the spatial (higher ratios at high latitudes) and seasonal variability (summer maxima and winter minima) (e.g., Savoie and Prospero, 1989; Prospero et al., 1991; Bates et al., 1992; Arimoto et al., 2001; Savoie et al., 2002; Yang et al., 2009; Jung et al., 2014; Legrand et al., 2017). Mungall et al. (2018), however, pointed out that the MSA/nss-SO$_4$ ratio could have limitation that may preclude its use in quantitatively unravelling the chemical and biological processes at play in the marine boundary layer due to the conversion of MSA to nss-SO$_4$ by OH radical in aerosol particles (the order of days to weeks), although it remains useful as a qualitative indicator of marine biological influence." (these sentences were written on page 11, line number 9−19 in the previous manuscript). Besides, we have changed the title of section 3.4 "Contribution of biogenically-derived nss-SO$_4^{2-}$ to total nss-SO$_4^{2-}$" to "MSA/nss-SO$_4^{2-}$ ratios over the Southern Ocean and the Amundsen Sea" because the contribution of biogenically-derived nss-SO$_4^{2-}$ is not discussed in this section (page 10, line number 24). We also moved the following sentence to the first paragraph of section 3.4 to give information to the readers (page 10, line number 25−28): "There have been several field studies investigating the MSA/nss-SO$_4^{2-}$ ratio in the Southern Ocean during the austral summer (range: 0.32–0.53, the South Pacific (40°S−45°S), Berresheim et al., 1990; range: 0.12–0.24, Cape Grim (40°41′S , 144°41′E), Ayers et al., 1991; range: 0.17–0.32, the South Pacific (30°S−60°S), Bates et al., 1992; range: 0.096–0.49, the South Pacific (40°S−56°S), Jung et al., 2014)." (this sentence was written on page 11, line number 29−32 in the previous manuscript).

**2. As reviewers noted, the constant value of MSA/SO4 is not sufficient to show biogenic sulfate fraction for the whole year, so please remove all bio/anthro attribution from article (including title, abstract, conclusions). The title should say what was measured: MSA, sulfate, organic mass.**

(Response) We thank Co-Editor for Co-Editor's comment. As Co-Editor suggested, we have changed the title of our manuscript from "Characteristics of biogenically-derived aerosols over the Amundsen Sea, Antarctica" to "Characteristics of methanesulfonic acid, non-sea-salt sulfate and organic carbon aerosols over the Amundsen Sea, Antarctica" (page 1, line number 1−2). Besides, we have removed all bio/anthropogenic attribution related to nss-SO$_4^{2-}$ according to Co-Editor's comment as follows:

1. We have revised "To investigate the influence of marine biological activity on aerosols," to "To investigate characteristics of methanesulfonic acid (MSA), non-sea-salt sulfate (nss-SO$_4^{2-}$) and organic carbon (OC) aerosols," (page 1, line number 12−13). Also, we have revised "atmospheric methanesulfonic acid (MSA)" to "atmospheric MSA" (page 1, line number 15) and "Unlike MSA, mean non-sea-salt sulfate (nss-SO$_4^{2-}$)" to "Unlike MSA, mean nss-SO$_4^{2-}$" (page 1, line number 17).

2. We have revised "Unlike MSA, mean nss-SO$_4^{2-}$ concentration in the Amundsen Sea was comparable to that in the Southern Ocean, suggesting significant influences of marine biological activity on atmospheric sulfur species in the Amundsen Sea." to "Unlike MSA, mean nss-SO$_4^{2-}$ concentration in the Amundsen Sea was comparable to that in the Southern Ocean." (page 1, line number 17−18).

3. We have removed "The results from this study provide significant new observational data on biogenically-derived sulfur and organic carbon species in the Amundsen Sea." from the abstract (this sentence was written on page 2, line number 2−4 in the previous manuscript).

4. We have revised "To understand the influence of marine biological activities on atmospheric biogenically-derived aerosols in the Amundsen Sea," to "To understand the influence of marine biological activities on atmospheric marine aerosols in the Amundsen Sea," (page 3, line number 22).

5. We have removed "(3) estimate the contribution of biogenic nss-$SO_4^{2-}$ to total nss-$SO_4^{2-}$" from the objectives of this study (page 3, line number 26−28). We also have removed "The results from this study provide quantitative insight into ambient levels of biogenically-derived sulfur and OC species in the marine boundary layer in the Amundsen Sea." from the introduction. (this sentence was written on page 4, line number 1−3 in the previous manuscript).

6. We have removed "biogenically-derived" from the following sentence in conclusion: "Characteristics of biogenically-derived atmospheric sulfur (i.e., MSA and nss-$SO_4^{2-}$) and OC (i.e., WSOC and WIOC) species in marine aerosols, " (page 15, line number 5−6).

7. We have removed the following sentence from the conclusion: "Furthermore, biogenically-derived nss-$SO_4^{2-}$ dominated the atmospheric budget of nss-$SO_4^{2-}$ in the Amundsen Sea, contributing ~86% to total nss-$SO_4^{2-}$ (this sentence was written on page 19, line number 6−7 in the previous manuscript). We also revised ", suggesting significant influences of marine biological activities on atmospheric sulfur species" to ", suggesting significant influences of marine biological activities on atmospheric MSA." (page 15, line number 8) and "There results were attributed to…." to "The higher MSA concentration was attributed to…" (page 15, line number 8−9).

8. We have revised "biogenically-derived aerosols" to "marine aerosols" (page 3, line number 5).

**3. Since the assertion that the nssSO4 is biogenic has little supporting evidence, it is weak and should simply be removed. A stronger case could be made with tracers such as BC and back trajectories (or isotopes) but in lieu of that please just remove that term as it is an unsupported assertion (title, p.19 line 2,6, 21, etc.).**

(Response) We thank Co-Editor for Co-Editor's comment. As we have already responded to Co-Editor's comments #1 and #2, we have changed the title of our manuscript to "Characteristics of methanesulfonic acid, non-sea-salt sulfate and organic carbon aerosols over the Amundsen Sea, Antarctica" (page 1, line number 1−2). We have already changed or removed the unsupported assertions Co-Editor mentioned. Please see our responses to Co-Editor's comments #1 and #2. Besides, we have revised "biogenically-derived OC species" to "OC species" (page 15, line number 18−19).

**4. No figure in this paper shows a correlation between sulfate, msa, or organic aerosol mass with chl or biomass (no relationship is evident in 7c,d), so remove all discussion of biological coupling: p.19 line 16-18; p.15 line 6 to p.16 line 8.**

(Response) We thank Co-Editor for Co-Editor's comment. As Co-Editor suggested, we have removed all discussion of biological coupling, which was written on page 19, line number 16−18, and on page 15, line number 6−page 16, line number 8 in the previous manuscript (please see the second paragraph in the conclusions on page 15 and the first paragraph of section 3.6 on page 12). We also removed "In addition, the submicron WIOC concentration was quite related to the relative biomass of *P. antarctica*, suggesting that extracellular polysaccharide mucus produced by *P. antarctica* was a significant factor affecting atmospheric WIOC concentration in the Amundsen Sea." from the abstract. (this sentence was written on page 1, line number 25−28 in the previous manuscript). Besides, we have removed Figures 7e, 7f from the manuscript (page 31), and S5 from the Supplement.

**5. This statement "A good correlation was found between the relative biomass of P. antarctica and the submicron WIOC concentration." In the conclusions does not seem supported by a figure; remove. Fig.11 shows relationships to fluorescence intensity not mass concentration, and the latter two things are not equivalent.**

(Response) We thank Co-Editor for Co-Editor's comment. As Co-Editor suggested, we have removed "A good correlation was found between the relative biomass of *P. antarctica* and the submicron WIOC concentration, suggesting that extracellular polysaccharide mucus generated by *P. antarctica* is a significant source of atmospheric WIOC in the Amundsen Sea." from the conclusions (page 15, the third paragraph). Besides, we have revised "Moreover, the fluorescence properties of WSOC revealed that the majority of WSOC (i.e., protein-like components) was most likely derived from BVOCs as a result of biological

processes of diatoms, by showing the significant positive relationship between the relative biomass of diatoms and protein-like component in marine aerosols in the Amundsen Sea." to "Moreover, the fluorescence properties of WSOC revealed that protein-like components are most likely produced as a result of biological processes of diatoms." (page 15, line number 19−20). We have also revised "These results suggest that protein-like component is most likely produced as a result of biological processes of diatoms, which play a crucial role in forming the submicron WSOC observed over the Southern Ocean and the Amundsen Sea, and that phytoplankton community structure is a significant factor affecting atmospheric organic carbon species." to "These results suggest that protein-like component is most likely produced as a result of biological processes of diatoms in the Amundsen Sea (page 1, line number 28−30).

**6. Section 3.8 does discuss the measured biology and provides interesting context, but a relationship beyond that to the aerosol should not be discussed unless it is explicitly shown.**

(Response) We thank Co-Editor for Co-Editor's comment. As Co-Editor suggested, we have revised "Consequently, our results suggest that protein-like components are most likely produced as a result of biological processes of diatoms, which play a key role in forming the submicron WSOC observed over the Southern Ocean and the Amundsen Sea, and that phytoplankton community structure is a significant factor affecting atmospheric OC species since the submicron WIOC was quite related to the relative biomass of *P. antarctica* (see section 3.6)." to "Consequently, our results suggest that protein-like components are most likely produced as a result of biological processes of diatoms." (page 14, line number 31−32). We also have revised "The high BIX values also supported that the majority of WSOC was derived from biological processes." to "The high BIX values also supported that the fluorescence properties of WSOC were influenced by marine biological activities." (page 15, line number 3).

**7. The term enrichment is used problematically in many places, e.g. p.14 lines 1,4, where it is not clear what is enriched with respect to what. Compared to seawater? Other sizes? Clarify or remove this word.**

(Response) We thank Co-Editor for Co-Editor's comment. As Co-Editor suggested, we have clarified the meaning of "enriched or enrichment" by revising it to other expressions as follows:

1. We have revised "WSOC and WIOC were highly enriched in the submicron sea spray particles, " to "WSOC/$Na^+$ and WIOC/$Na^+$ ratios in the fine mode aerosol particles were higher, " (page 1, line number 23−24).

2. We have revised "About ~80% (median values for all data) of MSA was enriched in the fine mode aerosols." to "About ~80% (median values for all data) of MSA existed in the fine mode aerosols." (page 7, line number 17−18).

3. We have revised "Both WSOC and WIOC mainly existed in fine mode particles, and the enrichment (i.e., the percentage of WSOC or WIOC present in fine aerosol particles) of WSOC and WIOC in fine mode particles were ~93% and ~74%, respectively (median value for all data)." to "Both WSOC and WIOC mainly existed in fine mode particles, and the percentages of WSOC and WIOC present in fine aerosol particles were ~93% and ~74%, respectively (median value for all data)." (page 11, line number 13−14).

4. We have revised "During the cruise, ~76% of $Na^+$, a tracer of sea spray, was enriched in the coarse mode particle (Fig. 7a)." To "During the cruise, ~76% of $Na^+$, a tracer of sea spray, was associated with the coarse mode particle (Fig. 7a)." (page 12, line number 17).

5. We have revised "WSOC and WIOC were highly enriched in the fine mode sea spray particles," to "WSOC/$Na^+$ and WIOC/$Na^+$ ratios in the fine mode aerosol particles were higher than those in the coarse mode aerosol particles," (page 12, line number 20−21).

6. We have revised "however, WIOC was much more enriched in the fine mode sea spray particles than WSOC," to "however, WIOC/$Na^+$ ratio in the fine mode aerosol particles was much higher than WSOC/$Na^+$," (page 12, line number 27).

7. We have revised "the higher enrichment of OC in sea spray aerosols," to "the higher OC/Na$^+$ ratios" (page 12, line number 29−30).

8. We have revised "although WIOC was highly enriched in the fine mode sea spray particles (Fig. 7b)." to "although WIOC/Na$^+$ ratio in the fine mode aerosol particles was much higher (Fig. 7b)." (page 13, line number 3).

9. We have revised "the high enrichment of WIOC in the fine mode sea spray particles" to "the high WIOC/Na$^+$ ratio in the fine mode aerosol particles" (page 13, line number 12).

10. We have revised "However, WSOC and WIOC were highly enriched in the submicron sea spray particles," to "However, the higher WSOC/Na$^+$ and WIOC/Na$^+$ ratios were observed in the submicron aerosol particles," (page 15, line number 14−15).

**8. Sections 3.6 and 3.7 are confusing and repetitive, and they should be combined and shortened. Fig. 8 shows neither OC depends on wind speed, so please remove all discussion of that as it is not relevant. Fig. 7c,d shows a negative correlation of OC/Na to wind speed, but I expect that is entirely because Na is positively correlated to wind speed and OC has no dependence. If so, remove this discussion and delete p. 19 line 15-18.**

(Response) We thank Co-Editor for Co-Editor's comment. As Co-Editor suggested, we have combined sections 3.6 and 3.7. First of all, we have changed the title of section 3.6 to "WIOC/Na$^+$ and WSOC/Na$^+$ ratios and relationships of WIOC and WSOC with Na$^+$ over the Southern Ocean and the Amundsen Sea" (page 12, line number 9−10). We also have removed all discussion related to relationships of WIOC/Na+ (Fig. 7c), WSOC/Na+ (Fig. 7d), WIOC (Fig. 8a) and WSOC concentrations (Fig. 8b) with wind speed (please see section 3.6). Besides, we have removed "We found significant inverse relationships between WSOC/Na·, WIOC/Na· ratios and the mean wind speed, suggesting that the wind speed affected the organic mass fractions of sea spray aerosols in our study region." from the conclusions (this sentence was written on page 19, line number 14−15 in the previous manuscript). We also have removed Figures 7c, 7d, 8a, and 8b from the manuscript. In addition, we have changed section number (i.e., section 3.8 to section 3.7) and Figure numbers (e.g., Figure 9 to Figure 8).

**Other minor comments: p.2 Line 23: suggest Sanchez et al. 2018 PNAS is more relevant.**

(Response) We thank Co-Editor for Co-Editor's comment. As Co-Editor suggested, we have revised "Lana et al., 2012" to "Sanchez et al., 2018" (page 2, line number 18−19).

---

## Author Response (AR4)

**Responses to Referees' comments**

29/02/2020

Journal: Atmospheric Chemistry and Physics

Title: Characteristics of methanesulfonic acid, non-sea-salt sulfate and organic carbon aerosols over the Amundsen Sea, Antarctica

Authors: Jinyoung Jung, Sang-Bum Hong, Meilian Chen, Jin Hur, Liping Jiao, Youngju Lee, Keyhong Park, Doshik Hahm, Jung-Ok Choi, Eun Jin Yang, Jisoo Park, Tae-Wan Kim, and SangHoon Lee

Manuscript number: acp-2019-133

Manuscript type: Research Article

Note: Referee's comments are high lightened in **blue**, while our responses to reviewers are in **black**. The revisions in the manuscript was highlighted to make easily readable for the Co-Editor and the Referees.

Prof. Lynn M. Russell

Co-Editor

Atmospheric Chemistry and Physics

USCD, Scripps Institution of Oceanography

United States

Dear Professor Russell,

We have attached an electronic copy of manuscript file ready to go to press entitled "Characteristics of methanesulfonic acid, non-sea-salt sulfate and organic carbon aerosols over the Amundsen Sea, Antarctica" by Jinyoung Jung, Sang-Bum Hong, Meilian Chen, Jin Hur, Liping Jiao, Youngju Lee, Keyhong Park, Doshik Hahm, Jung-Ok Choi, Eun Jin Yang, Jisoo Park, Tae-Wan Kim, and SangHoon Lee for publication in Atmospheric Chemistry and Physics (acp-2019-133). We have modified our text based on the Referees' comments. We appreciated that the comments from Referees improved our manuscript a lot. We believe that the comments from Referees were clearly responded in our modified manuscript.

We are looking forward to hearing about your decision. Thank you for your consideration.

Sincerely yours,

Jinyoung Jung

**Anonymous Referee #1:**

The authors have improved the quality of the manuscript by focusing more on their results that are supported by evidence. Below are a couple very minor comments. I suggest the manuscript be published after minor revisions.

Page 10 line 10 - "without another anthropogenic tracer data set (e.g., 210Pb)" A citation showing this as an anthropogenic tracer is necessary here.

Page 10 line 12 "Nevertheless, our results suggest that nss-SO$_4^{2-}$ in the two samples collected over the Southern Ocean were most likely affected by anthropogenic sources, resulting in the lack of correlation." I suggest changing "were most likely" to "may be" due to the lack of evidence. Also since you mention shortly after this sentences that atmospheric processes could play a role.

(Response) We thank Referee #1 for Referee #1's comments. The comments from Referee #1 were related to that from Referee #7, who strongly recommended to remove the discussion on aerosol NO$_3^-$ and the influence of anthropogenic sources from the manuscript. According to Referee #7's comment, we have removed from section 3.3 the following sentences: "the statistically insignificant relationship between nss-SO$_4^{2-}$ and MSA in the Southern Ocean could result from the influence of anthropogenic sources in the Southern Ocean. During the cruise, as shown in Fig. 5a, high (> 0.1 µg m$^{-3}$) nitrate (NO$_3^-$) concentrations were observed during the collection of aerosol samples A1 and A13 when air masses originated from the Southern Ocean and thereafter swept over large regions of New Zealand (Fig. S4); however, NO$_3^-$ concentrations in the other samples were quite low (< 0.03 µg m$^{-3}$). Although NO$_3^-$ was observed in the other aerosol samples, we cannot assert that all aerosol samples were influenced by anthropogenic sources without another anthropogenic tracer data set (e.g., $^{210}$Pb) because, in addition to anthropogenic sources, NO$_3^-$ can be produced in clouds (Hegg and Hobbs, 1988) or derived from a weak oceanic source (Luo et al., 2018). Nevertheless, our results suggest that nss-SO$_4^{2-}$ in the two samples collected over the Southern Ocean were most likely affected by anthropogenic sources, resulting in the lack of correlation." (these sentences were written on page 10, line number 5−13 in the previous manuscript).

**Anonymous Referee #7:**

All over the manuscript, the key scientific terms are not well defined. MSA is present in both the gas and the condensed forms in the atmosphere. As it is appeared that the author only present the condensed form, a clear notation such as MSA (s) would prevent a such confusion. In addition, the authors should make it clear that the presented DMS flux is not directly observed rather calculated assessments from wind and dissolved DMS observations. Furthermore, a quantitive assessment on uncertainty associated with the flux calculation should be discussed.

(Response) We thank Referee #7 for Referee #7's comments. We have differentiated between gaseous MSA (MSA$_{(g)}$) and particulate MSA (MSA$_{(p)}$) according to Referee #7's comment. To differentiate between MSA$_{(g)}$ and MSA$_{(p)}$, we have revised the sentences in Introduction as follows: "After emission to the atmosphere, DMS is oxidized by the hydroxyl (OH), nitrate (NO$_3$), and bromine oxide (BrO) radicals to form either gaseous methanesulfonic acid (MSA$_{(g)}$) or sulfur dioxide (SO$_2$). While MSA$_{(g)}$ rapidly condenses onto existing particles (forming particulate MSA, MSA$_{(p)}$), SO$_2$ is further oxidized to nss-SO$_4^{2-}$" (page 2, line number 12−14). Besides, we have revised "MSA" to "MSA$_{(p)}$" in the manuscript, including Abstract, Introduction, Results and discussion, Conclusions and Figures.

Referee #7's pointed out that we should make it clear that the presented DMS fluxes in this study were not directly observed, but calculated assessments. We agree to the Referee #7's comment. Referee #7 also

pointed out that a quantitative assessment on uncertainty associated with the flux calculation should be discussed. However, we have used the DMS flux data reported by Kim et al. (2017), who calculated sea–air DMS fluxes using sea surface DMS concentrations and shipboard wind speed data monitored during our cruise. Besides, each DMS flux in Figs. 4(a) and 4(b) represents its mean value and standard deviation for each aerosol sampling time. Nevertheless, we have estimated the uncertainty in DMS flux in the Amundsen Sea region to be about 25% using four different gas transfer velocity (k) values, which have been discussed in section 3.2 (page 8, line number 30−page 9, line number 4). Consequently, we have revised our manuscript as follows: "To investigate the relationship between atmospheric MSA$_{(p)}$ and DMS flux, we used the DMS flux data reported by Kim et al. (2017), who calculated sea–air DMS fluxes using sea surface DMS concentrations and shipboard wind speed data monitored during our cruise. Details of the measurement of sea surface DMS concentration and the sea–air DMS flux calculation are given by Kim et al. (2017)." (page 8, line number 25−28). In addition, we have revised the caption of Figure 4 as follows: "Figure 4. Dimethylsulfide (DMS) flux (reported by Kim et al., 2017) (a) against aerosol sample I.D. and the correlation between MSA$_{(p)}$ concentration and DMS flux (b). The DMS fluxes were calculated using sea surface DMS concentrations and shipboard wind speed data monitored during the cruise. Details of the measurement of sea surface DMS concentration and DMS flux calculation are given by Kim et al. (2017). Each DMS flux in both panels (a) and (b) represents its mean value and standard deviation for each aerosol sampling time. The gray hatched areas in panel (a) denote that no DMS measurement was conducted. Pink solid circle line in panel (a) indicates the mean and standard deviation of wind speeds for each aerosol sampling time. Double arrows in panel (a) show the sampling locations of aerosol samples. In panel (b), the red line indicates the correlation between MSA$_{(p)}$ concentration and DMS flux." (page 28).

Discussion on the controlling factor of MSA (s) is mainly dependent upon the DMS flux. It seems that the discussion assumes the calculated DMS flux is directly related with DMS concentration in the atmosphere and subsequently MSA (s) should be linearly correlates with the DMS flux. However, I am not entirely convinced by the underlying assumption. The material flux is a determinant for the atmospheric presence of the given substance but not necessarily for the only determinant. As the authors described, DMS over the Southern Ocean has a fairly lengthy lifetime (a few days), so transport may also play an important role to determine the presence of DMS. The subsequent gas phase oxidation process of DMS is also quite complicated, which likely results in a non-linear relationship between DMS and MSA (g). Therefore, the relationship between DMS and MSA (s) cannot be contained in a linear relationship. Furthermore, it comes to me as an insufficient argument that MSA (s) was observed in the higher level over the Amundsen Sea. According to Figure 4a), the statistics is mostly driven by one data point (A9), otherwise, the MSA (s) over the Amundsen Sea appears around the same level to the other part of the Southern Ocean.

(Response) We thank Referee #7 for Referee #7's comments. We have realized that the underlying assumption was not reasonable as Referee #7 pointed out. According to Referee #7's comments, we have revised our manuscript as follows: "The DMS flux (reported by Kim et al., 2017) averaged for the duration of each aerosol sampling showed a somewhat similar variation trend to that of atmospheric MSA$_{(p)}$ concentration (Fig. 4a), but no correlation was found between atmospheric MSA$_{(p)}$ and DMS flux (r = 0.18, p > 0.05, Fig. 4b). DMS fluxes typically rely on gas transfer velocity (k), which is frequently parameterized as a function of wind speed (Wanninkhof, 2014). Measurement and parameterization of the gas transfer velocity are more challenging and subject to greater uncertainty, particularly at high wind speeds (Smith et al., 2018). Wanninkhof et al. (2014) reported that there is a considerable uncertainty in k, especially, under the strong wind condition. About 20% uncertainty was estimated at a global mean wind speed (7.3 m s$^{-1}$). When we applied four different k values (W92, W99, N00, W14), the uncertainty in DMS flux in the Amundsen Sea region was about 25%. As the gas transfer velocity increases with increasing wind speed, DMS flux can be overestimated especially, in higher latitudes where DMS is commonly found at high concentrations in surface water, and where both low temperatures and high winds are typical (McGillis et al., 2000). In addition to the uncertainty in DMS flux, the insignificant relationship between atmospheric MSA$_{(p)}$ and DMS flux could result from various complexities in the rate of oxidation of DMS to form

atmospheric MSA$_{(p)}$ and long-range transport of atmospheric MSA$_{(p)}$ from biogenically active region, given the lifetime of DMS is approximately 1–2 days (Kloster et al., 2006; Read et al., 2008). Although we found no significant relationship between atmospheric MSA$_{(p)}$ concentration, in situ sea surface Chl-a concentration, the relative biomass of *P. antarctica* and the local sea–air DMS flux, the higher atmospheric MSA$_{(p)}$ concentrations observed over the Amundsen Sea compared to those over the Southern Ocean and in coastal Antarctic regions most likely resulted from complex linkage between these factors." (page 8, line number 28−page 9, line number 10). Besides, we have removed from section 3.6 the following sentence: "As described in section 3.2, Kim et al. (2017) observed extremely high DMS concentrations (> 150 nM) in surface water during our cruise, and MSA$_{(p)}$ concentration showed a strong correlation with DMS flux in the Amundsen Sea." (this sentence was written on page 13, line number 20−22 in the previous manuscript). We also have modified Figure 4b (page 28).

In addition, Referee #7 pointed out that the MSA$_{(p)}$ concentrations observed over the Amundsen Sea appear to be the same level to those over the Southern Ocean if one data point (A9) is excluded. Although the higher atmospheric MSA$_{(p)}$ was mostly driven by one data point, we believe that we cannot rule out it. Besides, atmospheric MSA$_{(p)}$ observed over the Amundsen Sea showed a significant relationship with WSOC as shown in Fig. 8c. Consequently, it is reasonable to describe that the higher atmospheric MSA$_{(p)}$ level was observed over the Amundsen Sea, compared to those over the Southern Ocean.

As the authors acknowledged, higher nitrate concentrations in the aerosol phase cannot be attributed to the anthropogenic influences without further evidences. I would strongly recommend to remove the discussion from the manuscript.

(Response) We thank Referee #7 for Referee #7's comment. As Referee #7 pointed out, we have realized that higher aerosol NO$_3^-$ concentrations cannot be attributed to the anthropogenic influences without further evidences. Therefore, we have removed from section 3.3 the following sentences: "the statistically insignificant relationship between nss-SO$_4^{2-}$ and MSA in the Southern Ocean could result from the influence of anthropogenic sources in the Southern Ocean. During the cruise, as shown in Fig. 5a, high (> 0.1 µg m$^{-3}$) nitrate (NO$_3^-$) concentrations were observed during the collection of aerosol samples A1 and A13 when air masses originated from the Southern Ocean and thereafter swept over large regions of New Zealand (Fig. S4); however, NO$_3^-$ concentrations in the other samples were quite low (< 0.03 µg m$^{-3}$). Although NO$_3^-$ was observed in the other aerosol samples, we cannot assert that all aerosol samples were influenced by anthropogenic sources without another anthropogenic tracer data set (e.g., $^{210}$Pb) because, in addition to anthropogenic sources, NO$_3^-$ can be produced in clouds (Hegg and Hobbs, 1988) or derived from a weak oceanic source (Luo et al., 2018). Nevertheless, our results suggest that nss-SO$_4^{2-}$ in the two samples collected over the Southern Ocean were most likely affected by anthropogenic sources, resulting in the lack of correlation." (these sentences were written on page 10, line number 5−13 in the previous manuscript). We also have removed Figure 5(a) (page 29) and the related references (i.e., Hegg and Hobbs, 1988 and Luo et al., 2018) from the manuscript. Besides, to clarify the context, we have revised our manuscript as follows: "On the other hand, a possible explanation for the insignificant relationship between nss-SO$_4^{2-}$ and MSA$_{(p)}$ in the Southern Ocean is entrainment of nucleated nss-SO$_4^{2-}$ particles into the marine boundary layer of the Southern Ocean from the free troposphere by turbulent diffusion and large-scale transport." (page 10, line number 6−8). We also have revised "The lack of correlation between nss-SO$_4^{2-}$ and MSA$_{(p)}$ in the Southern Ocean, therefore, could have resulted from the influence of anthropogenic sources and the input of nss-SO$_4^{2-}$ from the free troposphere." to "The lack of correlation between nss-SO$_4^{2-}$ and MSA$_{(p)}$ in the Southern Ocean, therefore, could have resulted from the input of nss-SO$_4^{2-}$ from the free troposphere." (page 10, line number 11−12).

Finally, the discussion on attributing the protein-like substances in the aerosol phase to the BVOCs are highly misleading. BVOCs are by-products of photosynthetic processes but not biomaterial containing protein. Therefore, the discussion, appearing in Section 3.7 requires a substantial overhaul.

(Response) We are grateful to Referee #7 for Referee #7's comments. We have realized that the discussion on attributing the protein-like components in aerosols to the BVOCs is highly misleading, as Referee #7 pointed out. According to Referee #7's comment, we have revised the discussion on the relationship between the protein-like component and the relative biomass of diatoms as follows: "Ice algae, commonly found in polar sea ice and surrounding waters, are largely dominated by diatoms (Roberts et al., 2007), which are an important contributor to aerosols by emission of aerosol-forming volatile (e.g., alkyl-amines) and non-volatile (e.g., mycosporine-like amino acids) organic nitrogen in the Antarctic sea ice region (Dall'Osto et al., 2017). Previous studies (e.g., Facchini et al., 2008b; Miyazaki et al., 2011) provided the evidence that volatile emissions of alkyl-amine from marine algae can represent an important source of marine secondary organic aerosol. Moreover, Dall'Osto et al. (2017) observed that the fluorescence signal for protein-like component was positively correlated to organic nitrogen originated from the melted Antarctic sea ice floes, indicating that protein-like component was associated with organic nitrogen derived from the microbiota of sea ice and sea ice-influenced ocean. Although we have no aerosol water-soluble organic nitrogen dataset, our results provide additional evidence that marine algae can influence fluorescent property of marine aerosols. Interestingly, we found that fluorescence intensity of C1 showed a significant positive relationship with the relative biomass of diatoms ($r = 0.89$, $p < 0.01$); however, it was negatively correlated with the relative biomass of *P. antarctica* ($r = -0.79$, $p < 0.05$) (Figs. 10c and 10d). Given the dominance of diatoms in the marginal sea ice zone during the cruise and the significant positive relationship between fluorescence intensity of protein-like C1 and the relative biomass of diatoms, it is plausible, therefore, that biological processes of diatoms are an important factor in controlling the abundance of protein-like component in water-soluble organic aerosols over the Southern Ocean and the Amundsen Sea, although further studies are necessary to clarify this point." (page 14, line number 18−33).

Because we have found out that fluorescence properties in aerosol samples do not represent fluorescence properties of WSOC, but those of water-soluble organic aerosols, we have revised the title of section 3.7 from "Fluorescence properties of WSOC over the Southern Ocean and the Amundsen Sea" to "Fluorescence properties of water-soluble organic aerosols over the Southern Ocean and the Amundsen Sea" (page 13, line number 22). Consequently, we also have revised the sentences related to the revised title as follows:

1. "To further elucidate the sources of water-soluble organic aerosols, we investigated the fluorescence properties of submicron aerosols using EEM-PARAFAC. Fluorophores in water-soluble organic aerosols were divided into three primary types on the basis of their peak position." (page 13, line number 30−32).

2. "Moreover, the fluorescence properties of water-soluble organic aerosols revealed that protein-like components are most likely produced as a result of biological processes of diatoms." (page 15, line number 20−21).

3. "The fluorescence properties of water-soluble organic aerosols investigated using fluorescence excitation-emission matrix coupled with parallel factor analysis (EEM-PARAFAC) revealed that protein-like components were dominant in our marine aerosol samples, representing 69–91% of the total intensity." (page 1, line number 25−27).

4. "(4) estimate the source of atmospheric water-soluble organic aerosols using a fluorescence technique (page 3, line number 28−29).

We also have added the references (i.e., Dall'Osto et al., 2017; Facchini et al., 2008b; Roberts et al., 2007) to the manuscript.

**References**

Dall'Osto, M., Ovadnevaite, J., Paglione, M., Beddows, D. C. S., Ceburnis, D., Cree, C., Cortes, P.,

Zamanillo, M., Nunes, S. O., Perez, G. L., Ortega-Retuerta, E., Emelianov, M., Vaque, D., Marrase, C., Estrada, M., Sala, M. M., Vidal, M., Fitzsimons, M. F., Beale, R., Airs, R., Rinaldi, M., Decesari, S., Facchini, M. C., Harrison, R. M., O'Dowd, C. and Simo, R.: Antarctic sea ice region as a source of biogenic organic nitrogen in aerosols, Sci Rep, 7(1), 1–10, doi:10.1038/s41598-017-06188-x, 2017.

Facchini, M. C., Decesari, S., Rinaldi, M., Carbone, C., Finessi, E., Mircea, M., Fuzzi, S., Moretti, F., Tagliavini, E., Ceburnis, D. and O'Dowd, C. D.: Important source of marine secondary organic aerosol from biogenic amines, Environ. Sci. Technol., 42(24), 9116–9121, doi:10.1021/es8018385, 2008b.

Roberts, D., Craven, M., Cai, M., Allison, I. and Nash, G.: Protists in the marine ice of the Amery Ice Shelf, East Antarctica, Polar Biol., 30, 143–153, 2007.

---

## Author Response (AR5)

**Responses to Referees' comments**

30/03/2020

Journal: Atmospheric Chemistry and Physics

Title: Characteristics of methanesulfonic acid, non-sea-salt sulfate and organic carbon aerosols over the Amundsen Sea, Antarctica

Authors: Jinyoung Jung, Sang-Bum Hong, Meilian Chen, Jin Hur, Liping Jiao, Youngju Lee, Keyhong Park, Doshik Hahm, Jung-Ok Choi, Eun Jin Yang, Jisoo Park, Tae-Wan Kim, and SangHoon Lee

Manuscript number: acp-2019-133

Manuscript type: Research Article

Note: Referee's comments are high lightened in **blue**, while our responses to reviewers are in **black**. The revisions in the manuscript was highlighted to make easily readable for the Co-Editor and the Referees.

Prof. Lynn M. Russell

Co-Editor

Atmospheric Chemistry and Physics

USCD, Scripps Institution of Oceanography

United States

Dear Professor Russell,

We have attached an electronic copy of manuscript file ready to go to press entitled "Characteristics of methanesulfonic acid, non-sea-salt sulfate and organic carbon aerosols over the Amundsen Sea, Antarctica" by Jinyoung Jung, Sang-Bum Hong, Meilian Chen, Jin Hur, Liping Jiao, Youngju Lee, Keyhong Park, Doshik Hahm, Jung-Ok Choi, Eun Jin Yang, Jisoo Park, Tae-Wan Kim, and SangHoon Lee for publication in Atmospheric Chemistry and Physics (acp-2019-133). We have modified our text based on the Referees' comments. We appreciated that the comments from Referees improved our manuscript a lot. We believe that the comments from Referees were clearly responded in our modified manuscript.

We are looking forward to hearing about your decision. Thank you for your consideration.

Sincerely yours,

Jinyoung Jung

**Anonymous Referee #1:**

I do not understand the notation of these gas transfer velocity (k) values (W92, W99, N00, W14). Are these supposed to be velocity units? What equation was used for determining the uncertainty in DMS flux?

(Response) We thank Referee #1 for Referee #1's comments. The gas transfer velocity (k) values, W92, W99, N00, W14, indicate the gas transfer velocity values calculated from the equations suggested by Wanninkhof et al. (1992), Wanninkhof and McGillis (1999), Nightingale et al. (2000), and Wanninkhof (2014), respectively. The unit of gas transfer velocity is in cm hr$^{-1}$. We applied four different gas transfer values calculated from the equations suggested by Wanninkhof et al. (1992), Wanninkhof and McGillis (1999), Nightingale et al. (2000), and Wanninkhof (2014), respectively, to the equation for sea-air DMS flux calculation reported by Kim et al. (2017).

According to Referee #1's comments, we have revised our manuscript as follows: "When we applied four different k values (the units of k are in cm hr$^{-1}$) calculated from the equations suggested by Wanninkhof et al. (1992), Wanninkhof and McGillis (1999), Nightingale et al. (2000), and Wanninkhof (2014), respectively, to the equation for sea-air DMS flux calculation reported by Kim et al. (2017), the uncertainty in DMS flux in the Amundsen Sea region was about 25% (one standard deviation of the four different mean DMS fluxes)." (page 9, line number 10–14).

We also have added the references (i.e., Wanninkhof et al. (1992); Wanninkhof and McGillis (1999); Nightingale et al. (2000)) to our manuscript (page 24, line number 18–19; page 24, line number 22–23; page 22, line number 3–5).

References

Nightingale, P. D., Malin, G., Law, C. S., Watson, A. J., Liss, P. S., Liddicoat, M. I., Boutin, J. and Upstill-Goddard, R. C.: In situ evaluation of air-sea gas exchange parameterizations using novel conservative and volatile tracers, Global Biogeochem. Cycles, 14(1), 373–387, doi:10.1029/1999GB900091, 2000.

Wanninkhof, R.: Relationship between wind-speed and gas-exchange over the ocean, J. Geophys. Res., 97(C5), 7373–7382, doi:10.1029/92JC00188, 1992.

Wanninkhof, R. and McGillis, W. R.: A cubic relationship between air-sea $CO_2$ exchange and wind speed, Geophys. Res. Lett., 26(13), 1889–1892, doi:10.1029/1999GL900363, 1999.